# Dermal macrophages set pain sensitivity by modulating the amount of tissue NGF through an SNX25–Nrf2 pathway

Tatsuhide Tanaka [1] ✉, Hiroaki Okuda [2], Ayami Isonishi[1], Yuki Terada[1], Masahiro Kitabatake[3], Takeaki Shinjo[1], Kazuya Nishimura[1], Shoko Takemura[1], Hidemasa Furue[4], Toshihiro Ito[3], Kouko Tatsumi[1] & Akio Wanaka [1] ✉

Cross-talk between peripheral neurons and immune cells is important in pain sensation. We identified *Snx25* as a pain-modulating gene in a transgenic mouse line with reduced pain sensitivity. Conditional deletion of *Snx25* in monocytes and macrophages, but not in peripheral sensory neurons, in mice (*Snx25*^cKO mice) reduced pain responses in both normal and neuropathic conditions. Bone marrow transplantation using *Snx25*^cKO and wild-type mice indicated that macrophages modulated pain sensitivity. Expression of sorting nexin (SNX)25 in dermal macrophages enhanced expression of the neurotrophic factor NGF through the inhibition of ubiquitin-mediated degradation of Nrf2, a transcription factor that activates transcription of *Ngf*. As such, dermal macrophages set the threshold for pain sensitivity through the production and secretion of NGF into the dermis, and they may cooperate with dorsal root ganglion macrophages in pain perception.

The skin is frequently stressed by mechanical trauma. Sensory stimuli impinging on skin are encoded by peripheral sensory neurons that can be classified into low-threshold mechanoreceptors, which detect innocuous tactile stimuli, and nociceptors, which exclusively respond to harmful stimuli[1,2]. Small-diameter neurons of the dorsal root ganglion (DRG) are pain-sensing neurons, while medium- to large-diameter neurons preferentially detect low-threshold mechanical stimulation[3]. Skin damage leads to the release of inflammatory mediators by activated nociceptors or by nonneural cells that reside within or infiltrate the injured area, including macrophages, mast cells and keratinocytes. Tissue macrophages can be divided into nerve-associated and blood vessel-associated subsets[4]. A subset of skin macrophages is closely associated with peripheral nerves and promotes their regeneration when damaged[5]. In neuropathic conditions, macrophages can accelerate pain sensation by sensing tissue angiotensin 2 (ref. [6]) or complement 5a[7,8].

NGF is a small, secreted protein and a member of the neurotrophin family of growth factors. NGF modulates pain sensation in several acute and chronic pain states[9]. NGF is expressed in immune cells, including macrophages[7], and facilitates pain transmission by sensory neurons through a variety of mechanisms. NGF enhances the activity, gene expression and membrane localization of nociceptive ion channels, which increase sensory neuron excitability[9]. In humans, mutations in the *NGF* gene cause hereditary sensory and autonomic neuropathy type V (HSAN V) (OMIM 608654), characterized by a marked absence of pain sensibility[10]. Mouse models of HSAN V show a significant reduction of sensory innervations, which leads to decreased pain perception[11]. However, the mechanisms underlying the regulation of NGF remain to be determined.

In a serendipitously discovered pain-insensitive transgenic mouse line, forward genetic analyses identified *Snx25* as a pain-modulating

[1]Department of Anatomy and Neuroscience, Faculty of Medicine, Nara Medical University, Kashihara, Japan. [2]Department of Functional Anatomy, Graduate School of Medical Science, Kanazawa University, Kanazawa, Japan. [3]Department of Immunology, Faculty of Medicine, Nara Medical University, Kashihara, Japan. [4]Department of Neurophysiology, Hyogo College of Medicine, Nishinomiya, Japan. ✉e-mail: ttanaka@naramed-u.ac.jp; akiow@naramed-u.ac.jp

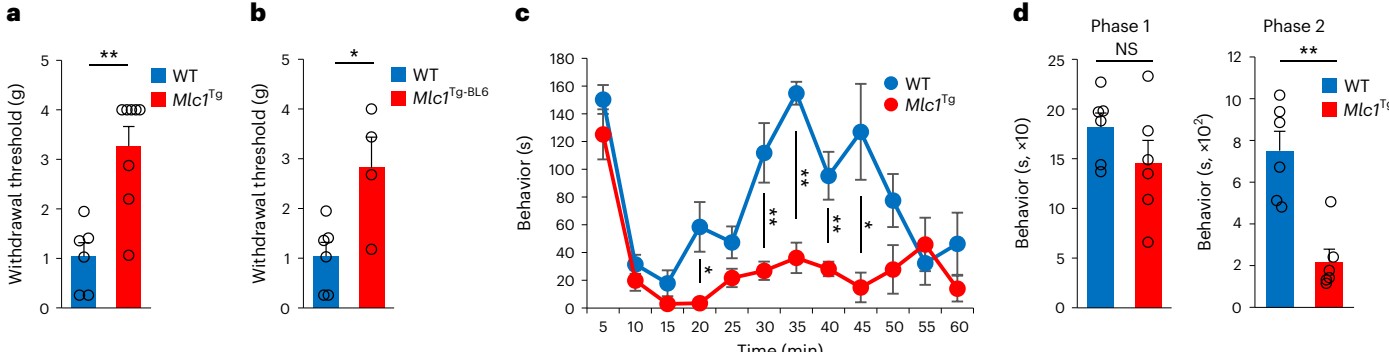

**Fig. 1 | $Mlc1^{Tg}$ mice were insensitive to pain. a**, Comparison of paw-withdrawal thresholds to mechanical stimulation with von Frey's filaments between WT ($n = 6$) and $Mlc1^{Tg}$ mice on a mixed 129S6–CBA–C57BL/6J background ($Mlc1^{Tg}$; $n = 8$). $P = 0.001$. g, gram. **b**, VF thresholds in $Mlc1^{Tg}$ mice backcrossed with C57BL/6J mice for seven generations (WT, $n = 6$; $Mlc1^{Tg-BL6}$, $n = 4$). $P = 0.017$. g, gram. **c**, Formalin responses plotted for 5-min periods in WT ($n = 6$) and

$Mlc1^{Tg}$ ($n = 6$) mice. s, second. **d**, Duration of pain-related behavior during phase 1 (0–10 min) (left, $P = 0.208$) and phase 2 (20–60 min) (right, $P = 0.001$) of the response in mice as in **c** (WT, $n = 6$; $Mlc1^{Tg}$, $n = 6$). s, second. Results are represented as mean ± s.e.m. Statistical significance was calculated using two-tailed Student's $t$-test. *$P < 0.05$, **$P < 0.01$; NS, not significant.

gene. SNX proteins are involved in membrane trafficking, cell signaling and organelle motility[12]. Here we show that expression of SNX25 in dermal macrophages (hereafter dMacs) induced the production of NGF by inhibiting ubiquitin-mediated degradation of Nrf2, a CNC-bZIP transcription factor that activates *Ngf* mRNA transcription[13]. SNX25 expressed in dMacs modulated acute pain sensing under both normal and pain-inducing conditions by signaling through the NGF–TrkA (tropomyosin receptor kinase A, its receptor) pathway. As such, macrophage–neuron signaling is important in pain processing in naive skin and in neuropathic or inflammatory situations.

## Results

### $Snx25^{+/-}$ mice showed a pain-insensitive phenotype

During handling and genotyping, we serendipitously found that pain responses to mechanical stimuli were reduced in mice with transgenic expression of a gene associated with a congenital leukoencephalopathy, *Mlc1* ($Mlc1^{Tg}$ mice[14], for details, see Methods) compared to C57BL/6J mice (Fig. 1a). Because $Mlc1^{Tg}$ mice were on a mixed 129S6, CBA and C57BL/6J background, to exclude the possibility that the mixed background contributed to the pain insensitivity, we backcrossed them with C57BL/6J mice for seven generations. The withdrawal threshold to mechanical stimuli by von Frey's filaments (hereafter, VF threshold) was increased in $Mlc1^{Tg}$ mice backcrossed to C57BL/6J mice compared to C57BL/6J mice under normal conditions without neuropathic or inflammatory stimuli (Fig. 1b). Pain responses to intradermal injection of 5% formalin, which induces acute inflammatory pain, such as shaking and licking of paws (hereafter formalin responses) were significantly reduced in $Mlc1^{Tg}$ mice compared to C57BL/6J mice (Fig. 1c,d). Formalin injection into the hind paw skin resulted in less c-Fos$^+$ activated neurons in the dorsal horn of the L4 spinal cord, which receives hind paw sensation, in $Mlc1^{Tg}$ mice than in wild-type (WT) mice (Extended Data Fig. 1a–c). $Mlc1^{Tg}$ mice harbor a bacterial artificial chromosome (BAC) transgene (clone RP23-114I6, 198 kb), and next-generation sequencing of genomic DNA indicated the insertion of the BAC transgene (83 kb of 198 kb) into 8qB1.1 of chromosome 8, resulting in the deletion of three genes (*Snx25*, *Slc25a4* and *Cfap97*) (Extended Data Fig. 1d). Quantitative PCR with reverse transcription (RT–qPCR) analysis indicated that expression of BAC-borne *Mlc1* and *Mov10l1* was indistinguishable from that of WT mice (Extended Data Fig. 1e,f), while complementary DNA (cDNA) microarray analyses indicated that expression of *Snx25*, *Slc25a4* and *Cfap97* was almost null (Extended Data Fig. 1g–i). To investigate the role of *Snx25* in regulating pain sensation, we used mice in which an *En2SA*-IRES-*lacZ* cassette is inserted upstream of exon 4 of

*Snx25* to create a null allele by splicing and premature termination of the transcript (*Snx25*-knockout (KO) mice; Extended Data Fig. 2a) and which allow monitoring of SNX25 expression by the β-galactosidase (LacZ) reporter[15]. Expression of SNX25 in the lungs of $Snx25^{+/-}$ mice was approximately 50% of that in WT mice (Fig. 2a). The VF thresholds (Fig. 2b) in $Snx25^{+/-}$ male mice were elevated compared to those of WT mice and similar to those in $Mlc1^{Tg}$ mice. Formalin responses were reduced in $Snx25^{+/-}$ mice compared to those in WT mice (Fig. 2c,d). Although thermal nociception was not affected in 2-month-old $Snx25^{+/-}$ mice, 6- to 8-month-old $Snx25^{+/-}$ mice had longer latency to respond to heat stimuli (Extended Data Fig. 2b). Mechanical hypersensitivity induced by spared nerve injury (SNI) was significantly attenuated in $Snx25^{+/-}$ mice compared to in WT mice (Fig. 2e). Cellular size distribution (Extended Data Fig. 2c) and expression of CGRP$^+$ small sensory neurons (Extended Data Fig. 2d) and NF200$^+$ large sensory neurons (Extended Data Fig. 2e) in the DRG were similar in adult $Snx25^{+/-}$ mice and WT mice, indicating that abnormal reactions to pain stimuli were not the result of the loss of neuronal subsets. Because the pain-insensitive phenotype of $Snx25^{+/-}$ mice was apparent beyond 3 weeks of age (Extended Data Fig. 3a), we examined whether it was due to impaired neuronal development. Cellular size distribution and expression of CGRP and NF200, which are small and large neuron markers, respectively, in the DRG were similar in 3-week-old $Snx25^{+/-}$ mice and WT mice (Extended Data Fig. 3b–e). To test whether SNX25 deficiency reduced sprouting and/or arborization of peripheral sensory fibers, we compared protein gene product (PGP)9.5$^+$ sensory fibers in the dermis of WT and $Snx25^{+/-}$ mice. The area of PGP9.5$^+$ fibers in the hind paw skin of $Snx25^{+/-}$ mice was comparable to that of WT mice at 2 months of age (Extended Data Fig. 3f,g).

To further investigate the role of SNX25 in pain sensation, we examined expression of pain-related factors in $Snx25^{+/-}$ mice. Consistent with the pain-insensitive phenotype, expression of transient receptor potential cation channel, subfamily V, member 1 (TRPV1) and TrkA, which are involved in pain sensation, was downregulated in the DRG (Fig. 2f–i), the sciatic nerve and the spinal cord (Extended Data Fig. 3h,i) of $Snx25^{+/-}$ mice compared to WT mice. Capsaicin, an active component of chili peppers that stimulates TRPV1 channels and produces a sensation of pain, elevated the intracellular Ca$^{2+}$ level in cultured primary DRG neurons, but the amplitude of this Ca$^{2+}$ elevation was smaller in $Snx25^{+/-}$ neurons than in WT neurons (Fig. 2j), indicating reduced expression of the TRPV1 channel in SNX25-deficient DRG neurons. mRNA expression of *Trpv1*, *Scn9a* (encoding Na$_v$1.7) and *Scn10a* (encoding Na$_v$1.8), all of which are related to pain perception[9], was reduced in $Snx25^{+/-}$ DRGs compared to WT DRGs (Fig. 2k).

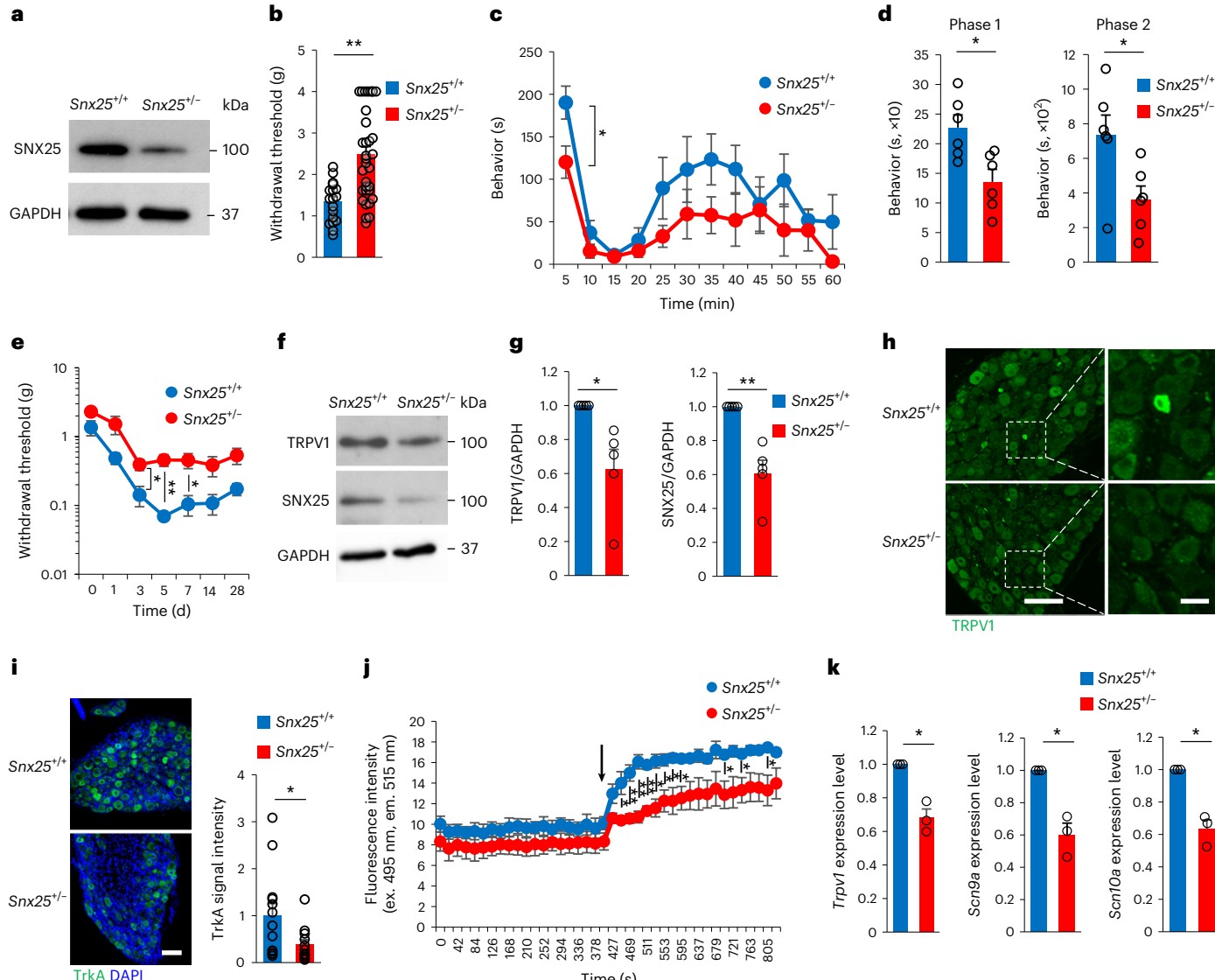

**Fig. 2 | *Snx25*+/− mice showed a pain-insensitive phenotype. a**, Immunoblot showing the expression level of SNX25 in the lung of WT and *Snx25*+/− mice. **b**, VF thresholds of WT and *Snx25*+/− mice (WT, *n* = 19; *Snx25*+/−, *n* = 33). *P* = 7.844 × 10⁻⁵. g, gram. **c**, Pain-related behavior time plotted for 5-min periods in WT (*n* = 6) and *Snx25*+/− (*n* = 6) mice with injection of formalin into hind paws. s, second. **d**, Pain-related behavior time during phase 1 (0–10 min, *P* = 0.01) and phase 2 (20–60 min, *P* = 0.029) in mice as in **c**. s, second. **e**, VF thresholds plotted after SNI in WT and *Snx25*+/− mice (WT, *n* = 4; *Snx25*+/−, *n* = 7) at day 3 (*P* = 0.032), day 5 (*P* = 0.008) and day 7 (*P* = 0.04). g, gram. **f**, Representative immunoblots showing expression of TRPV1 and SNX25 in the DRG of WT and *Snx25*+/− mice. **g**, Semi-quantitative analyses of immunoblots of TRPV1 and SNX25 in DRGs from WT (*n* = 5) and *Snx25*+/− (*n* = 5) mice. TRPV1, *P* = 0.033; SNX25, *P* = 0.007. **h**, Confocal microscopy of the DRG stained with anti-TRPV1 antibody in WT and *Snx25*+/− mice. Scale bar, 100 μm. Right, magnified views of boxed areas in the corresponding left panels. Representative of three independent experiments.

Scale bar, 20 μm. **i**, Confocal microscopy of the DRG of WT and *Snx25*+/− mice, stained with anti-TrkA antibody (left; scale bar, 100 μm) and quantification of mean TrkA fluorescence intensity (WT, *n* = 13; *Snx25*+/−, *n* = 12 DRG sections from four different mice) (right). *P* = 0.042. Representative of three independent experiments. DAPI, 4,6-diamidino-2-phenylindole. **j**, Fluo-4 Ca²⁺ imaging of primary DRG neurons from an entire well dissociated from WT and *Snx25*+/− mice (WT, *n* = 3; *Snx25*+/−, *n* = 3). The arrow indicates the time when capsaicin was added to a well. *P* values are as follows: 448 s, *P* = 0.033; 469 s, *P* = 0.008; 490 s, *P* = 0.04; 511 s, *P* = 0.002; 532 s, *P* = 0.014; 553 s, *P* = 0.03; 574 s, *P* = 0.038; 595 s, *P* = 0.034; 700 s, *P* = 0.046; 742 s, *P* = 0.049; 805 s, *P* = 0.049. ex., excitation; em., emission. **k**, mRNA expression for pain-related factors in the DRG of WT and *Snx25*+/− mice (WT, *n* = 3; *Snx25*+/−, *n* = 3). *Trpv1*, *P* = 0.026; *Scn9a*, *P* = 0.032; *Scn10a*, *P* = 0.022. Results are represented as mean ± s.e.m. Statistical significance was calculated using two-tailed Student's *t*-test (**b**–**e**, **i** and **j**) or two-tailed Welch's *t*-test (**g**,**k**). **P* < 0.05, ***P* < 0.01.

These observations indicated that the pain-insensitive phenotype of *Snx25*+/− mice was due to reduced expression of pain-related factors in peripheral sensory neurons.

## DRG-specific *Snx25*cKO mice are sensitive to pain

To define cells responsible for the pain-insensitive phenotype of *Snx25*+/− mice, we generated conditional alleles by removing the KO cassette in *Snx25*+/− mice with flippase (FLP), leaving *loxP* sites on either side of the critical exon 4 (ref. [15]) (*Snx25*fl/fl mice, Extended Data Fig. 4a). VF thresholds (Extended Data Fig. 4b) and formalin responses (Extended Data Fig. 4c) reverted to normal levels in *Snx25*fl/fl mice, indicating that *Snx25* deletion mediated the pain-insensitive phenotype. Next, we conditionally deleted *Snx25* in the DRG by crossing *Snx25*fl/fl mice with *Advillin* (*Avil*)CreERT2 mice[16] and administered 0.05% tamoxifen (TAM) orally for 2 weeks (Extended Data Fig. 4d) to induce recombination[17,18]. TAM administration in *Avil*CreERT2/WT*Snx25*fl/fl mice (hereafter

$Snx25^{Avil\text{-}cKO}$ mice) markedly reduced expression of SNX25 in the DRG at week 3 after the first TAM feed, compared to that in $Snx25^{fl/fl}$ mice (Extended Data Fig. 4e). $Snx25^{Avil\text{-}cKO}$ mice had normal VF thresholds and formalin responses (Extended Data Fig. 4f–h) and normal expression of $Trpv1$, $Scn9a$ and $Scn10a$ mRNA (Extended Data Fig. 4i). These results suggested that SNX25 in the DRG did not regulate expression of pain-related factors or pain sensation.

### SNX25 in bone marrow-derived macrophages modulated pain sensation

A population of dermal major histocompatibility complex (MHC)-II[+], CD206[+] or F4/80[+] macrophages was SNX25[+] (Extended Data Fig. 5a) and was closely associated with PGP9.5[+] sensory fibers (Fig. 3a and Extended Data Fig. 5b,c) compared with other myeloid cells tested (CD117[+] mast cells, CD4[+] helper T cells, CD8a[+] killer T cells, CD19[+] B cells, NK1.1[+] NK cells and Gr1[+] or Ly6G[+] neutrophils) (Extended Data Fig. 5b–d). Immunohistochemistry indicated that the number of MHC-II[-]CD206[+]F4/80[+] macrophages (Extended Data Fig. 5e,f) and expression of CD206 (Extended Data Fig. 5g,h) were similar in the hind paw skin of $Snx25^{+/-}$ and WT mice. Transmission electron microscopy showed that there was no significant difference in overall morphology between bone marrow (BM)-derived macrophages (BMDMs) of $Snx25^{+/-}$ and WT mice (Extended Data Fig. 5i).

dMacs are replenished by BM-derived cells[5,19,20]. To confirm these features, we intravenously injected BM cells from green fluorescent protein (GFP) mice (Methods) into WT mice pretreated with the alkylating agent busulfan, which ablates BM cells[21]. At week 10 after BM transplantation (BMT), 78% of leukocytes in the peripheral blood were of donor origin (Fig. 3b and Extended Data Fig. 6a) and the donor-derived dMacs were MHC-II[+], CD206[+], F4/80[+] or Lyve1[+] (Fig. 3c and Extended Data Fig. 6b–d) and $Cx3cr1$ mRNA[+] (Fig. 3d). GFP[+] cells were predominant in MHC-II[+] cells and F4/80[+] cells (Extended Data Fig. 6e,f), while only a few donor-derived GFP[+]Gr1[+] neutrophils, CD19[+] B cells, CD8a[+] T cells, CD4[+] T cells and NK1.1[+] NK cells were detected in the hind paw skin of recipient mice at week 5 after BMT (Extended Data Fig. 6g–k). Flow cytometry indicated that the majority of MHC-II[+] or F4/80[+] dMacs were replaced by donor-derived GFP[+]MHC-II[+] or GFP[+]F4/80[+] cells in recipient mice (Fig. 3e,f), while other populations in GFP[+] cells were rare in the hind paw skin (Extended Data Fig. 6l) and the back skin (Extended Data Fig. 6m) at week 10 after BMT. GFP[+] cells were not found in the gray matter of the spinal dorsal horn (Extended Data Fig. 6n), consistent with reports that spinal cord microglia are not derived from BM in the adult[22].

To gain further insight into the contribution of dMacs to pain sensation, we made BM chimera by transplanting WT or $Snx25^{+/-}$ BM cells into $Snx25^{+/-}$ or WT mice, respectively. At day 28 after BMT, we observed an increase in VF thresholds in WT mice that received $Snx25^{+/-}$ BMT and a reduction of the thresholds in $Snx25^{+/-}$ mice that received WT BM cells (Fig. 3g). Mice treated solely with busulfan had normal VF thresholds (Extended Data Fig. 6o). BMT between the same genotypes (WT to WT or $Snx25^{+/-}$ to $Snx25^{+/-}$) did not also affect VF thresholds (Extended Data Fig. 6p).

Next, we investigated the role of SNX25 in dMacs in an inflammatory environment. Immunohistochemistry indicated fewer Iba1[+]CD206[+] dMacs (Extended Data Fig. 7a,b) and lower expression of a cluster of chemokines (Extended Data Fig. 7c) at day 3 after formalin injection into $Snx25^{+/-}$ mice compared to WT mice. We also observed an upregulation of transforming growth factor (TGF)-β receptor 1 (TGF-βR1), which has a suppressive role during immune responses[23] and is known to be degraded by SNX25 (ref. [24]), in the hind paw skin of $Snx25^{+/-}$ mice compared to WT mice (Extended Data Fig. 7d). Immunohistochemistry indicated lower accumulation of CCR2[+] infiltrating immune cells in the hind paw skin and DRG of $Snx25^{+/-}$ mice than in WT mice at day 7 after formalin injection, albeit the differences were not statistically significant (Extended Data Fig. 7e,f). These observations indicated that loss of SNX25 in BM-derived dMacs affected pain sensation under steady state and in inflammatory conditions.

### Macrophage-specific $Snx25^{cKO}$ mice were insensitive to pain

Next, we crossed $Snx25^{fl/fl}$ mice with $Cx3cr1^{CreERT2/WT}$ mice[25] to generate mice with conditional deletion of SNX25 in monocytes and macrophages (hereafter $Snx25^{Cx3cr1\text{-}cKO}$). $Snx25^{Cx3cr1\text{-}cKO}$ mice showed elevated VF thresholds (Fig. 4a) and reduced formalin responses (Fig. 4b) compared to $Snx25^{fl/fl}$ mice. Expression of $Scn9a$ and $Scn10a$ was reduced in the DRG of $Snx25^{Cx3cr1\text{-}cKO}$ mice compared to $Snx25^{fl/fl}$ mice (Fig. 4c), while the size distribution of DRG neurons was normal in $Snx25^{Cx3cr1\text{-}cKO}$ mice (Extended Data Fig. 7g). Expression of $Cxcl5$, $Cxcl2$, $Il1b$ and $Cxcl3$ mRNA was lower in the hind paw skin of $Snx25^{Cx3cr1\text{-}cKO}$ mice than in $Snx25^{fl/fl}$ mice (Fig. 4d and Extended Data Fig. 7h). Expression of $Ccl2$, $Ccl3$, $Ccl4$ and $Cxcl2$ mRNA was also lower in CD45[+]CD11b[+]F4/80[+] cells sorted from the skin of $Snx25^{Cx3cr1\text{-}cKO}$ mice than in $Snx25^{fl/fl}$ mice (Extended Data Fig. 7i–k), although the proportion of CD11b[+]F4/80[+] cells in the hind paw skin did not change (Extended Data Fig. 7l). These results indicated that SNX25 contributed to the inflammatory response in dMacs after chemical stimulation as well as to pain sensation at steady state.

Lyve1[lo]MHC-II[hi]CX3CR1[hi] macrophages where shown to colocalize with peripheral nerves[4]. To determine the relationship between SNX25[+] dMacs and peripheral nerves, we crossed $Snx25^{Cx3cr1\text{-}cKO}$ mice with Ai39[Tg/+] mice (Methods; hereafter $Snx25^{Cx3cr1\text{-}cKO}$;Ai39[Tg/+] mice)[26]. TAM administration resulted in yellow fluorescent protein (YFP) expression in CX3CR1[+]MHC-II[+] dMacs but not in CD117[+] mast cells (Fig. 4e). In the dermis of $Snx25^{Cx3cr1\text{-}cKO}$;Ai39[Tg/+] mice, YFP[+]CX3CR1[+] dMacs were apposed to PGP9.5[+] fibers (Fig. 4f), suggesting close association of SNX25[+] dMacs with peripheral sensory fibers. Immunohistochemistry indicated that F4/80[+] cells and CD206[+] cells were colocalized with PGP9.5[+] peripheral nerves in the skin of WT mice (Fig. 4g).

CX3CR1 is expressed by central nervous system microglia[27], which regulate neuronal and synaptic activities to change pain behavior[28]. To test whether microglia contributed to the pain insensitivity in $Snx25^{Cx3cr1\text{-}cKO}$ mice, we made BM chimera by BMT from $Snx25^{Cx3cr1\text{-}cKO}$ mice into $Snx25^{fl/fl}$ mice and treated them with TAM orally for 2 weeks. VF thresholds were significantly increased in TAM-treated BM chimera compared to those before BMT (Fig. 4h), while the same experimental condition without TAM treatment yielded thresholds comparable to

---

**Fig. 3 | SNX25 in macrophages derived from BM contributed to pain sensation. a**, Confocal microscopy of the plantar skin of the naive hind paw of WT mice, immunolabeled for PGP9.5 and MHC-II. Representative of three independent experiments. Scale bar, 50 μm. **b**, Replacement rates of myeloid cells by transplanted BM of GFP mice in peripheral blood plotted against time after BMT ($n = 4$). **c**, Confocal microscopy of hind paw skin labeled for GFP and MHC-II in WT mice that received BM from GFP mice imaged at weeks 1, 3, 4, 5, 7 and 10 after BMT. Arrowheads denote double-labeled cells. Representative of three independent experiments. Scale bar, 100 μm. **d**, Confocal microscopy of hind paw skin labeled for GFP (Alexa 594) and $Cx3cr1$ mRNA (fluorescent in situ hybridization) in WT mice that received BM from GFP mice at week 5 after transplantation. Arrowheads show BM-derived GFP[+] cells positive for $Cx3cr1$

mRNA. Representative of two independent experiments. Scale bar, 100 μm. Bottom, magnified views of the boxed area in the upper panel. Scale bar, 50 μm. **e**, Flow cytometry strategy to sort donor-derived macrophages (MHC-II[+] or F4/80[+]) using propidium iodide (PI), CD45, F4/80, MHC-II and GFP expression from hind paw skins of WT mice that received BMT from GFP mice. FSC, forward scatter; SSC, side scatter. **f**, Percentage of GFP[+] cells among MHC-II[+] or F4/80[+] cells. Results are presented as mean ± s.e.m. of three different mice that received BMT from GFP mice. Values are 85.1% and 80.8%, respectively. **g**, VF thresholds in $Snx25^{+/-}$ → WT BM chimeras ($n = 10$, $P = 0.015$) and WT → $Snx25^{+/-}$ BM chimeras ($n = 13$, $P = 0.049$) at day 28 after BMT. g, gram. Results are represented as mean ± s.e.m. Statistical significance was calculated using two-tailed Student's $t$-test. *$P < 0.05$.

those of $Snx25^{Cx3cr1\text{-}cKO}$ mice before BMT (Fig. 4h). During SNI, mechanical hypersensitivities were attenuated in TAM-treated BM chimeric mice that received $Snx25^{Cx3cr1\text{-}cKO}$ BMT compared to those without TAM

(Fig. 4i). These results indicated that SNX25 in dMacs, but not microglia, modulated pain sensitivity under steady-state and neuropathic conditions.

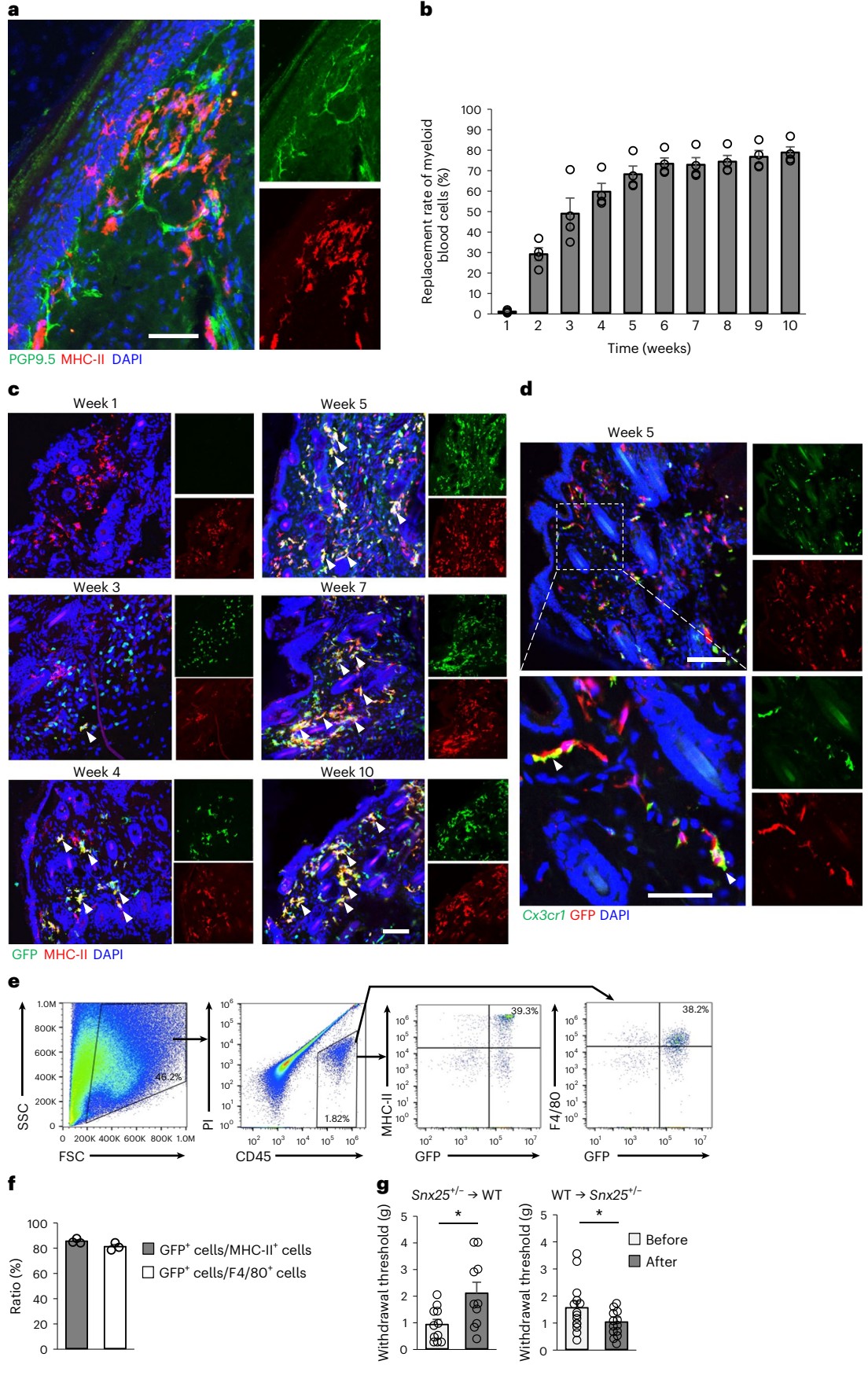

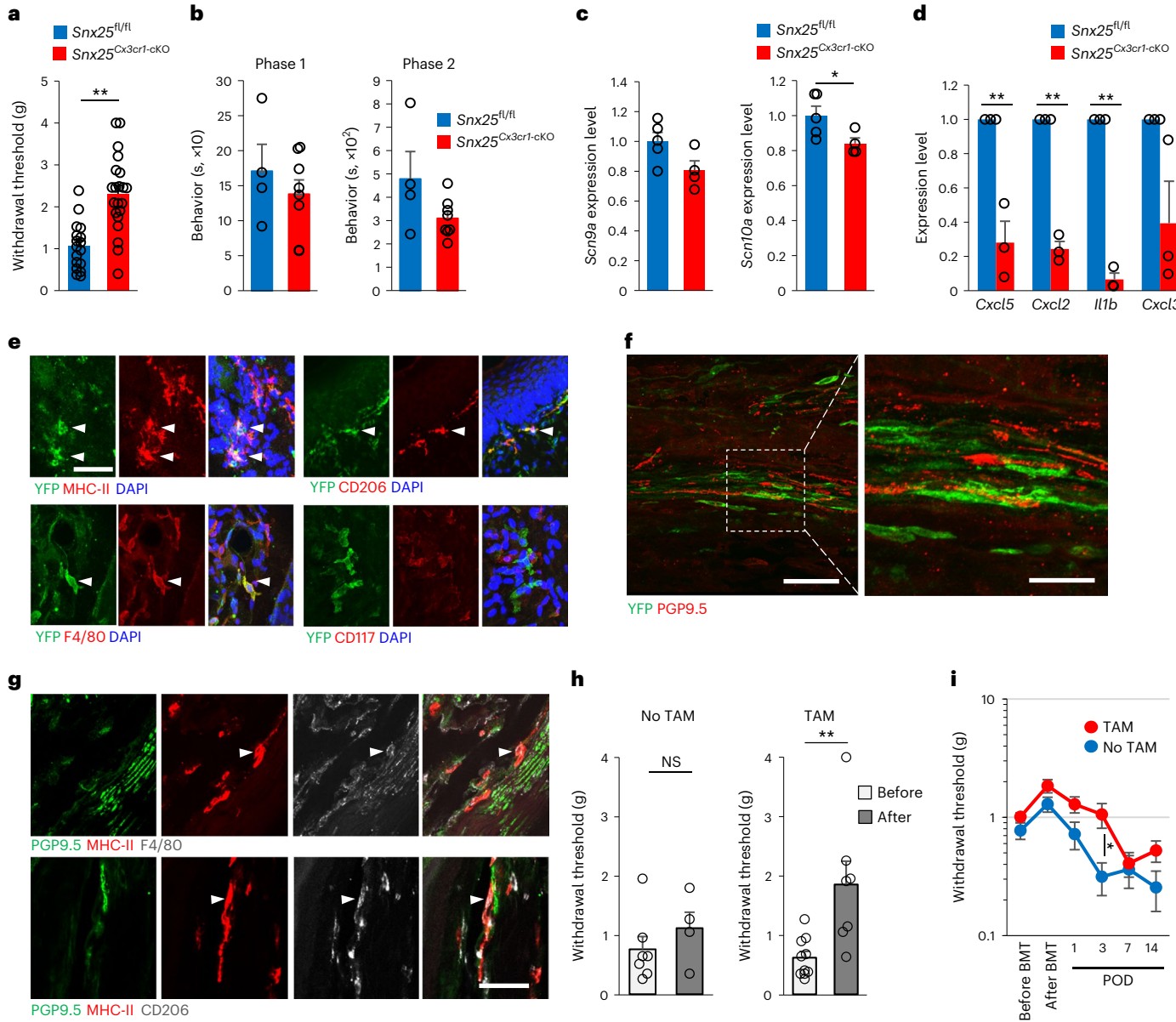

**Fig. 4 | Snx25 conditional KO in macrophages yielded a pain-insensitive phenotype. a**, VF thresholds in $Snx25^{Cx3cr1-cKO}$ mice ($n = 17$) and $Snx25^{fl/fl}$ mice ($n = 25$) treated with TAM for 2 weeks. $P = 2.61 × 10^{-5}$. g, gram. **b**, Formalin responses in $Snx25^{Cx3cr1-cKO}$ mice ($n = 8$) and $Snx25^{fl/fl}$ mice ($n = 4$) treated with TAM for 2 weeks. Pain-related behaviors are plotted for phase 1 (0–10 min, $P = 0.421$) and phase 2 (20–60 min, $P = 0.089$). s, second. **c**, Expression of mRNA encoding Na$^+$ channels in DRGs of $Snx25^{Cx3cr1-cKO}$ mice ($n = 4$) and $Snx25^{fl/fl}$ mice ($n = 5$). $Scn9a$, $P = 0.064$; $Scn10a$, $P = 0.049$. **d**, Quantification of $Cxcl5$, $Cxcl2$, $Il1b$ and $Cxcl3$ mRNA by RT–qPCR in the hind paw skin from $Snx25^{fl/fl}$ mice ($n = 3$) and $Snx25^{Cx3cr1-cKO}$ mice ($n = 3$). $Cxcl5$, $P = 0.004$; $Cxcl2$, $P = 6.81 × 10^{-5}$; $Il1b$, $P = 1.444 × 10^{-5}$; $Cxcl3$, $P = 0.069$. **e**, Confocal microscopy of naive hind paw skin from $Snx25^{Cx3cr1-cKO}$; Ai39$^{Tg/+}$ mice expressing YFP (green) and MHC-II, F4/80, CD206 (macrophage markers, red) or CD117 (mast cell marker, red). Arrowheads denote double-labeled cells. Scale bar, 50 μm. **f**, Confocal microscopic images of the hind paw skin of $Snx25^{Cx3cr1-cKO}$;Ai39$^{Tg/+}$ mice stained for YFP (green) and PGP9.5 (red). Scale bar, 50 μm. Right, magnified view of the boxed area. Scale bar, 20 μm. **g**, Confocal microscopy of the hind paw skin stained for PGP9.5 (green), MHC-II (red) and F4/80 or CD206 (white) in WT mice. Scale bar, 50 μm. **h**, VF thresholds plotted for $Snx25^{fl/fl}$ mice before and after BMT from $Snx25^{Cx3cr1-cKO}$ mice without (no TAM, $n = 7$, $P = 0.35$) or with (TAM, $n = 10$, $P = 0.005$) TAM treatment at day 35 after BMT. g, gram. **i**, Establishment and time course of mechanical allodynia plotted after BMT and SNI in BM chimeric mice as in **h** (no TAM, $n = 13$; TAM, $n = 18$). Three days, $P = 0.012$. POD, postoperative day. g, gram. Results are represented as mean ± s.e.m. Significance was calculated using two-tailed Student's t-test (**a–c,h,i**) or two-tailed Welch's t-test (**d**). *$P < 0.05$, **$P < 0.01$. Representative of three independent experiments (**e–g**).

## SNX25 regulated pain sensitivity through NGF signaling

Mutations in *Ngf* cause painless phenotypes[29,30], and NGF derived from immune cells regulates the expression of *Trpv1*, *Scn9a* and *Scn10a*[30]. As such, we tested whether the NGF concentration in the dermis was partly maintained by dMacs. Expression of NGF in the hind paw skin was lower at steady state (Fig. 5a) and at 30 min after formalin injection (Fig. 5b) in $Snx25^{+/-}$ mice than in WT mice. NGF was expressed in WT MHC-II$^+$, F4/80$^+$ and Iba1$^+$ dMacs in situ (Fig. 5c), and its expression was reduced in $Snx25^{+/-}$ BMDMs (Fig. 5d). The sciatic nerve was analyzed 8 h after nerve ligation to assess the cumulative axonal transport rate of TrkA, the cognate receptor for NGF[31]. Immunohistochemistry showed that accumulation of TrkA on the distal side of the nerve ligature was significantly reduced in $Snx25^{+/-}$ nerves compared to that in WT mice (Fig. 5e), suggesting diminished retrograde transport of the

NGF–TrkA complex in the $Snx25^{+/-}$ DRG. RT–qPCR showed decreased *Ngf* mRNA in $Snx25^{+/-}$ BMDMs (Fig. 5f) and in BMDMs in which *Snx25* was knocked down with small interfering RNA (siRNA) compared to a scramble siRNA control (Fig. 5g). To further test the role of SNX25 in regulating macrophage-derived NGF, we crossed $Snx25^{Cx3cr1-cKO}$ mice with Ai32$^{Tg/+}$ mice (Methods) to generate $Snx25^{Cx3cr1-cKO}$;Ai32$^{Tg/+}$ mice, in which *Snx25*-KO macrophages can be tracked by YFP expression. Treatment with the soluble TAM derivative 4-OH TAM (4-OHT) induced YFP expression in $Snx25^{Cx3cr1-cKO}$;Ai32$^{Tg/+}$ BMDMs (Fig. 5h,i). *Ngf* mRNA was reduced in sorted Ai32$^{Tg/+}$-derived YFP$^+$ BMDMs compared to YFP$^-$ BMDMs (Fig. 5j), indicating that SNX25 modulated the expression of *Ngf* mRNA. In WT mice that received BM from $Snx25^{Cx3cr1-cKO}$;Ai32$^{Tg/+}$ mice, Ai32$^{Tg/+}$-derived YFP expression was detected in 44% of MHC-II$^+$ dMacs in situ following oral TAM treatment (Fig. 5k). Immunoblotting showed that NGF expression was lower in the hind paw skin of $Snx25^{Cx3cr1-cKO}$ mice than in $Snx25^{fl/fl}$ mice (Extended Data Fig. 8a).

dMacs were defined by expression of CD64 (Fc-γ receptor) in lineage (Lin; CD3, CD19, Ly6G, NK1.1, TER119, CD24)$^-$CD45$^+$CD11b$^+$Ly6C$^-$ cells and subdivided by the expression of MHC-II[5,20]. To investigate *Ngf* mRNA expression in dMacs, we sorted Lin$^-$CD11b$^+$CD64$^+$Ly6C$^-$MHC-II$^+$ dMacs, Lin$^-$CD11b$^+$CD64$^-$Ly6C$^+$MHC-II$^{lo}$ dermal monocytes (hereafter dMonos) and Lin$^-$CD11b$^+$CD64$^-$Ly6C$^-$MHC-II$^+$ dermal DCs (hereafter dDCs) from the enzymatically digested saline-perfused back skin of a WT mouse, as enough cells could not be isolated from the hind paw skin (Extended Data Fig. 8b). RT–qPCR indicated that *Snx25* mRNA expression was lower in dDCs than in dMacs and dMonos but not significantly different between dMacs and dMonos of WT mice (Fig. 5l). The percentages of dMacs, dMonos and dDCs were comparable between $Snx25^{+/-}$ and WT mice (Extended Data Fig. 8c). Expression of *Ngf* mRNA was lower in $Snx25^{+/-}$ or $Snx25^{Cx3cr1-cKO}$ dMacs than in WT dMacs (Fig. 5m,n) but not in $Snx25^{+/-}$ dMonos or dDCs compared to WT counterparts (Extended Data Fig. 8d–g). Next, we made BM chimera by transplanting whole BM from $Snx25^{Cx3cr1-cKO}$;Ai32$^{Tg/+}$ mice into WT mice and treated them with TAM from week 1 until week 8 after BMT (Extended Data Fig. 9a). Expression of *Ngf* mRNA was decreased in donor-derived MHC-II$^+$YFP$^+$ dMacs compared to MHC-II$^+$YFP$^-$ dMacs sorted from the skin of the recipient mice at week 8 after BMT (Fig. 5o and Extended Data Fig. 9b). $Snx25^{+/-}$ mice regained normal VF thresholds at 24 h after intradermal injection of NGF, while mice injected with PBS did not (Fig. 5p). These results suggested that SNX25 in dMacs regulates NGF expression and thereby modulates pain sensitivity.

## SNX25 regulated *Ngf* mRNA expression through Nrf2

We next investigated the molecular mechanisms connecting SNX25 to *Ngf* mRNA synthesis. Consistent with reports that Nrf2 regulates *Ngf* mRNA induction in glial cells[13], *Nrf2*-specific siRNA-mediated knockdown in BMDMs significantly reduced constitutive expression of *Ngf* mRNA compared to the scramble siRNA control (Fig. 6a). Cellular expression of Nrf2 is regulated by continuous ubiquitination and proteasome degradation, which is blocked by the Kelch-like ECH-associated protein 1 (Keap1) protein[32]. The amount of poly-ubiquitinated Nrf2 protein was increased in 293T cells treated with the proteasome inhibitor MG132 (Fig. 6b) and was further elevated by siRNA-mediated knockdown of *Snx25* (Fig. 6c). In turn, transient transfection with vectors for overexpression of SNX25 and Nrf2 decreased poly-ubiquitinated Nrf2 in 293T cells compared to cells overexpressing *Nrf2* only (Fig. 6d). In vitro treatment with 4-OHT increased the amount of poly-ubiquitinated Nrf2 in $Snx25^{Cx3cr1-cKO}$ BMDMs compared to the vehicle-treated control (Fig. 6e). SNX25 was co-immunoprecipitated with Nrf2 in 293T cells overexpressing mouse *Snx25* and *Nrf2* (Fig. 6f). An in vitro ubiquitination assay indicated an increase in ubiquitinated Nrf2 protein in Snx25-knockdown BMDMs compared to scramble siRNA-expressing BMDMs (Fig. 6g). Consistent with this, heme oxygenase 1 (HO-1), a target of Nrf2, was decreased in the hind paw skin of $Snx25^{+/-}$ mice compared to that of WT mice (Fig. 6h), and knockdown of Keap1, which accelerates Nrf2 degradation[32], with Keap1 siRNA rescued *Ngf* expression in $Snx25^{+/-}$ BMDMs (Fig. 6i). These results suggested that SNX25 modulates Nrf2 ubiquitination and thereby *Ngf* mRNA expression.

## SNX25 in dMacs is a key factor in pain sensation

To address whether dMacs were sufficient to initiate pain sensation without neuropathic intervention or inflammation, we depleted dMacs by intradermal injection of clodronate liposome[33] twice into one side of the hind paw. Immunohistochemistry showed that the numbers of CD206$^+$ or MHC-II$^+$ macrophages significantly decreased at day 3 after the second clodronate liposome injection compared to skin injected with control liposome (Fig. 7a). VF thresholds were increased in mice injected with clodronate but not in those injected with control liposome at day 3 after the second injection (Fig. 7b). Immunoblot analyses showed that NGF, SNX25 and CD206 expression was lower in skin injected with clodronate than in skin injected with control liposome (Fig. 7c,d). These findings indicated that dMacs were required for pain sensation under steady state.

**Fig. 5 | NGF expression in macrophages was reduced in $Snx25^{+/-}$ mice. a**, Representative immunoblot showing NGF expression in the hind paw skin of WT and $Snx25^{+/-}$ mice. The graph shows semi-quantitative analyses of the immunoblots (WT, $n = 4$; $Snx25^{+/-}$, $n = 4$). $P = 0.002$. **b**, Representative immunoblot showing NGF levels in the hind paw skin of WT and $Snx25^{+/-}$ mice 30 min after formalin injection and semi-quantitative analyses of NGF levels in the ipsilateral (ipsi) hind paw skin of WT ($n = 3$) and $Snx25^{+/-}$ mice ($n = 3$). $P = 0.015$. Cont, contralateral. **c**, Confocal microscopy of the hind paw skin immunolabeled for NGF and MHC-II, F4/80 and Iba1 (macrophage markers) and CD117 (mast cell marker) in WT mice. Arrowheads denoted double-labeled cells. Representative of three independent experiments. Scale bar, 50 μm. **d**, Immunoblot of NGF in BMDMs of WT and $Snx25^{+/-}$ mice, normalized to glyceraldehyde-3-phosphate dehydrogenase (GAPDH) content and analyzed semi-quantitatively (WT, $n = 4$; $Snx25^{+/-}$, $n = 5$, $P = 0.015$). **e**, Confocal microscopic images of sciatic nerve sections immunolabeled for TrkA at 8 h after nerve ligation (arrows indicate ligation site) in WT and $Snx25^{+/-}$ mice (left), magnified views of the boxed areas in the corresponding left panels (middle) and semi-quantitative analysis of TrkA accumulation on the distal side of the nerve ligature (right) ($n = 4$ sciatic nerve sections from three different mice, $P = 0.008$). Representative of two independent experiments. Scale bar, 200 μm. **f**, Expression profiles of *Snx25* and *Ngf* mRNA in BMDMs of WT and $Snx25^{+/-}$ mice (WT, $n = 6$; $Snx25^{+/-}$, $n = 6$; *Snx25*, $P = 0.049$; *Ngf*, $P = 0.085$). **g**, *Ngf* mRNA quantified by RT–qPCR in BMDMs transfected with either *Snx25* siRNA (si*Snx25*, $n = 3$) or scramble siRNA (siCtr, $n = 3$; $P = 0.0085$). **h**, Confocal microscopy of YFP-labeled BMDMs derived from $Snx25^{Cx3cr1-cKO}$;Ai32$^{Tg/+}$ mice without (top) or with (bottom, 1 μM, 7–8 d) 4-OHT treatment. Representative of three independent experiments. Scale bar, 100 μm. **i**, Flow cytometry of PI and YFP expression in BMDMs cultured from $Snx25^{Cx3cr1-cKO}$; Ai32$^{Tg/+}$ mice. **j**, Expression of *Snx25* and *Ngf* in BMDMs cultured and sorted from $Snx25^{Cx3cr1-cKO}$;Ai32$^{Tg/+}$ mice (YFP$^-$, $n = 3$; YFP$^+$, $n = 3$; *Snx25*, $P = 0.027$; *Ngf*, $P = 0.009$). **k**, Confocal microscopy of the hind paw skin immunolabeled for YFP and MHC-II in WT mice that received BMT from $Snx25^{Cx3cr1-cKO}$;Ai32$^{Tg/+}$ mice treated with TAM for 2 weeks. Boxed areas (i–iii) in the upper panel are magnified in the lower panels. Representative of three independent experiments. Scale bar, 100 μm. **l**, Expression patterns of *Snx25* and *Ngf* in dMacs ($n = 3$), dMonos ($n = 3$) and dDCs ($n = 3$). **m**, Expression of *Snx25* and *Ngf* in dMacs of WT and $Snx25^{+/-}$ mice (WT, $n = 5$; $Snx25^{+/-}$, $n = 5$; *Snx25*, $P = 0.002$; *Ngf*, $P = 0.014$). **n**, Expression of *Snx25* and *Ngf* in dMacs of $Snx25^{fl/fl}$ and $Snx25^{Cx3cr1-cKO}$ mice ($Snx25^{fl/fl}$, $n = 3$; $Snx25^{Cx3cr1-cKO}$, $n = 3$; *Snx25*, $P = 0.007$; *Ngf*, $P = 0.057$). **o**, Expression of *Yfp*, *Snx25* and *Ngf* mRNA in dMacs of WT mice that received BMT from $Snx25^{Cx3cr1-cKO}$;Ai32$^{Tg/+}$ mice (YFP$^-$, $n = 5$; YFP$^+$, $n = 5$; *Yfp*, $P = 0.043$; *Snx25*, $P = 0.008$; *Ngf*, $P = 0.009$). **p**, VF thresholds before and 24 h after injection in WT or $Snx25^{+/-}$ mice injected with NGF (10 ng μl$^{-1}$, 10 μl) or PBS (NGF, WT, $n = 5$, $P = 0.017$; $Snx25^{+/-}$, $n = 7$, $P = 0.014$. PBS, WT, $n = 4$; $Snx25^{+/-}$, $n = 5$). g, gram. Results are represented as mean ± s.e.m. Statistical analyses were performed using two-tailed Student's *t*-test (**d,e**), two-tailed Welch's *t*-test (**a,b,f,g,j,m–o**) or one-way ANOVA (**l,p**), and significant differences between group means were identified with the Tukey–Kramer test. *$P < 0.05$, **$P < 0.01$.

Next, we intradermally injected 4-OHT into the right hind paws and vehicle into the left hind paws of $Snx25^{Cx3cr1\text{-}cKO}$;Ai32$^{Tg/+}$ mice daily for 7 d to test the effect of local $Snx25$ conditional KO (Extended Data

Fig. 10a). At day 8 after the last injection, the hind paws injected with 4-OHT but not those injected with vehicle showed elevated VF thresholds (Fig. 7e). In hind paws injected with 4-OHT, 96% of YFP$^+$ cells were

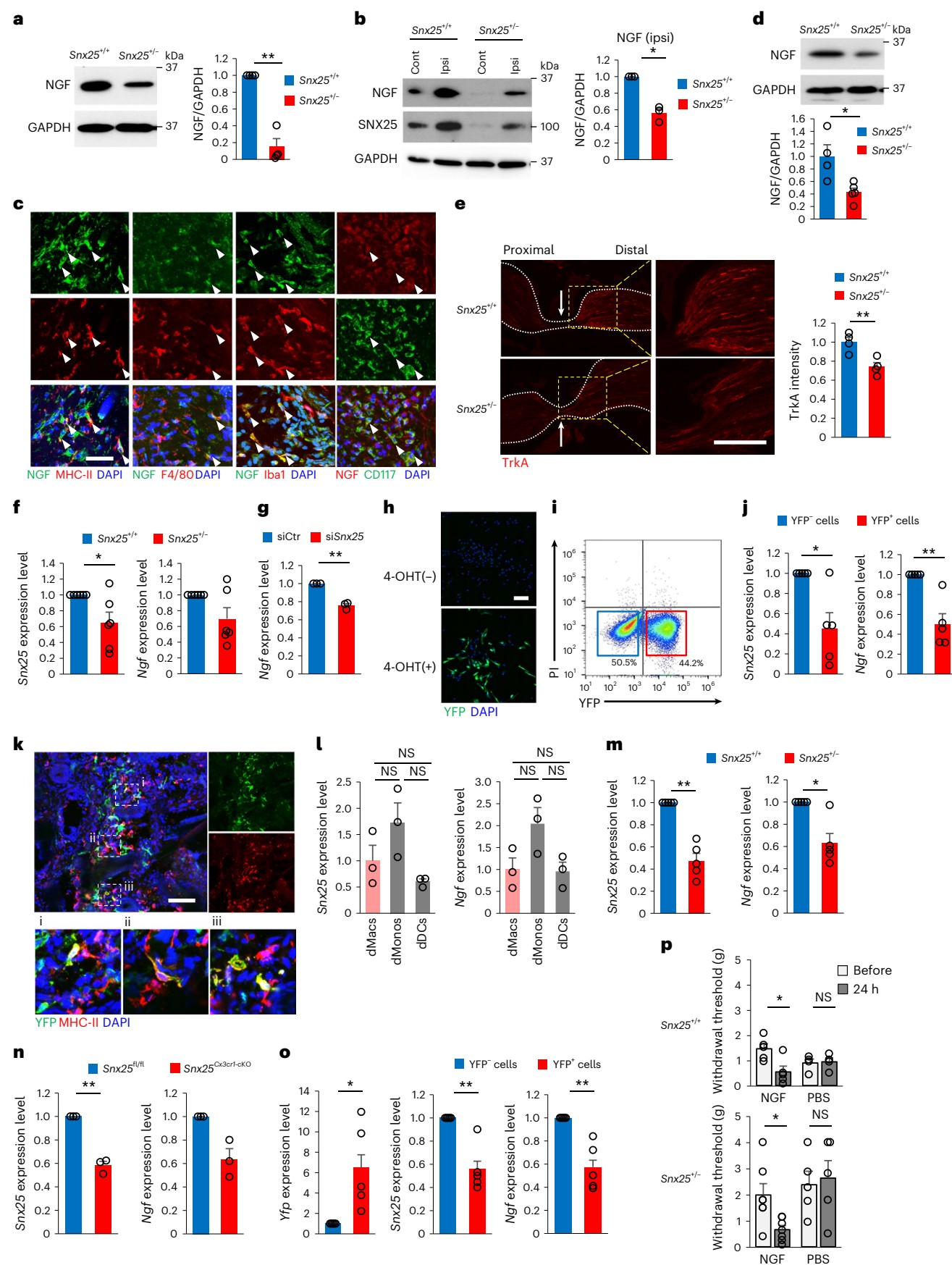

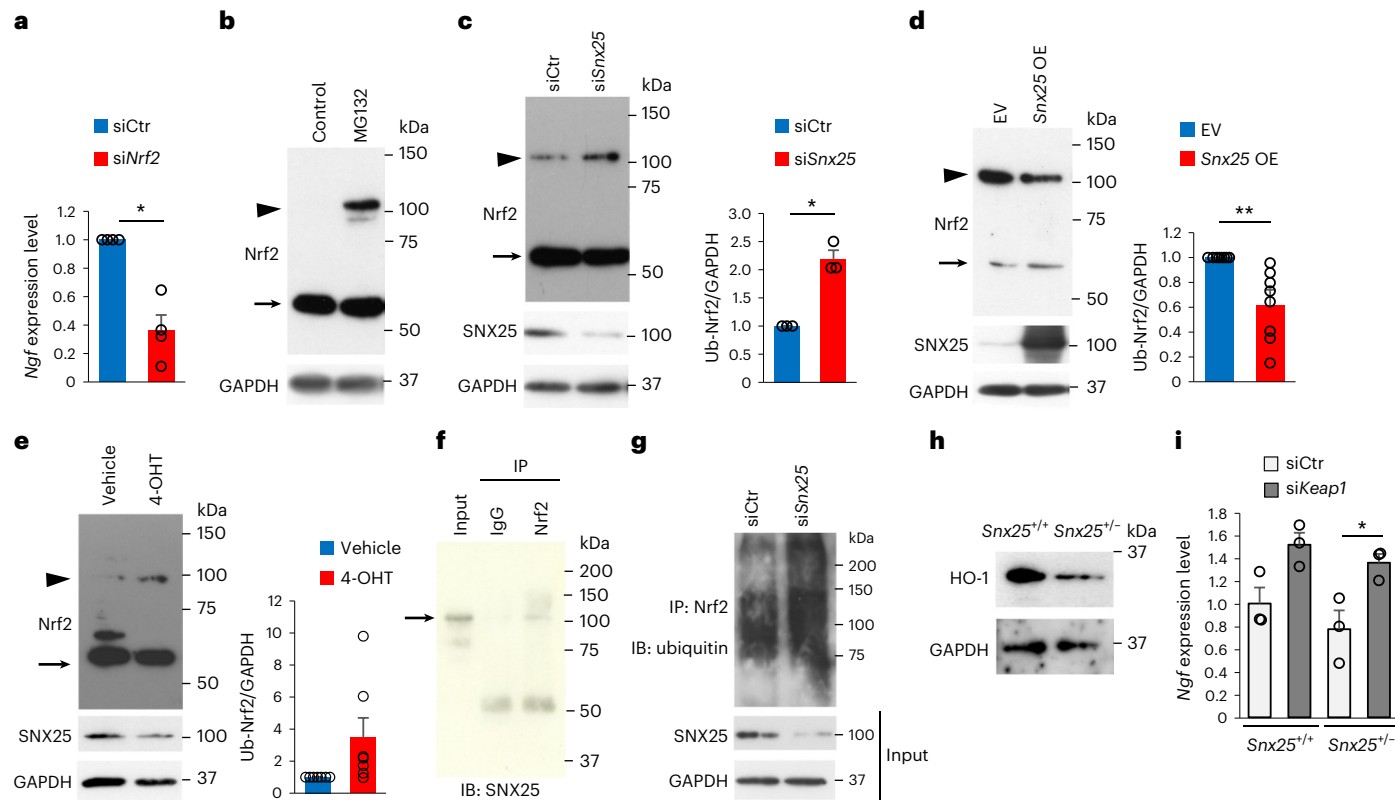

**Fig. 6 | SNX25 activated *Ngf* production by inhibiting ubiquitin-mediated degradation of Nrf2. a**, *Ngf* mRNA expression in BMDMs transfected with either Nrf2 siRNA or scramble siRNA analyzed semi-quantitatively by RT–qPCR (si*Nrf2*, $n = 4$; siCtr, $n = 4$, $P = 0.01$). **b**, Representative immunoblot showing Nrf2 protein levels in 293T cells in the presence or absence of MG132. Arrow, Nrf2 (61–68 kDa); arrowhead, poly-ubiquitinated Nrf2 (100–110 kDa). **c**, Ubiquitination levels of Nrf2 protein in 293T cells transfected with Snx25 siRNA or scramble siRNA in the presence of MG132. Arrow, Nrf2; arrowhead, poly-ubiquitinated Nrf2. The band intensity of poly-ubiquitinated Nrf2 (Ub-Nrf2) was analyzed semi-quantitatively (siCtr, $n = 3$; si*Snx25*, $n = 3$, $P = 0.017$). **d**, Representative immunoblot showing Nrf2 and poly-ubiquitinated Nrf2 levels in 293T cells transfected with the full-length *Snx25* expression vector (*Snx25* OE) or empty vector (EV) in addition to the *Nrf2* expression vector in the presence of MG132. Arrow, Nrf2; arrowhead, poly-ubiquitinated Nrf2. Semi-quantitative analysis of poly-ubiquitinated Nrf2 bands is shown (empty vector and *Nrf2* vector, $n = 8$; *Snx25* vector and *Nrf2* vector, $n = 8$, $P = 0.006$). **e**, Immunoblot of Nrf2 in BMDMs of *Snx25*[Cx3cr1-cKO] mice treated with

4-OHT or vehicle in the presence of MG132 (4-OHT, $n = 7$; vehicle, $n = 7$, $P = 0.093$). Arrow, Nrf2; arrowhead, poly-ubiquitinated Nrf2. **f**, Co-immunoprecipitation (IP) of SNX25 and Nrf2 in 293T cells expressing *Snx25* and *Nrf2*. Cell lysates were immunoprecipitated with anti-Nrf2 antibody and immunoblotted with anti-SNX25 antibody. Normal IgG (IgG) was used as a negative control. Arrow, SNX25; IB, immunoblot. **g**, Detection of ubiquitin-bound Nrf2 in SNX25-knockdown or scramble siRNA-treated BMDMs treated with MG132 followed by immunoprecipitation of cell lysates with anti-Nrf2 antibody and immunoblotting with anti-ubiquitin antibody. **h**, Representative immunoblot of HO-1 in the hind paw skin of WT and *Snx25*[+/−] mice. **i**, *Ngf* mRNA quantification by RT–qPCR in BMDMs transfected with either Keap1 siRNA or scramble siRNA (siCtr, $n = 3$; si*Keap1*, $n = 3$). *Snx25*[+/−], $P = 0.046$. Results are represented as mean ± s.e.m. Statistical analyses were performed using two-tailed Welch's *t*-test (**a**,**c**–**e**) or one-way ANOVA (**i**), and significant differences between group means were identified with the Tukey–Kramer test. *$P < 0.05$, **$P < 0.01$.

SNX25⁻ (Extended Data Fig. 10b,c), indicating successful local conditional KO of the *Snx25* gene in dMacs. Ai32[Tg]-derived YFP expression was detected in MHC-II⁺ dMacs from hind paws injected with 4-OHT but not in hind paws injected with vehicle (Fig. 7f) or in MHC-II⁺ macrophages in the sciatic nerve or the DRG (Fig. 7f).

Macrophages in the DRG are reported to mediate neuropathic pain[34]. In both WT and *Snx25*[+/−] mice, double-labeling immunohistochemistry revealed that MHC-II⁺ DRG macrophages were associated with PGP9.5⁺ nerves (Fig. 8a) and that CD206⁺ or F4/80⁺ DRG macrophages had low to moderate expression of SNX25 (Fig. 8b). To test whether DRG macrophages contributed to pain sensitivity in BM chimeras, we intravenously transplanted whole BM cells from GFP mice into WT mice pretreated with busulfan. At 5 weeks after the transfer, approximately 60% of MHC-II⁺ DRG macrophages were GFP⁺ (Fig. 8c–e), whereas very few GFP⁺MHC-II⁺ macrophages were detected in the sciatic nerve (Fig. 8f,g), suggesting that homeostatic turnover of macrophages occurred in DRGs. To test the contribution of DRG macrophages to pain sensitivity, we directly injected 4-OHT into surgically exposed DRGs (L4 and L5) in *Snx25*[Cx3cr1-cKO];Ai32[Tg/+] mice and

*Snx25*[fl/fl];Ai32[Tg/+] mice (Extended Data Fig. 10d). At day 5 after 4-OHT injection, immunohistochemistry detected Ai32[Tg/+]-derived YFP⁺ macrophages in the DRGs of *Snx25*[Cx3cr1-cKO];Ai32[Tg/+] mice but not in *Snx25*[fl/fl];Ai32[Tg/+] mice (Fig. 8h). In the DRGs of the *Snx25*[Cx3cr1-cKO];Ai32[Tg/+] mice, 91% of YFP⁺ cells were SNX25⁻, indicating successful recombination by local 4-OHT administration (Extended Data Fig. 10e). Quantification of SNX25 immunofluorescence intensities further showed that CD206⁺ DRG macrophages in *Snx25*[Cx3cr1-cKO];Ai32[Tg/+] mice expressed only 14% of SNX25 immunoreactivities compared to *Snx25*[fl/fl];Ai32[Tg/+] mice at day 5 after 4-OHT injection (Fig. 8i), confirming that 4-OHT treatment eliminated SNX25 expression in DRG macrophages. At day 5 after 4-OHT injection into DRGs, ipsilateral hind paws of *Snx25*[Cx3cr1-cKO];Ai32[Tg/+] mice exhibited VF thresholds comparable to those of the contralateral hind paws of the same mice, to those of the ipsilateral hind paws before injection and to those of *Snx25*[fl/fl];Ai32[Tg/+] mice (Fig. 8j), indicating that SNX25 in DRG macrophages was not involved in pain sensation under normal conditions. NGF expression was low in F4/80⁺ DRG macrophages at steady state (Fig. 8k,l), suggesting that DRG macrophages might have different regulatory mechanisms of NGF expression and

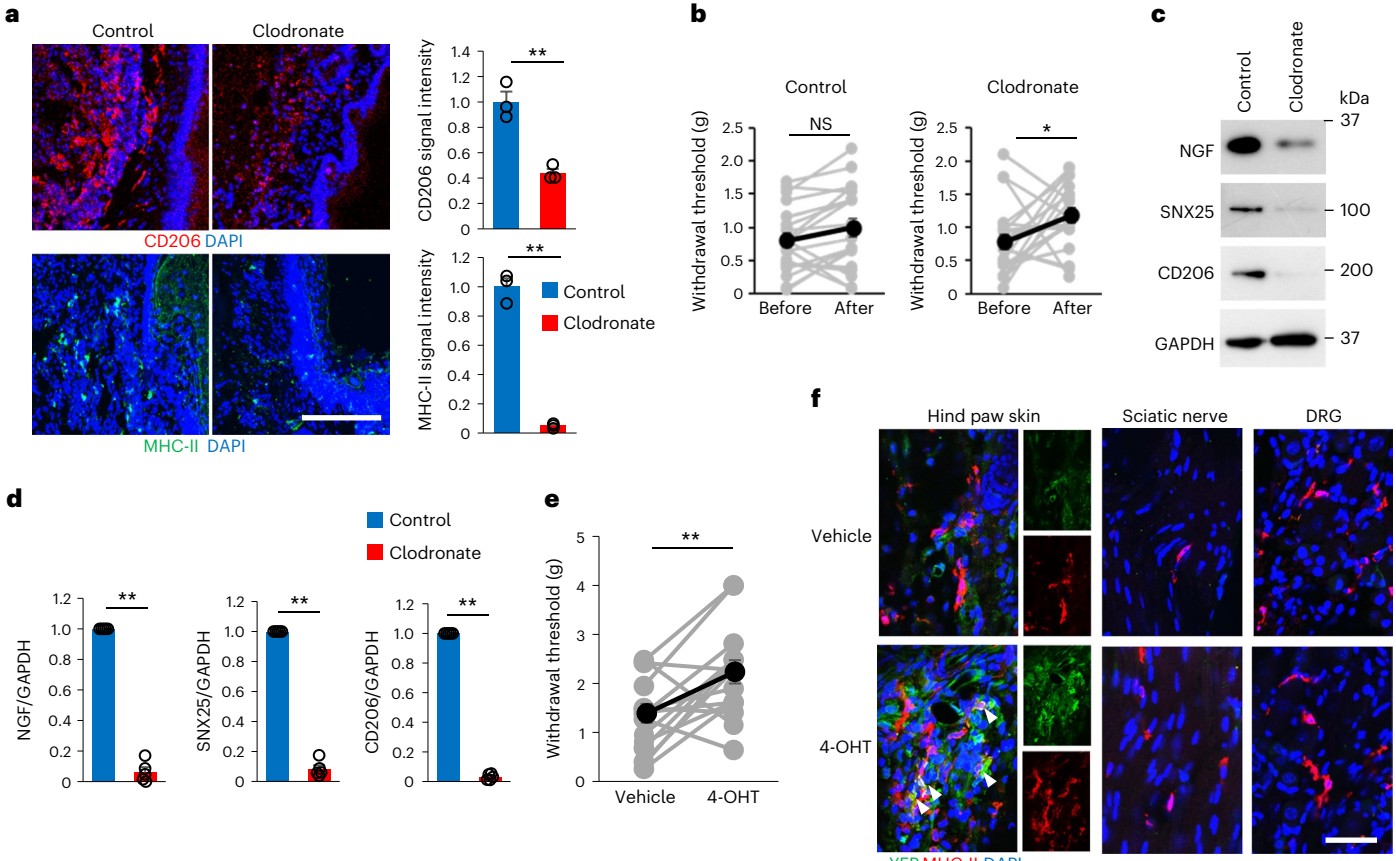

**Fig. 7 | dMacs were sufficient to initiate pain sensation. a**, Confocal microscopy of the hind paw skin immunolabeled for CD206 or MHC-II in WT mice injected with control liposome or clodronate liposome. Right, quantification of mean fluorescence intensities for CD206 and MHC-II (*n* = 3 hind paw skin sections from three different mice, CD206, *P* = 0.003; MHC-II, *P* = 8.42 × 10⁻⁵). Scale bar, 200 μm. **b**, VF thresholds on the side injected with control liposome (*n* = 20, *P* = 0.296) or the side injected with clodronate liposome (*n* = 20, *P* = 0.023) of WT mice. g, gram. **c,d**, Expression of NGF, SNX25 and CD206 in the hind paw skin of sides injected with control liposome or clodronate liposome in WT mice examined by immunoblotting (**c**) and semi-quantitatively compared for

NGF, SNX25 and CD206 (**d**). Control, *n* = 5; clodronate, *n* = 5. NGF, *P* = 6.69 × 10⁻⁶; SNX25, *P* = 2.87 × 10⁻⁶; CD206, *P* = 4.22 × 10⁻⁸. **e**, VF thresholds of sides injected with vehicle or 4-OHT (left and right side, respectively) of the hind paws of *Snx25*^Cx3cr1-cKO^;Ai32^Tg/+^ mice (*n* = 15, *P* = 0.009). g, gram. **f**, Confocal microscopy of the hind paw skin, the sciatic nerve and the DRG immunolabeled for MHC-II and YFP of the *Snx25*^Cx3cr1-cKO^;Ai32^Tg/+^ mice shown in **e**. Sections of the side injected with vehicle (top) and the side injected with 4-OHT (bottom). Scale bar, 50 μm. Results are represented as mean ± s.e.m. Statistical analyses were performed using two-tailed Student's *t*-test (**a,b,e**) or two-tailed Welch's *t*-test (**d**). *P < 0.05, **P < 0.01. Representative of three independent experiments (**a,f**).

pain conduction than dMacs. These data indicate that SNX25 in dMacs but not DRG macrophages modulate acute pain sensing under steady state (Extended Data Fig. 10f).

## Discussion

Here we showed that *Snx25*^+/−^ mice and *Snx25*^Cx3cr1-cKO^ mice had reduced pain responses under both normal and pain-inducing conditions. SNX25 inhibited ubiquitination and proteasome degradation of Nrf2, which regulated the transcription of *Ngf* mRNA and, as such, maintained the production of NGF in dMacs. Loss of SNX25 accelerated Nrf2 degradation and lowered NGF expression, which led to insensitivity to pain.

Progress in gene-cataloging techniques has broadened our knowledge of tissue macrophages. In the lung, Lyve1^lo^MHC-II^hi^CX-3CR1^hi^ interstitial macrophages are associated with nerves, whereas Lyve1^hi^MHC-II^lo^CX3CR1^lo^ counterparts preferentially localize around blood vessels[4]. These interstitial macrophages were in part derived from BM[4], consistent with fate-mapping studies[5,20]. In the hind paw skin of WT mice that received BMT from *Snx25*^Cx3cr1-cKO^;Ai39^Tg/+^ mice, Ai39^Tg^-derived YFP⁺ dMacs were frequently found in close proximity to PGP9.5⁺ nerves in the dermis, consistent with a report that CX3CR1^hi^ macrophages colocalize with peripheral nerves and contribute to the surveillance and regeneration of local nerves[5]. Production of NGF

by dMacs likely contributes to the regeneration of local nerves, in addition to the maintenance of pain sensibility. We speculate that the mechanosensing ability of dMacs is linked to the production of NGF and thereby to the regulation of mechanical pain sensitivity. Supporting this hypothesis, clodronate liposome-mediated deletion of dMacs led to decreased NGF expression and concomitant insensitivity to pain. Macrophages in the DRG were also reported to mediate neuropathic pain[34]. Given the pain-sensitive phenotype in *Snx25* conditional KO in DRG macrophages, these cells likely contribute to pain sensation through different mechanisms.

We showed that SNX25 regulates cellular amount of the Nrf2 protein by modulating its ubiquitination. Nrf2 regulates mechanical stretch-induced gene expression in cardiomyocytes[35]. The SNX25-Nrf2 pathway in dMacs may optimize the concentration of NGF that modulates neuronal responses to mechanical stimuli. The interleukin (IL)-23-IL-17A-TRPV1 axis was recently reported to regulate female-specific mechanical pain perception through macrophage-neuron interactions[36]. Mechanisms described in our study partially overlap with those of the study[36], but we did not investigate whether the SNX25-Nrf2 axis has sex specificity. In our experiments, VF thresholds in the paws tended to be higher in female *Snx25*^+/−^ mice, but the difference between sexes was not statistically significant. Therefore, we used mostly male mice in the present study.

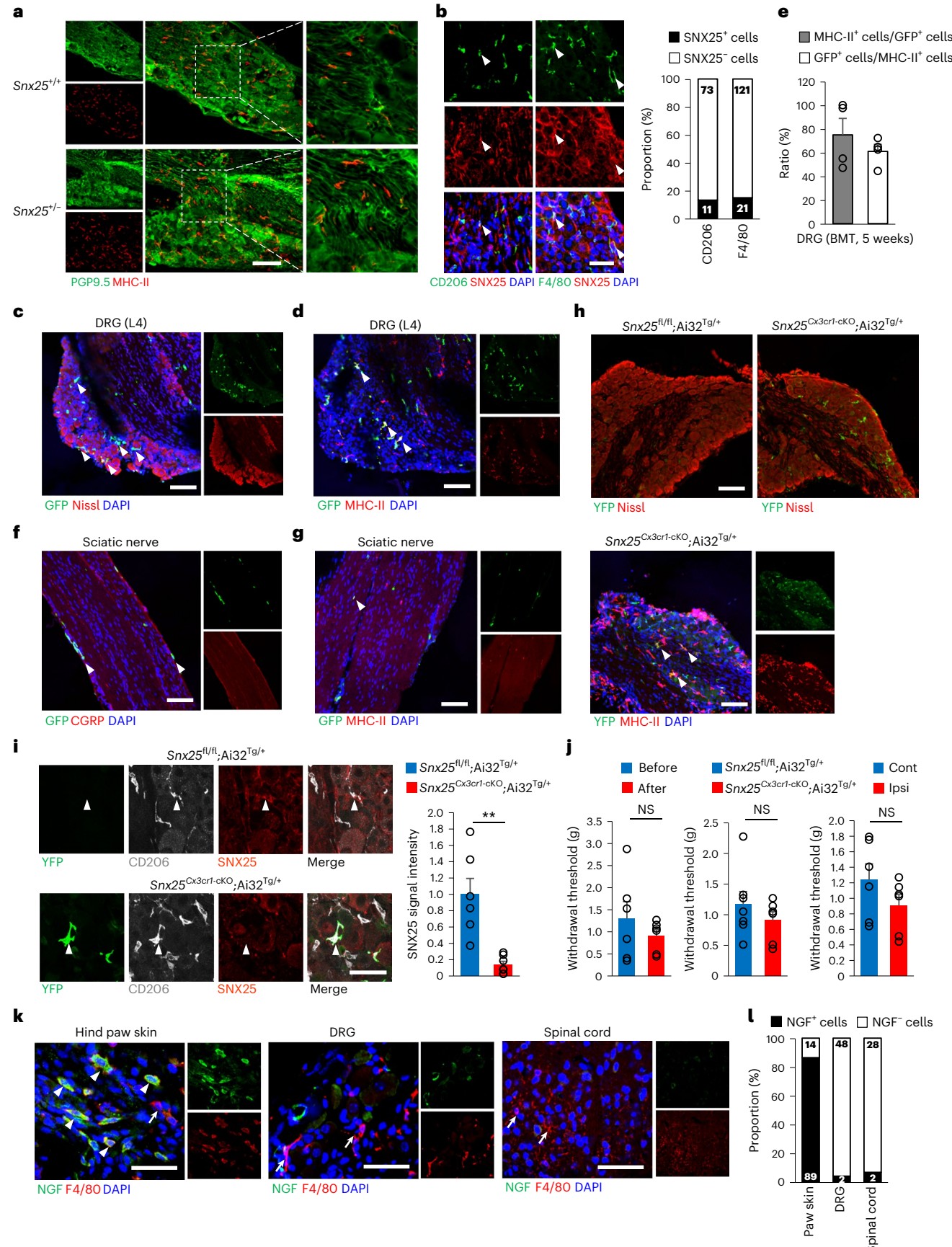

Lowering the level of NGF in naive skin for a relatively short period decreased pain sensitivity in $Snx25^{Cx3cr1\text{-cKO}}$ mice and in mice that received BMT from $Snx25^{Cx3cr1\text{-cKO}}$ mice. In humans, HSAN V is characterized by a marked loss of pain sensation and is caused by mutations in the *NGF* gene[10,11]. The suppression of NGF expression in $Snx25^{Cx3cr1\text{-cKO}}$ mice mimicked HSAN V pathology to some extent, but we did not observe

**Fig. 8 | SNX25⁺ macrophages in the DRG did not contribute to pain sensitivity.**
**a**, Confocal microscopy of the DRG (L4, naive) immunolabeled for PGP9.5 (green) and MHC-II (red) in WT and $Snx25^{+/-}$ mice. Right, magnified views of boxed areas in the middle panels. Scale bar, 100 μm. **b**, Confocal microscopic images of the DRG (L4, naive) immunolabeled for macrophage markers (CD206 or F4/80, red) and SNX25 (red) in WT mice. Scale bar, 50 μm. The graph shows the proportion of SNX25⁺ cells in each specific marker-positive cell population. The numbers in the columns indicate actual cell numbers counted. $n$ = 11 sections from three different mice. **c**, Confocal microscopy of a DRG (L4) section stained for GFP (green) and fluorescent Nissl (red) in WT mice that received BMT from GFP mice. Arrowheads denote BM-derived GFP⁺ cells. Scale bar, 100 μm. **d**, Confocal microscopy of the DRG (L4) double labeled for GFP and MHC-II in the mice shown in **c**. Arrowheads denote double-labeled cells. Scale bar, 100 μm. **e**, Percentages of MHC-II⁺ cells among GFP⁺ cells and of GFP⁺ cells among MHC-II⁺ cells (5 weeks after transplantation) ($n$ = 5 DRG sections from three different mice). **f**, Confocal microscopy of the sciatic nerve double labeled for GFP and CGRP in the mice shown in **c**. Arrowheads denote BM-derived GFP⁺ cells. Scale bar, 100 μm. **g**, Confocal microscopy of the sciatic nerve double labeled for GFP and MHC-II in the mice shown in **c**. Arrowhead denote a GFP⁺MHC-II⁺ cell. Scale bar, 100 μm. **h**, Confocal microscopy of the DRG (L4) labeled for fluorescent Nissl (red) and immunolabeled for YFP (green) after injection of 4-OHT directly into DRGs in

$Snx25^{Cx3cr1-cKO}$;Ai32$^{Tg/+}$ mice (left) and $Snx25^{fl/fl}$;Ai32$^{Tg/+}$ mice (right). Scale bar, 100 μm. Bottom, confocal microscopy of the DRG (L4) immunolabeled for MHC-II (red) and YFP (green) after injection of 4-OHT directly into DRGs. Arrowheads indicate cells double labeled for MHC-II and YFP and counterstained with DAPI (blue). Scale bar, 100 μm. **i**, Confocal microscopy of the DRG immunolabeled for YFP, CD206 and SNX25 in $Snx25^{fl/fl}$;Ai32$^{Tg/+}$ mice and $Snx25^{Cx3cr1-cKO}$;Ai32$^{Tg/+}$ mice. Arrowheads denote CD206⁺SNX25⁺ macrophages. Scale bar, 50 μm. The graph shows the fluorescence intensities of SNX25 in CD206⁺ macrophages in $Snx25^{Cx3cr1-cKO}$;Ai32$^{Tg/+}$ mice ($n$ = 7 sections from three different mice) and $Snx25^{fl/fl}$;Ai32$^{Tg/+}$ mice ($n$ = 6 sections from three different mice). $P$ = 0.001. **j**, VF thresholds before ($n$ = 6) and after ($n$ = 6) 4-OHT injection in $Snx25^{Cx3cr1-cKO}$;Ai32$^{Tg/+}$ mice ($P$ = 0.382), $Snx25^{fl/fl}$;Ai32$^{Tg/+}$ ($n$ = 7) and $Snx25^{Cx3cr1-cKO}$;Ai32$^{Tg/+}$ ($n$ = 6) mice ($P$ = 0.348) and in contralateral ($n$ = 6) and ipsilateral ($n$ = 6) sides of $Snx25^{Cx3cr1-cKO}$;Ai32$^{Tg/+}$ mice ($P$ = 0.218). g, gram. **k**, Confocal microscopy of the hind paw skin, the DRG and the spinal cord immunolabeled for NGF (green) and F4/80 (red) in WT mice. Arrowheads denote double-positive cells and arrows denote NGF⁻F4/80⁺ cells. Scale bar, 50 μm. **l**, Proportion of NGF⁺ cells (black column) in F4/80⁺ cells. Numbers inside columns are the actual numbers of cells counted. $n$ = 6 sections from three different mice. Results are represented as mean ± s.e.m. Statistical analyses were performed using the two-tailed Student's $t$-test. Representative of three independent experiments (**a**–**d**,**f**–**i**,**k**).

long-term NGF deficiency or morphological changes such as retraction of nerve endings[37]. Based on clinical manifestations of patients with HSAN V, NGF-neutralizing monoclonal antibodies were developed as a therapeutic means to mitigate refractory pain[38]. Although unexpected side effects such as arthralgia and osteonecrosis prevented these monoclonal antibodies from proceeding to clinics, NGF is still a good therapeutic target of pain-relieving medicine[39]. The SNX25–Nrf2 axis in dMacs has the potential to bridge the painless phenotype of HSAN V to hyperalgesic conditions and could represent a promising alternative to anti-NGF monoclonal antibodies. However, because NGF is also produced by noninflammatory cells, such as keratinocytes[40], further experiments are needed to determine the entire cellular and molecular mechanism controlling peripheral NGF levels.

## Online content

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

## Methods

### Mice

*Mlc1*[Tg] mice (B6; CBB6(129)-Tg(Mlc1-tTA)2Rhn) were a gift from K.F. Tanaka (Keio University) and were originally employed for the purpose of manipulating gene expression in astrocytes[14]. *Snx25*-constitutive KO (*Snx25*[+/−]) mice (C57BL/6N-A[tm1Brd] *Snx25*[tm1a(KOMP)Wtsi]/NjuMmucd) were obtained from the Nanjing Biomedical Research Institute of Nanjing University. *Snx25*-conditional KO mice were generated by first crossing our *Snx25*[LacZ/+] mice with CAG-Flpo mice (B6.Cg-Tg(CAG-FLPo)/1Osb), which were a gift from M. Ikawa (Osaka University), to excise the *lacZ* cassette framed by *Frt* sites and to obtain an allele with floxed exon 4 (*Snx25*[fl/fl] mice)[41]. We crossed *Avil*[Cre] mice (B6.Cg-Tg(*Avil*[CreERT2])AJwo/J) (Jackson Laboratory, 032027) with *Snx25*[fl/fl] mice to obtain *Snx25*[Avil-cKO] mice. We crossed *Cx3cr1*[CreERT2] mice (B6.129P2(C)-*Cx3cr1*[tm2.1(CreERT2)Jung]/J) (Jackson Laboratory, 020940) with *Snx25*[fl/fl] mice to obtain *Snx25*[Cx3cr1-cKO] mice. We crossed *Snx25*[Cx3cr1-cKO] mice with a reporter mouse (Ai39 mice, RCL-eNpHR3.0-EYFP, Jackson Laboratory, 014539) to obtain *Snx25*[Cx3cr1-cKO];Ai39[Tg/+] mice. We crossed *Snx25*[Cx3cr1-cKO] mice with a reporter mouse (Ai32 mice, RCL-ChR2(H134R)/EYFP, Jackson Laboratory, 012569) to obtain *Snx25*[Cx3cr1-cKO];Ai32[Tg/+] mice. GFP mice that express GFP ubiquitously (C57BL/6-Tg(CAG-EGFP)C14-Y01-FM131Osb) were purchased from Japan SLC. All mice were housed in standard cages under a 12-h light–dark cycle and temperature-controlled conditions. All protocols for animal experiments were approved by the Animal Care Committee of Nara Medical University in accordance with the policies established in the NIH Guide for the Care and Use of Laboratory Animals. This study was also carried out in compliance with the ARRIVE guidelines (https://arriveguidelines.org/).

### Behavioral test

Paw mechanical sensitivity was assessed using von Frey's filaments (Muromachi Kikai) based on the up–down method developed by Chaplan[42]. The von Frey's filaments used were 0.07, 0.16, 0.4, 0.6, 1, 1.4, 2 and 4 g. Animals were acclimatized for at least 15 min in individual clear acrylic cubicles (10 × 10 × 10 cm) placed on top of an elevated wire mesh. Quick withdrawal or licking of the paw after the 3-s stimulus was considered a positive response. Threshold values were derived according to the method described by Chaplan[42]. For the von Frey test after NGF (N-240, Alomone Labs) injection, 100 ng NGF (10 μl) was injected subcutaneously into the right hind paw and PBS (10 μl) was injected into the left hind paw of the same mouse. For the formalin test, 10 μl of 5% formalin was injected subcutaneously into the right hind paw and PBS (10 μl) was injected into the left hind paw. We calculated the durations of lifting, shaking and licking of the formalin-injected paw. For the hot plate test, mice were acclimatized for at least 2 h (1 h per day for 2 d) in individual clear acrylic cubicles placed on the pre-heated plate. The withdrawal latency in response to the stimulus was determined manually. In all behavioral tests, examiners were always blind to the genotypes of mice, the kinds of treatments and the sides of hind paws that received injections. After the evaluation was finished, the behavioral data were analyzed by a different researcher. We experienced discrepancy in mechanosensation between male and female *Snx25*[+/−] mice. For example, the von Frey test in female WT and *Snx25*[+/−] mice revealed that the withdrawal threshold tended to be higher than in male counterparts, but there was no significant difference. In the present study, we limited the analysis to male mice.

### Surgery for the spared nerve injury model

Surgical procedures were performed under 2% isoflurane anesthesia. SNI was made with a 6-0 polypropylene thread (EH7835, Ethicon) with tight ligation of the two branches of the right sciatic nerve, the common peroneal and the tibial nerves, followed by transection and removal of a 2-mm nerve portion. The sural nerve remained intact, and any contact with or stretching of this nerve was carefully avoided. Muscle and skin were closed in two distinct layers.

### Reagents

For TAM treatment, we employed oral administration. TAM (T5648, Sigma-Aldrich) was mixed with powdered chow (0.5 mg per g normal chow). This oral administration method is convenient for continuous administration and results in efficient induction of recombination while minimizing stress on mice[17]. For knockdown experiments in BMDMs or the macrophage cell line RAW264.7 (ECACC 91062702), we treated cells with *Snx25*-specific siRNA (Sigma-Aldrich), *Nrf2*-specific siRNA (Sigma-Aldrich) and *Keap1*-specific siRNA (Sigma-Aldrich) using Lipofectamine RNAiMAX Transfection Reagent (13778, Thermo Fisher Scientific). Scramble siRNA (SIC001, Sigma-Aldrich) was used as a control. For *Snx25* deletion in BMDMs derived from *Cx3cr1*[CreERT2/WT];*Snx25*[fl/fl] mice, we treated cells with 1 μM 4-OHT (H7904, Sigma-Aldrich). For inhibition of proteasomes, we used 5 μM MG132 (M7449, Sigma-Aldrich) for 4 h. For depletion of macrophages in hind paw skin, we used clodronate liposomes (MKV300, Cosmo Bio). Empty liposomes (MKV300, Cosmo Bio) were used as a control.

### Clodronate liposome treatment

Twenty microliters of 10 mg ml[−1] clodronate liposomes or control liposomes were subcutaneously injected into the right side of the hind paw skin on days 0 and 3.

### Treatment with 4-OHT

For depletion of SNX25 in dMacs, we administered 4-OHT (40 ng μl[−1], 10 μl) by intradermal injection daily for 7 d into *Cx3cr1*[CreERT2/WT];*Snx25*[fl/fl]; Ai32[Tg/+] mice. Vehicle was injected into the contralateral side of the same animal. At 8 d after the last injection, a von Frey test was performed. For depletion of SNX25 in the DRG, we injected 4-OHT (200 ng μl[−1], 20 μl) into the exposed DRG (L4 and L5) of *Cx3cr1*[CreERT2/WT]; *Snx25*[fl/fl];Ai32[Tg/+] mice. At 5 d after administration, the von Frey test was performed.

### Immunohistochemistry

Mice were anesthetized and perfused transcardially with saline followed by 4% paraformaldehyde (09154-85, Nacalai Tesque) in 0.1 M PB (phosphate buffer) (pH 7.4). Skin, DRG, sciatic nerve and spinal cord were removed, post-fixed overnight in the same fixative and then immersed in 30% sucrose in PB overnight. Next, the tissues were frozen in powdered dry ice, embedded in Tissue-Tek OCT compound (4583, Sakura Finetek) and stored at −80 °C before sectioning. Eighteen-micrometer-thick sections were immersed in PBS containing 5% BSA and 0.3% Triton X-100 for 1 h. The antibodies rabbit anti-CGRP (1:100, CA-08-220, Genosys), rabbit anti-c-Fos (1:10,000, 226003, Synaptic Systems), rabbit anti-TRPV1 (1:100, KM018, Trans Genic), rabbit anti-TrkA (1:150, ab76291, Abcam), mouse anti-NF200 (1:1,000, N0142, Sigma-Aldrich), mouse anti-PGP9.5 (1:500, ab8189, Abcam), rat anti-MHC-II (1:100, NBP1-43312, Novus Biologicals), goat anti-CD206 (1:500, AF2535, R&D Systems), rabbit anti-Iba1 (1:500, 019-19741, Wako), rabbit anti-F4/80 (1:2,000, 28463-1-AP, Proteintech), rat anti-F4/80 (1:500, NB600-404, Novus Biologicals), rabbit anti-NGF (1:1,000, sc-548, Santa Cruz Biotechnology), rat anti-CD117 (1:100, MAB1356, R&D Systems), mouse anti-NeuN (1:150, MAB377, Millipore), mouse anti-GFAP (1:500, MAB360, Millipore), rat anti-CCR2 (1:200, NBP1-48337, Novus Biologicals), rat anti-GFP (1:5,000, 04404-84, Nacalai Tesque), rabbit anti-GFP (1:5,000, A6455, Thermo Fisher Scientific), rabbit anti-SNX25 (1:500, 13294-1-AP, Proteintech), biotin mouse anti-CD4 (1:200, 100403, BioLegend), biotin mouse anti-CD8a (1:200, 100703, BioLegend), biotin mouse anti-Ly6G/Ly6C (1:200, 108403, BioLegend), biotin mouse anti-NK1.1 (1:200, 108703, BioLegend) and biotin mouse anti-CD19 (1:200, 13-0193-81, eBioscience) were applied overnight at 4 °C. Alexa Fluor 488-conjugated IgG and Alexa Fluor 594-conjugated IgG (1:1,000, Life Technologies) were used as secondary antibodies. Sections were subjected to fluorescent Nissl staining (N21483, Molecular Probes). Images were captured using a

confocal laser scanning microscope (C2, Nikon). For immunostaining for SNX25 in the DRG, signals were detected by enhancing the signal with a TSA Plus kit (NEL763001KT, Akoya Biosciences) according to the manufacturer's instructions because the endogenous signal was low. For 3,3′-diaminozidine staining, 8-μm-thick sections were immersed in PBS containing 5% BSA and 0.3% Triton X-100 for 1 h. Mouse anti-PGP9.5 (1:500, ab8189, Abcam) antibodies were applied overnight at 4 °C. After immunoreaction with 3,3′-diaminozidine containing 0.03% $H_2O_2$ solution, sections were enclosed with mounting medium.

## Microarray
Total RNA was isolated from the BM of C57BL/6 mice and $Mlc1^{Tg}$ mice using a NucleoSpin RNA kit (740955, Takara Bio). RNA samples were analyzed with Affymetrix GeneChip Mouse Genome 430 2.0 Arrays by Takara Bio.

## Next-generation sequencing
Whole-genome DNA was isolated from $Mlc1^{Tg}$ mice using a NucleoBond AXG column (Takara Bio). Identification of the loci of transgene insertion was performed by Takara Bio, followed by next-generation sequencing on the Illumina sequencing platform.

## Quantitative PCR with reverse transcription
Total RNA of cells or tissues was extracted using a NucleoSpin RNA kit (740955, Takara Bio) or the Arcturus PicoPure RNA Isolation Kit (Applied Biosystems). Total RNA extracts were reverse transcribed using random primers and the QuantiTect Reverse Transcription Kit (205311, Qiagen), according to the manufacturer's instructions. Real-time PCR was performed using a LightCycler Quick System 350S (Roche Diagnostics) or the Thermal Cycler Dice Real Time System (Takara Bio), with THUNDERBIRD SYBR qPCR Mix (QPS-201, Toyobo). PCR primers used in this study were as follows (all 5′ → 3′): *β-actin* (*Actb*) sense primer, AGCCATGTACGTAGCCATCC; *β-actin* antisense primer, CTCTCAGCTGTGGTGGTGAA; *Ccl2* sense primer, CCCACTCACCTGCTGCTACT; *Ccl2* antisense primer, TCTGGACCCATTCCTTCTTG; *Ccl3* sense primer, ATGAAGGTCTCCACCACTGC; *Ccl3* antisense primer, CCCAGGTCTCTTTGGAGTCA; *Ccl4* sense primer, GCCCTCTCTCTCCTCTTGCT; *Ccl4* antisense primer, GTCTGCCTCTTTTGGTCAGG-3′; *Cxcl2* sense primer, AGTGAACTGCGCTGTCAATG; *Cxcl2* antisense primer, TTCAGGGTCAAGGCAAACTT; *Gfp* sense primer, AGCTGACCCTGAAGTTCATCTG; *Gfp* antisense primer, AAGTCGTGCTGCTTCATGTG; *Mlc1* sense primer, CTGACTCAAAGCCCAAGGAC; *Mlc1* antisense primer, AGCGCAAATAATCCATCTCG; *Mov10l1* sense primer, TGCTTCTGAACGTGGGACAGG; *Mov10l1* antisense primer, ACACAGCCAATCAGCACTCTGG; *Ngf* sense primer, TCAGCATTCCCTTGACACAG; *Ngf* antisense primer, GTCTGAAGAGGTGGGTGGAG; *Scn9a* sense primer, AAGGTCCCAAGCCCAGTAGT; *Scn9a* antisense primer, AGGACTGAAGGGAGACAGCA; *Scn10a* sense primer, GCCTCAGTTGGACTTGAAGG; *Scn10a*, antisense primer, AGGGACTGAAGAGCCACAGA; *Snx25* sense primer, CATGGATCGTGTTCTGAGAG; *Snx25* antisense primer, GAAGTCATCTAAGAGCAGGATGG; *Trpv1* sense primer, CCCTCCAGACAGAGACCCTA; and *Trpv1* antisense primer, GACAACAGAGCTGACGGTGA. *Snx25* primers used in PCR after sorting samples by flow cytometry were as follows: sense primer, TGTGGACGGGAAGAAGGATTCC; antisense primer, GCACCCATTTAAACATTCCTCGAAG; probe, [6FAM]CCGATCAGCATGAAACACGGTTCAGCCA[TAM]. Real-time PCR was performed using a Thermal Cycler Dice Real Time System (Takara Bio), with the Takara Probe qPCR mix (Takara Bio). Quantification of gene expression was calculated by the ΔΔCt method[43]. Briefly, the relative concentrations of the target gene and the reference gene (encoding β-actin) were measured, and the relative concentration of the unknown concentration sample with respect to the reference gene was comparatively quantified. The signal value was denoted as a fold change corrected by the signal value of the control ($Snx25^{+/+}$ mice, scramble siRNA, etc.).

## Immunoblot analysis
Samples (cells or tissues) were lysed with 10 mM Tris, pH 7.4, containing 150 mM NaCl, 5 mM EDTA, 1% Triton X-100, 1% deoxycholic acid and 0.1% SDS. The homogenate was centrifuged at 20,600$g$ for 5 min, and the supernatant was stored at −20 °C. Protein concentration was measured using a bicinchoninic acid protein assay kit (23225, Thermo Fisher Scientific). Equal amounts of protein per lane were electrophoresed on SDS–polyacrylamide gels and then transferred to a polyvinylidene difluoride membrane. Blots were probed with rabbit anti-SNX25 (1:1,000, 13294-1-AP, Proteintech), rabbit anti-TRPV1 (1:100, KM018, TransGenic), rabbit anti-TrkA (1:10,000, ab76291, Abcam), goat anti-CD206 (1:1,000, AF2535, R&D Systems), rabbit anti-NGF (1:200, sc-548, Santa Cruz Biotechnology), rabbit anti-Nrf2 (1:500, sc-722, Santa Cruz Biotechnology), rabbit anti-HO-1 (1:500, ADI-SPA-896, Enzo Life Sciences), rabbit anti-TGF-βRI (1:200, sc-398, Santa Cruz Biotechnology) and rabbit anti-GAPDH (1:2,000, ABS16, Merck Millipore) antibodies. Immunoblot analysis was performed with horseradish peroxidase-conjugated anti-rabbit and anti-goat IgG using enhanced chemiluminescence immunoblot detection reagents (297-72403 or 290-69904, Wako). Data were acquired in arbitrary densitometric units using ImageJ software.

## Co-immunoprecipitation
Cells were lysed with 50 mM Tris, pH 7.4, containing 150 mM NaCl, 1% NP-40, 0.5% deoxycholic acid and protease inhibitor cocktail (25955-24, Nacalai Tesque) and then incubated at 4 °C for 20 min with rotation. The lysate was centrifuged at 21,500$g$ for 15 min, and the supernatant was collected. A rabbit IgG against Nrf2 (sc-722, Santa Cruz Biotechnology) was incubated with SureBeads Protein G Magnetic Beads (Bio-Rad) for 10 min. The mixture was added to the supernatant for immunoprecipitation and incubated for 1 h with rotation, and then the immunobound protein was eluted.

## Primary DRG neurons
DRGs from $Snx25^{+/-}$ and WT littermate mice were quickly collected in DMEM/F12 medium and incubated for 90 min at 37 °C in a solution of 0.2% collagenase. After dissociation, DRGs were transferred to a tube containing DMEM/F12 supplemented with 10% FBS and 1% penicillin–streptomycin solution. Ganglia were gently triturated using pipettes. After centrifugation, cells were resuspended in DMEM/F12 supplemented as described above and plated on poly-L-lysine-coated culture dishes. Neurons were kept at 37 °C with 5% $CO_2$, and the medium was changed to DMEM/F12 with B27 supplement 8 h after plating.

## Fluo-4 calcium ion assay
DRG neurons were seeded in 96-well cell culture plates at a density of $1.5 \times 10^4$ cells per well and cultured overnight. Intracellular calcium ion responses to capsaicin were measured using the calcium kit II Fluo-4 (CS32, Dojindo) in accordance with the manufacturer's instructions. The temperature of the platform was set to 37 °C. Cells were fluorescently imaged at an excitation wavelength of 495 nm every 7 s, and the fluorescence intensities of neurons were quantified at 515 nm. Fluorescence intensities of neurons were quantified simultaneously for the entire well. Capsaicin (1 μM) was added to measure the response.

## Bone marrow transplantation
BM recipients were 8-week-old male C57BL/6J, $Snx25^{+/+}$, $Snx25^{+/-}$ or $Snx25^{fl/fl}$ mice. Mice were intraperitoneally injected with the chemotherapeutic agent busulfan (30 μg per g body weight; B2635, Sigma-Aldrich) in a 1:4 solution of dimethyl sulfoxide and PBS at 7, 5 and 3 d before BM transfer. All mice were treated with antibiotics (trimethoprim (35039, Nacalai Tesque) and sulfamethoxazole (S7507, Sigma) for 14 d after busulfan treatment. BM-derived cells were obtained from the femur and tibia of 5-week-old GFP mice, $Snx25^{+/+}$ mice, $Snx25^{+/-}$ mice, $Cx3cr1^{CreERT2/WT};Snx25^{fl/fl}$ or $Cx3cr1^{CreERT2/WT};Snx25^{fl/fl};Ai32^{Tg/+}$ mice and resuspended in PBS with 2% FBS. BM-derived cells ($1 \times 10^6$ cells) were

transferred to recipient mice by tail vein injection. For quantitative analysis, engraftment was verified by determining the percentage of GFP-expressing cells in the blood. We counted the numbers of GFP+ cells in peripheral blood by flow cytometry and confirmed efficient chimerism as demonstrated by the large proportions of circulating blood leukocytes expressing GFP.

## BM-derived macrophage culture

BM cells were obtained from the femur and tibia of 8-week-old male C57BL/6, *Snx25*[+/+], *Snx25*[+/−], *Cx3cr1*[CreERT2/WT];*Snx25*[fl/fl] or *Cx3cr1*[CreERT2/WT];*Snx25*[fl/fl];Ai32[Tg/+] mice and cultured in RPMI-1640 medium containing 10% FBS, 1% penicillin–streptomycin and macrophage colony-stimulating factor (315-02, PeproTech, 5 ng ml⁻¹). After 6 d, BMDMs were transferred to 3.5-mm dishes in RPMI-1640 containing 10% FBS and 1% penicillin–streptomycin. For the flow cytometry experiment using BMDMs from *Cx3cr1*[CreERT2/WT];*Snx25*[fl/fl];Ai32[Tg/+] mice, cells were treated with 4-OHT (1 μM) for 7–8 d before collection. The medium was changed daily.

## PCR array

The mouse inflammatory response and autoimmunity RT2 Profiler PCR Array kit (PAMM-077Z, Qiagen) in a 96-well format was used. This kit profiles the expression of 84 genes that encode inflammatory response, autoimmunity and other genes related to inflammation. Hind paw skins were quickly dissected 3 d after formalin injection, frozen rapidly and stored at −80 °C until use. Total RNA was purified using the NucleoSpin RNA kit (Takara Bio) in accordance with the manufacturer's instructions. cDNA was obtained from purified RNA using the RT2 First Strand Kit (Qiagen) provided with the PCR Array kit. cDNA template mixed with PCR master mix was dispensed into each well, and real-time PCR was performed. Three independent arrays (corresponding to three animals) were performed. Quantification of gene expression was calculated by the ΔΔCt method[43]. The signal value was expressed as a fold change corrected by the signal value of the control (*Snx25*[+/+] mice or *Snx25*[fl/fl] mice).

## Fluorescent in situ hybridization

Fluorescent in situ hybridization was performed with a probe targeting *Cx3cr1* mRNA using the RNAscope Fluorescent Multiplex Reagent Kit (320850, Advanced Cell Diagnostics) according to the manufacturer's instructions.

## Nerve-ligation assay

To assess NGF–TrkA complex trafficking from the periphery toward DRG cell bodies, we carefully exposed the left sciatic nerve and tightly ligated the nerve with one 6.0 suture in WT and *Snx25*[+/−] mice. Eight hours after the surgery, mice were terminally anesthetized and quickly perfused with 4% paraformaldehyde. After perfusion, the left sciatic nerve was excised, post-fixed for 24 h in the same perfusion fixative, cryoprotected in 30% sucrose for 48 h at 4 °C and then frozen in tissue freezing compound. Longitudinal sections (18 μm) of the left sciatic nerve were cut on a cryostat and then stored at −30 °C before staining. Sciatic nerve sections were stained with rabbit anti-TrkA (1:150, ab76291, Abcam) primary antibody. Alexa Fluor 594-conjugated IgG (Life Technologies) was used as the secondary antibody.

## Generation of constructs and transient transfection of 293T cells

PCR cloning was performed to amplify *Snx25* and *Nrf2* cDNA with a primer having an optimal Kozak consensus sequence just before the in-frame first ATG of the mouse *Snx25* and *Nrf2* genes. Fragments were inserted into the pcDNA3.1/Myc-His vector (Invitrogen). Using the Lipofectamine reagent (11668-019, Invitrogen), 293T cells (ECACC 12022001) were transfected with an *Snx25* and *Nrf2* construct according to the manufacturer's instructions.

## Flow cytometry

For the analysis of myeloid populations in the skin, cells were obtained from hind paw skin or back skin using a Multi Tissue Dissociation Kit 1 (130-110-201, Miltenyi Biotec) with the gentleMACS Dissociator (Miltenyi Biotec) according to the manufacturer's instructions. The back skin, dehaired using forceps, or hind paw skin were minced by razor blade and then subjected to enzymatic digestion at 37 °C for 2 h with rotation. During the enzymatic digestion, cells were dispersed by the programs (h_tumor_01, h_tumor_02 and h_tumor_03) of gentleMACS. Debris was removed with a 70-μm cell strainer. In some experiments, cells were separated with 30/70% Percoll (17-0891, GE Healthcare) by centrifugation for 20 min at 400*g*. Cells were stained with various combinations of monoclonal antibodies. Fc-γII and Fc-γIII receptors were blocked by prior incubation with anti-CD16–CD32 antibody (1:100, 156604, BioLegend). The monoclonal antibodies used in this study were PE–anti-CD45 (1:100, 103106, BioLegend), biotin anti-Ly6G–Ly6C (1:100, 108403, BioLegend), biotin anti-CD19 (1:100, 115504, BioLegend), biotin anti-CD8a (1:100, 100704, BioLegend), biotin anti-CD4 (1:100, 100404, BioLegend), biotin anti-NK1.1 (1:100, 108704, BioLegend), biotin anti-MHC-II (1:100, 107603, BioLegend) or biotin anti-F4/80 (1:100, 123105, BioLegend), Alexa Fluor 488–anti-CD11b (1:100, 101219, BioLegend), Alexa Fluor 647–anti-F4/80 (1:100, 123121, BioLegend), PE-Cy7–anti-CD45 (1:100, 103133, BioLegend), FITC–anti-MHC-II (1:100, 107605, BioLegend), PE–anti-CD11b (1:100, 101207, BioLegend), APC–anti-CD64 (1:25, 139306, BioLegend), APC-Cy7–anti-Ly6C (1:100, 128025, BioLegend), Brilliant Violet 421–anti-CD24 (1:100, 101825, BioLegend), biotin anti-CD3 (1:100, 100243, BioLegend), biotin anti-TER119 (1:100, 116203, BioLegend), biotin anti-Ly6G (1:100, 127603, BioLegend), PerCP-Cy5.5–streptavidin (1:200, 405214, BioLegend) and APC–streptavidin (1:200, 405207, BioLegend). To exclude dead cells from analysis, cells were stained with PI (421301, BioLegend), 7-AAD (559925, BD Biosciences) or Fixable Viability Stain 700 (564997, BD Biosciences). Cells were analyzed and sorted using the FACSAria (BD Biosciences) or the Cell Sorter SH800S (Sony). Data were processed with FlowJo (version 10) (Tree Star). Purity of the CD45+CD11b^hiF4/80^hi population from the hind paw skin was more than 96%. For isolation of myeloid populations from the back skin, sticky or dead cells, which nonspecifically bind to microbeads and/or the column, were removed with Basic MicroBeads (130-048-001, Miltenyi Biotec) with an autoMACS separator, and then leukocytes were enriched with CD45 MicroBeads (Miltenyi Biotec, 130-052-301). dMacs (CD64+Ly6C−MHC-II+), dermal monocytes (CD64−Ly6C+MHC-II^lo) and dermal dendritic cells (CD64−Ly6C−MHC-II+) in CD11b+Lin (CD3, CD19, CD34, Ly6G, NK1.1 and TER119)-negative cells were isolated using a Cell Sorter SH800 (>94% purity). For analysis of BMDM generated from *Cx3cr1*[CreERT2/WT];*Snx25*[fl/fl];Ai32[Tg/+] mice, BMDMs were collected and stained with PI to exclude dead cells from analysis. YFP-positive or -negative cells were isolated with the Cell Sorter SH800 (Sony).

## Quantification and statistical analysis

Quantifications were performed from at least three independent experimental groups. Data are presented as mean ± s.e.m. Statistical analyses were performed using Student's *t*-test or Welch's *t*-test for two groups or one-way ANOVA for multiple groups, and significant differences between group means were identified with the Tukey–Kramer test (IBM statistics software (SPSS) version 23 and Statcel–The Useful Addin Forms on Excel–4th edn (Yanai, H., OMS, Tokyo, Japan, 2015). Data distribution was assumed to be normal, but this was not formally tested. All statistical tests were two-tailed, and $P < 0.05$ was considered significant. Statistical significance is indicated as asterisks (*$P < 0.05$, **$P < 0.01$). All sample numbers (*n*) are indicated in figure legends. Sample size was determined to be adequate based on the magnitude and consistency of measurable differences between groups. We confirmed that replicate experiments were successful by repeating at least three times for all experiments. Data collection (except for the behavioral test) and analysis were not performed blind to the conditions of the experiments.

**Reporting summary**

Further information on research design is available in the Nature Portfolio Reporting Summary linked to this article.

## Data availability

All data supporting the findings of this study are found within the text and its Supplementary Information and are available from the corresponding author upon reasonable request. Source data are provided with this paper.

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

## Acknowledgements

We thank M. Banja (Nara Medical University), K. Sango (Tokyo Metropolitan Institute of Medical Science), S. Wang (Hyogo University of Health Sciences), M. Yamano (Nara Medical University), K. Nakahara (Nara Medical University), S. Mitani (Osaka Dental University), A. Kamimura (Nara Medical University) and Y. Sumitomo (Nara Medical University) for technical assistance. We also thank M. Ikawa (Osaka University) for valuable suggestions. This work was supported by JSPS KAKENHI JP16K08451 (to H.O.), JP16K20112 (to Y.T.), JP18K16492 (to T.S.), JP19K07827 (to T.T.), JP19K18303 (to Y.T.), JP19K16480 (to A.I.), JP21K06758 (to A.W.), the Osaka Medical Research Foundation for Intractable Diseases (to T.T.), the Takeda Science Foundation (to T.T.), the Nakatomi Foundation (to T.T.), the Naito Foundation (to T.T.) and the Ichiro Kanehara Foundation (to T.T.).

## Author contributions

T.T., H.O., K.T. and A.W. conceived the project and designed the experiments. T.T., A.I., H.O., Y.T., T.S., S.T., M.K., K.N. and K.T. performed experiments. T.T., A.I., H.O., M.K., H.F., T.I. and K.T. analyzed data. T.T. and A.W. wrote the paper. A.W. coordinated and directed the project.

## Competing interests

The authors declare no competing interests.

## Additional information

**Extended data** is available for this paper at https://doi.org/10.1038/s41590-022-01418-5.

**Correspondence and requests for materials** should be addressed to Tatsuhide Tanaka or Akio Wanaka.

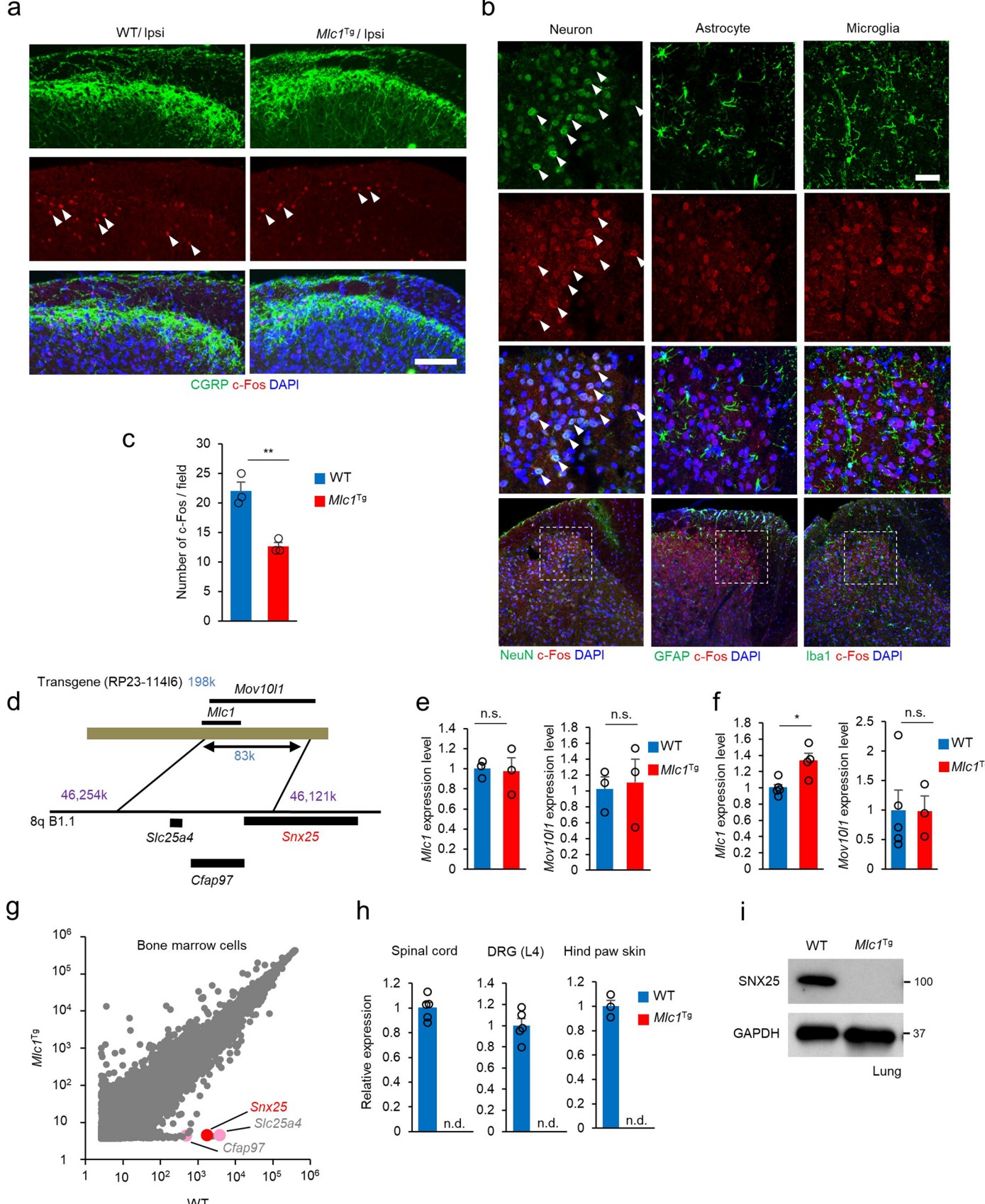

**Extended Data Fig. 1 | See next page for caption.**

**Extended Data Fig. 1 | Identification of the transgene insertion site in *Mlc1*[Tg] mice.** (**a**) Confocal microscopic images of the dorsal horn of the spinal cord (L4, 30 min after formalin injection, Ipsi: ipsilateral side to injection) immunolabeled for CGRP and c-Fos of WT and *Mlc1*[Tg] mice. Arrowheads denoted double-labeled cells. Scale bar, 100 μm. (**b**) Confocal microscopic images of the dorsal horn of the spinal cord immunolabeled for c-Fos (red) and cell markers (neuron, NeuN; astrocyte, GFAP; microglia, Iba1) of WT mice. Arrowheads denote double-labeled cells. Scale bar, 100 μm. (**c**) Quantification of c-Fos$^+$ activated neurons in WT and *Mlc1*[Tg] mice (WT: n = 3; *Mlc1*[Tg]: n = 3, $p$ = 0.005). (**d**) Diagram showing the insertion site of the RP23-114I6 transgene in chromosome 8 of the *Mlc1*[Tg] mice. The positions of three endogenous genes (*Snx25*, *Slc25a4*, and *Cfap97*) relative to the inserted transgene are indicated. (**e**) mRNA expression levels for *Mlc1* ($p$ = 0.856) and *Mov10l1* ($p$ = 0.816) in the brain of WT and *Mlc1*[Tg] mice (WT: n = 3; *Mlc1*[Tg] mice: n = 3). (**f**) mRNA expression levels for *Mlc1* ($p$ = 0.011) and *Mov10l1* ($p$ = 0.967) in the spinal cord of WT and *Mlc1*[Tg] mice (WT: n = 5; *Mlc1*[Tg] mice: n = 4). (**g**) cDNA microarray data of bone marrow cells of WT and *Mlc1*[Tg] mice (WT: n = 3; *Mlc1*[Tg] mice: n = 3) were plotted (Y axis: *Mlc1*[Tg]; X axis: C57BL/6 WT mice). *Snx25*, *Slc25a4*, and *Cfap97* are indicated by colored dots. (**h**) RT-qPCR analyses of *Snx25* mRNA in the spinal cord (WT: n = 5; *Mlc1*[Tg] mice: n = 4), DRG (WT: n = 5; *Mlc1*[Tg] mice: n = 5), and hind paw skin (WT: n = 3; *Mlc1*[Tg] mice: n = 5). (**i**) Immunoblot analysis showing the expression of SNX25 protein in the lung of WT and *Mlc1*[Tg] mice. Results are represented as mean ± SEM. Statistical analyses were performed using the two-tailed Student's $t$-test. *$p$ < 0.05, **$p$ < 0.01. n.s., not significant. n.d., not detected. Representative of two independent experiments (**a** and **b**).

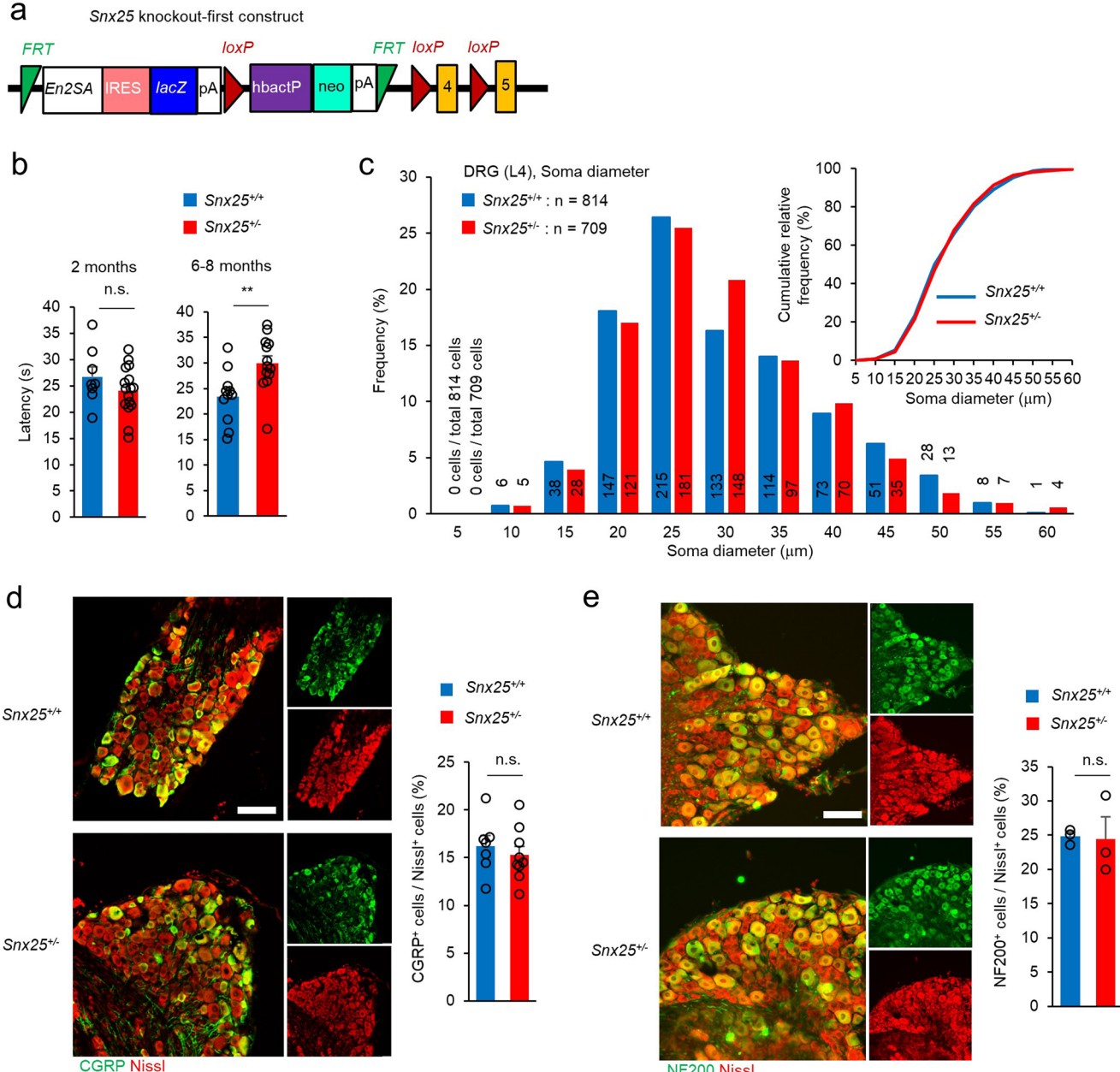

**Extended Data Fig. 2 | DRG neurons are normal in *Snx25*[+/−] mice. (a)** Scheme of the targeting construct used to knock out the *Snx25* gene. **(b)** Hot plate test of WT and *Snx25*[+/−] mice. Left panel shows latencies of 2-month-old mice (WT: n = 8; *Snx25*[+/−]: n = 16, p = 0.222). Right panel shows those of 6–8-month-old mice (WT: n = 11; *Snx25*[+/−]: n = 13, p = 0.006). s, second. **(c)** Size distribution of DRG neuron diameters is plotted for WT and *Snx25*[+/−] mice (L4 level, WT (blue columns): 814 cells from 3 mice; *Snx25*[+/−] mice (red columns): 709 cells from 3 mice). X-axis values indicate the maximum diameter in each 5-μm range (for example, 10 indicates diameters ranging from 5 μm to 10 μm). Numbers above or inside columns are the actual numbers of cells in each diameter range. Inset, cumulative frequency distribution of soma diameters. **(d)** Left, representative confocal

microscopic images showing CGRP-immunoreactive neurons in the DRG (L4) of WT and *Snx25*[+/−] mice. Scale bar, 100 μm. Right, percentage of CGRP-positive cells among Nissl-positive cells in WT and *Snx25*[+/−] mice (WT: n = 7; *Snx25*[+/−]: n = 9 DRG sections from at least 3 different mice of each genotype, p = 0.521). **(e)** Left, representative confocal microscopic images of fluorescent Nissl-stained and NF200-immunoreactive neurons in the DRG (L4) of WT and *Snx25*[+/−] mice. Scale bar, 100 μm. Right, percentage of NF200-positive cells among Nissl-positive cells (WT: n = 3; *Snx25*[+/−]: n = 3 DRG sections from at least 3 different mice of each genotype, p = 0.903). Results are represented as mean ± SEM. Statistical analyses were performed using the two-tailed Student's t-test. **p < 0.01. n.s., not significant. Representative of three independent experiments (**d** and **e**).

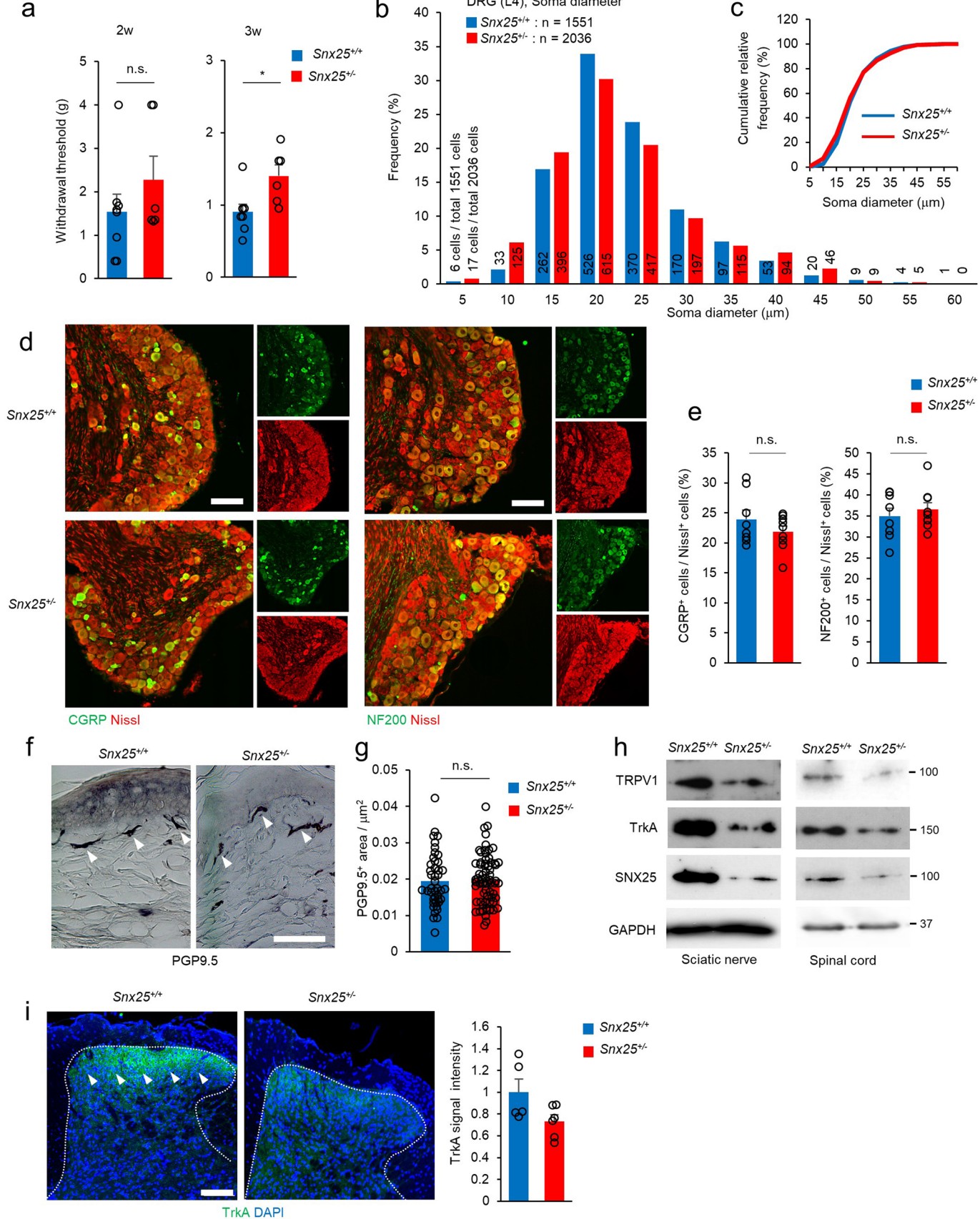

**Extended Data Fig. 3 | See next page for caption.**

**Extended Data Fig. 3 | *Snx25*+/− mice develop normally. (a)** VF thresholds of young (2 and 3 weeks after birth) WT (n = 8) and *Snx25*+/− mice (n = 6). 2w, *p* = 0.288; 3w, *p* = 0.016. g, gram. **(b)** Size distribution of DRG neuron diameters is plotted for 3-week-old WT and *Snx25*+/− mice (L4 level, WT (blue columns): 1551 cells from 3 mice; *Snx25*+/− mice (red columns): 2036 cells from 3 mice). X-axis values indicate the maximum diameter in each 5-μm range. Numbers above or inside columns are the actual numbers of cells in each diameter range. **(c)** Cumulative frequency distribution of DRG neuron diameters. **(d)** Representative confocal microscopic images showing CGRP- and NF200-immunoreactive neurons in the DRG (L4) of WT and *Snx25*+/− mice. Scale bars, 100 μm. **(e)** Left, percentage of CGRP+ cells among Nissl-positive cells (WT: n = 8; *Snx25*+/−: n = 9 DRG sections from at least 3 different mice of each genotype, *p* = 0.27). Right, percentage of NF200+ cells among Nissl-positive cells (WT: n = 8; *Snx25*+/−: n = 9 DRG sections from at least 3 different mice of each genotype, *p* = 0.544).

**(f)** Representative images of PGP9.5+ immunoreactivities in the plantar skin of WT and *Snx25*+/− mice. Arrowheads indicate neuronal fibers labeled for PGP9.5. Scale bar, 50 μm. **(g)** PGP9.5+ region per unit area (μm²) was measured and plotted for each genotype (WT: 44 sections from 3 mice; *Snx25*+/−: 68 sections from 4 mice, *p* = 0.805). **(h)** Representative immunoblot showing TRPV1, TrkA, and SNX25 proteins in the sciatic nerve and spinal cord of WT and *Snx25*+/− mice. **(i)** Representative immunofluorescence images showing TrkA-immunoreactive terminals (green fluorescence, arrowheads) in the dorsal horn of the WT and *Snx25*+/− mice counterstained with DAPI (blue fluorescence). Right, quantification of mean fluorescence intensity (WT: n = 5; *Snx25*+/−: n = 6 spinal cord sections from 3 different mice of each genotype, *p* = 0.066). Scale bar, 100 μm. Results are represented as mean ± SEM. Statistical analyses were performed using the two-tailed Student's *t*-test. *$p < 0.05$. n.s., not significant. Representative of three independent experiments (**d**, **f**, and **i**).

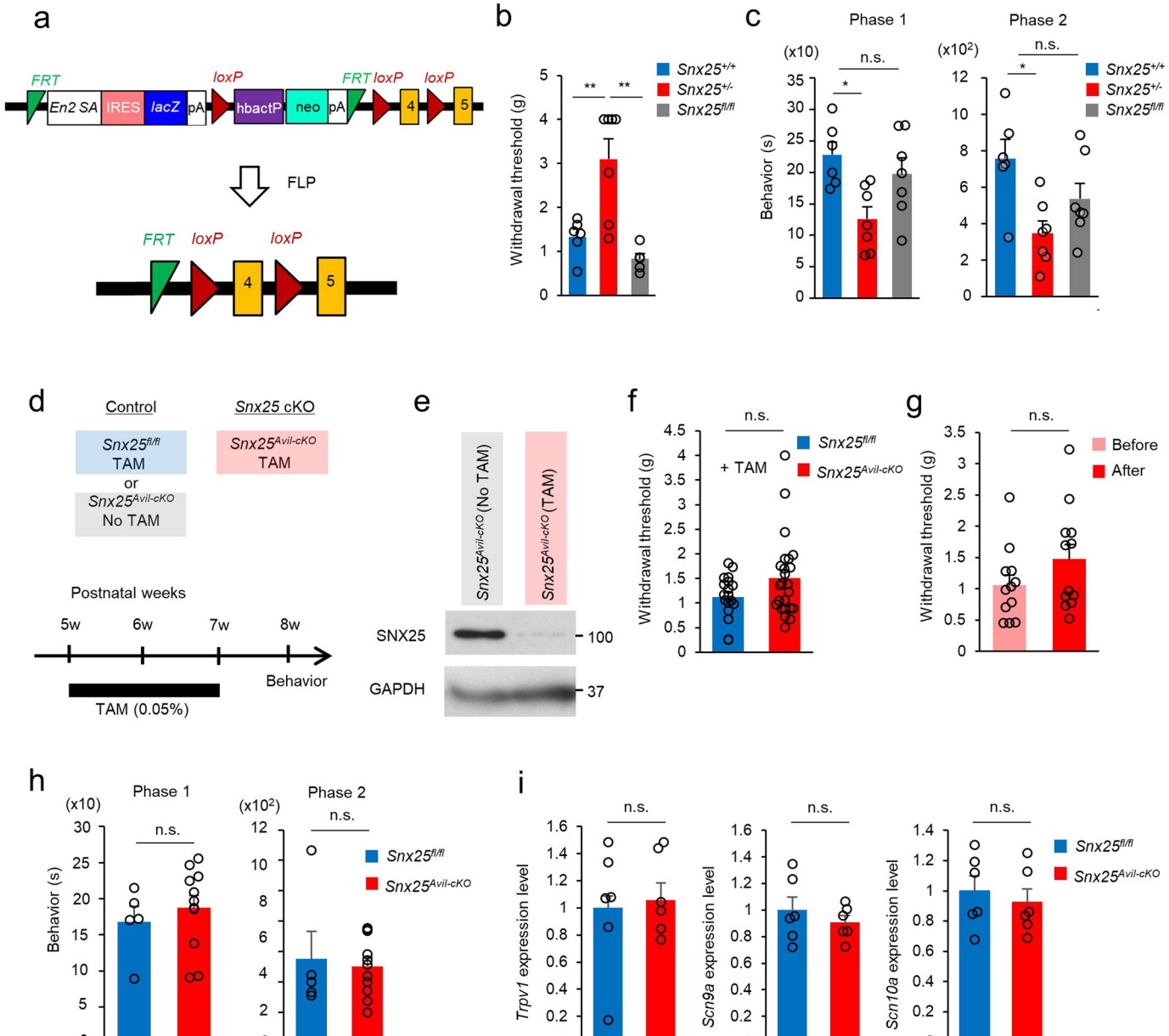

**Extended Data Fig. 4 | *Snx25* cKO in DRGs does not yield the pain-insensitive phenotype. (a)** Schematic representation showing that the initial targeting construct (upper; knock-out first construct) for the *Snx25* gene was transformed to an *Snx25*[fl/fl] (for exon 4) construct after recombination by flippase (FLP). **(b)** VF thresholds of WT, *Snx25*[+/−], and *Snx25*[fl/fl] mice are shown (WT: n = 6; *Snx25*[+/−]: n = 7; *Snx25*[fl/fl]: n = 5). *Snx25*[+/+]/*Snx25*[+/−], p = 0.004; *Snx25*[+/+]/*Snx25*[fl/fl], p = 0.001. g, gram. **(c)** Formalin responses in the phase 1 (left, 0–10 min, *Snx25*[+/+]/*Snx25*[+/−], p = 0.014) and phase 2 (right, 20–60 min, *Snx25*[+/+]/*Snx25*[+/−], p = 0.011) for three lines of mice (WT: n = 6; *Snx25*[+/−]: n = 7; *Snx25*[fl/fl]: n = 7). s, second. **(d)** Experimental paradigm and schedule. The *Avil Cre* driver functions specifically in DRG neurons. TAM feeding for two weeks was employed to differentiate control (No TAM) and experimental (TAM) groups. Another control was *Snx25*[fl/fl] mice without *Cre* driver and with TAM feeding. **(e)** Representative immunoblotting showed the expression levels of SNX25 in the DRG (L4) of *Snx25*[Avil-cKO] mice in the presence or absence of TAM. **(f)** VF thresholds are plotted for *Snx25*[fl/fl] and *Snx25*[Avil-cKO] mice treated with TAM (*Snx25*[fl/fl]: n = 19; *Snx25*[Avil-cKO]: n = 24, p = 0.068). g, gram. **(g)** VF thresholds were plotted for *Snx25*[Avil-cKO] mice before and after TAM treatment (n = 12, p = 0.165). g, gram. **(h)** Formalin responses of the two groups of mice (*Snx25*[fl/fl]: n = 5, *Snx25*[Avil-cKO]: n = 11). The responses in the phase 1 (left, 0–10 min, p = 0.529) and phase 2 (right, 20–60 min, p = 0.747) are indicated. **(i)** Expression profiles of mRNAs for pain-related factors (*Trpv1*, *Scn9a*, *Scn10a*) in DRG (L4) of *Snx25*[fl/fl] and *Snx25*[Avil-cKO] are shown (*Snx25*[fl/fl]: n = 6; *Snx25*[Avil-cKO]: n = 6). *Trpv1*, p = 0.687; *Scn9a*, p = 0.422; *Scn10a*, p = 0.576. Results are represented as mean ± SEM. Statistical analyses were performed using the two-tailed Student's *t*-test (**f**–**i**) or one-way ANOVA (**b** and **c**), and significant differences between group means were identified with the Tukey–Kramer test. *p < 0.05, **p < 0.01. n.s., not significant.

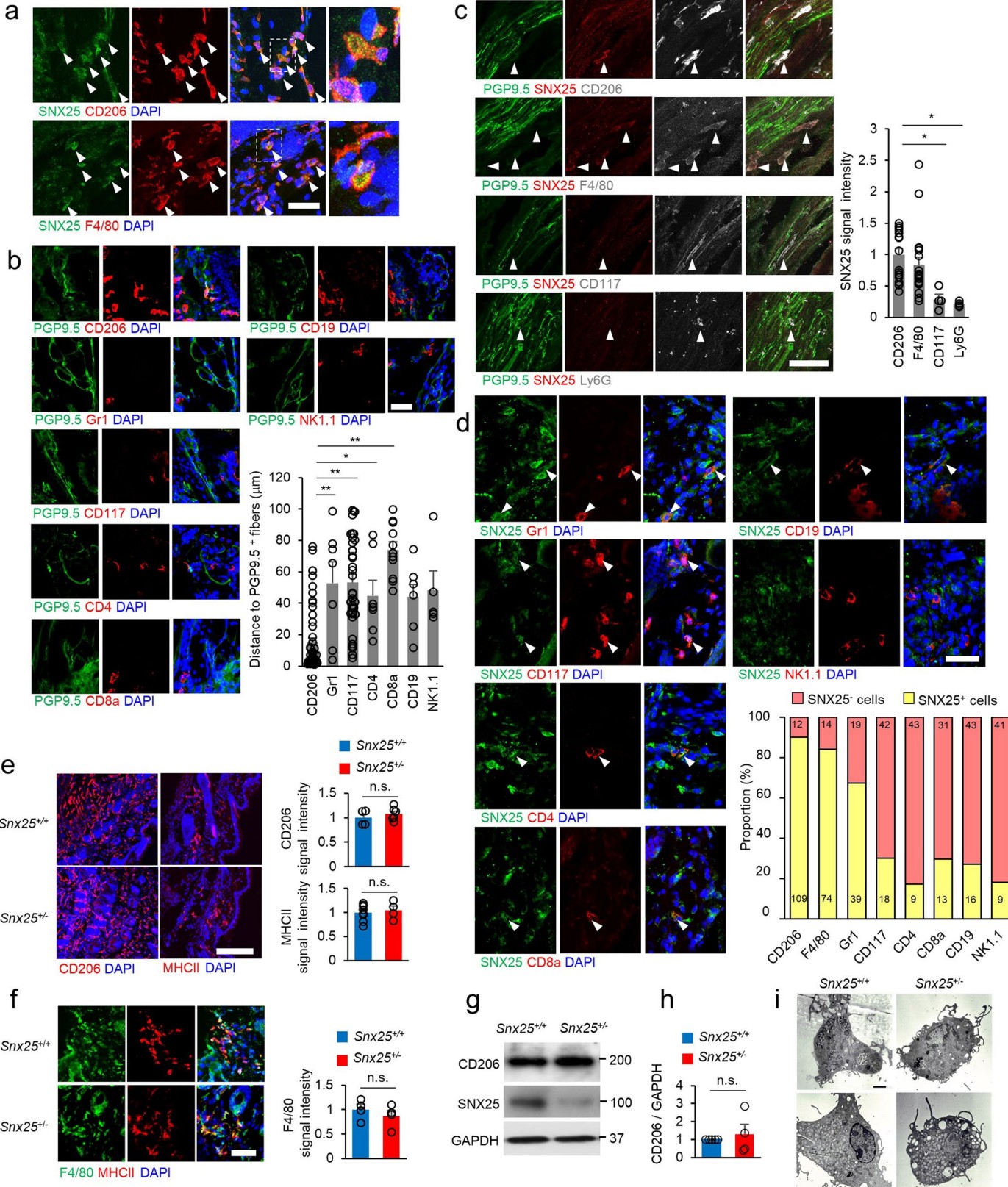

**Extended Data Fig. 5 | See next page for caption.**

**Extended Data Fig. 5 | Dermal macrophages located near nerve fibers express SNX25.** (**a**) Confocal microscopic images of the hind paw skin immunolabeled for SNX25 and macrophage markers (CD206 (upper panels) or F4/80 (lower panels)) of WT mice. Scale bar, 50 μm. The rightmost panels are enlarged images of the boxed areas in the panels to their left. Arrowheads indicate cells double labeled for SNX25 and macrophage markers. (**b**) Confocal microscopic images of hind paw skin immunolabeled for PGP9.5 (green) and the indicated cell-specific markers (red) of WT mice. Scale bar, 50 μm. The summary graph shows the distance between a PGP9.5$^+$ fiber and each specific marker$^+$ cell. CD206: n = 56; Gr1: n = 7; CD117: n = 36; CD4: n = 7; CD8a: n = 11; CD19: n = 6; NK1.1: n = 5 from at least 3 different mice. The other myeloid cells are considerably further from fibers, but a few neutrophils (Gr1) and mast cells (CD117) are localized near fibers. CD206/Gr1, $p = 0.003$; CD206/CD117, $p = 6.12e\text{-}10$; CD206/CD4, $p = 0.046$; CD206/CD8a, $p = 6.59e\text{-}10$. (**c**) Confocal microscopic images of the hind paw skin immunolabeled for PGP9.5 (green), SNX25 (red), and the indicated cell-specific markers (white) of WT mice. Scale bar, 50 μm. The summary graph shows the SNX25 fluorescence intensity in each specific marker$^+$ cell located near a fiber. CD206: n = 15; F4/80: n = 19; CD117: n = 4; Ly6G: n = 4 from at least 3 different mice. CD206/CD117, $p = 0.036$; CD206/Ly6G, $p = 0.017$. (**d**) Confocal microscopic images of hind paw skin immunolabeled for SNX25 and specific cell markers (Gr1, CD117, CD4, CD8a, CD19, and NK1.1) in WT mice. Each antibody was used as a marker for neutrophils, mast cells, helper T cells, killer T cells, B cells, and NK cells, respectively. Scale bar, 50 μm. The summary graph shows the proportion of SNX25$^+$ cells (yellow column) in each specific marker$^+$ cells. The percentages of SNX25$^+$ cells are 90.1% (CD206), 84.1% (F4/80), 67.2% (Gr1), 30.0% (CD117), 17.3% (CD4), 29.5% (CD8a), 27.1% (CD19), and 18.0% (NK1.1). Numbers inside columns are the actual numbers of cells counted. (**e**) Confocal microscopic images of the hind paw skin immunolabeled for CD206 or MHCII in WT and *Snx25*$^{+/-}$ mice. Right panel shows quantification of signal intensities for CD206 and MHCII in the two groups of mice (CD206, WT: n = 4; *Snx25*$^{+/-}$: n = 6, MHCII, WT: n = 8; *Snx25*$^{+/-}$: n = 4 hind paw skin sections from at least 3 different mice). CD206, $p = 0.424$; MHCII, $p = 0.693$. Scale bar, 200 μm. (**f**) Confocal microscopic images of the hind paw skin immunolabeled for F4/80 and MHCII in WT and *Snx25*$^{+/-}$ mice. Right panel shows quantification of F4/80 signal intensity in the two group of mice (n = 4 hind paw skin sections from 2 different mice, $p = 0.429$). Scale bar, 50 μm. (**g**) Representative immunoblot shows the expressions of CD206 and SNX25 proteins in the hind paw skins of WT and *Snx25*$^{+/-}$ mice. (**h**) Semi-quantitation of CD206 protein levels in the hind paw skins of WT and *Snx25*$^{+/-}$ mice (WT: n = 4; *Snx25*$^{+/-}$: n = 4, $p = 0.649$). (**i**) Electron microscopic images of representative BMDMs derived from WT and *Snx25*$^{+/-}$ mice. Note that there is no overt difference in the morphology. Scale bar, 2 μm. Results are represented as mean ± SEM. Statistical analyses were performed using the two-tailed Student's *t*-test (**e**, **f**) or two-tailed Welch's *t*-test (**h**) or one-way ANOVA (**b** and **c**), and significant differences between group means were identified with the Tukey–Kramer test. *$p < 0.05$, **$p < 0.01$. n.s., not significant. Representative of three (**a**, **b**, **c**, **d**, **e**, **f**) or two (**i**) independent experiments.

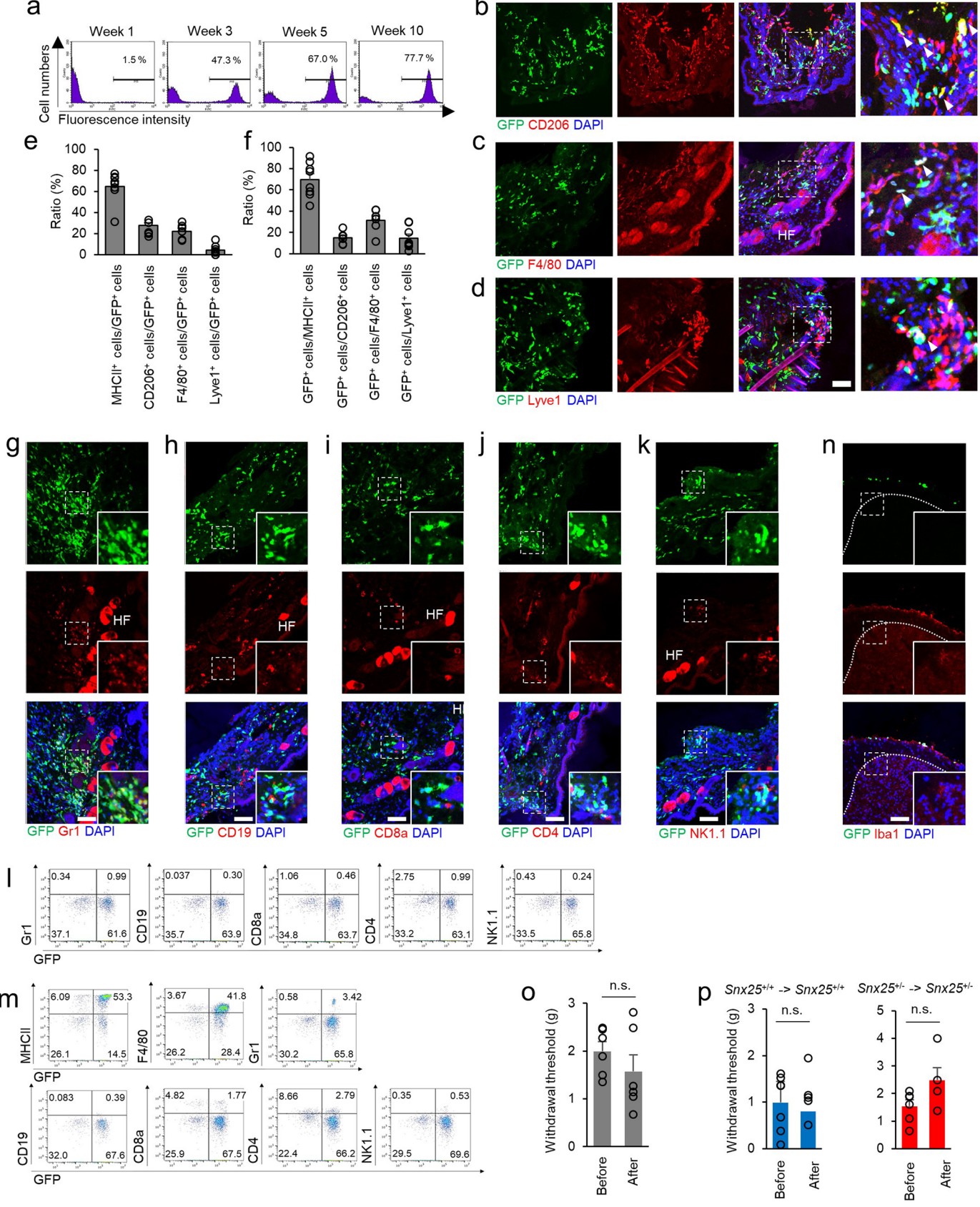

**Extended Data Fig. 6 | See next page for caption.**

**Extended Data Fig. 6 | Dermal macrophages in the skin are derived from bone marrow.** (**a**) Time course (1–10 weeks after BMT) of myeloid cell chimerism in peripheral blood of WT mice received BMT from GFP mice. The proportions of GFP⁺ cells (positive ranges were indicated by bars) in total myeloid cells are expressed as percentages (n = 4). (**b–d**) Confocal microscopic images of the hind paw skin of WT mice that received BMT from GFP mice. Sections are double labeled for GFP and cell markers (CD206 (**b**), F4/80 (**c**), and Lyve1(**d**)). HF denotes hair follicles showing non-specific fluorescence. (Scale bar, 100 μm. (**e**) Percentage of MHCII⁺ (64.6%), CD206⁺ (27.7%), F4/80⁺ (22.3%), and Lyve1⁺ (4.3%) cells among GFP⁺ cells (5 weeks after transplantation) (MHCII: n = 8; CD206: n = 6; F4/80: n = 6; Lyve1: n = 8 hind paw skin sections from at least 3 different mice). (**f**) Percentage of GFP⁺ cells among MHCII⁺, CD206⁺, F4/80⁺, and Lyve1⁺ cells (5 weeks after transplantation) (MHCII: n = 8; CD206: n = 6; F4/80: n = 6; Lyve1: n = 8 hind paw skin sections from at least 3 different mice). Ratios are 70.0%, 15.1%, 31.5%, and 14.5%, respectively. (**g–k**) Confocal microscopic images of the sections stained for Gr1 (**g**), CD19 (**h**), CD8a (**i**), CD4 (**j**), and NK1.1 (**k**) of the hind paw skin of WT mice that received BMT from GFP mice. Each antibody was used as a marker for neutrophils, B cells, killer T cells, helper T cells, and NK cells, respectively. HF; hair follicles with non-specific fluorescence. Scale bars, 100 μm. (**l**) Flow cytometry strategy using Gr1, CD19, CD8a, CD4, and NK1.1 marker expression in the hind paw skin. (**m**) Flow cytometry strategy using MHCII, F4/80, Gr1, CD19, CD8a, CD4, and NK1.1 marker expression in the back skin. (**n**) Confocal microscopic images of the dorsal horn of the spinal cord (L4) of WT mice that received BMT from GFP mice. The gray matter is demarcated by the dotted line. Insets show magnified images of the boxed areas. Note the absence of GFP⁺ cells in the gray matter. Scale bar, 100 μm. (**o**) VF thresholds in WT mice before and after busulfan treatment (n = 6, p = 0.328). g, gram. (**p**) VF thresholds before and after BMT (WT, before BMT: n = 8; after BMT: n = 5, p = 0.748; Snx25⁺/⁻ mice, before BMT: n = 6; after BMT: n = 4, p = 0.108) in the WT and Snx25⁺/⁻ mice that received BMT from the mice of the same genotype. g, gram. Results are represented as mean ± SEM. Statistical analyses were performed using the two-tailed Student's t-test. n.s., not significant. Representative of three independent experiments (**b-d**, **g-k**, and **n**).

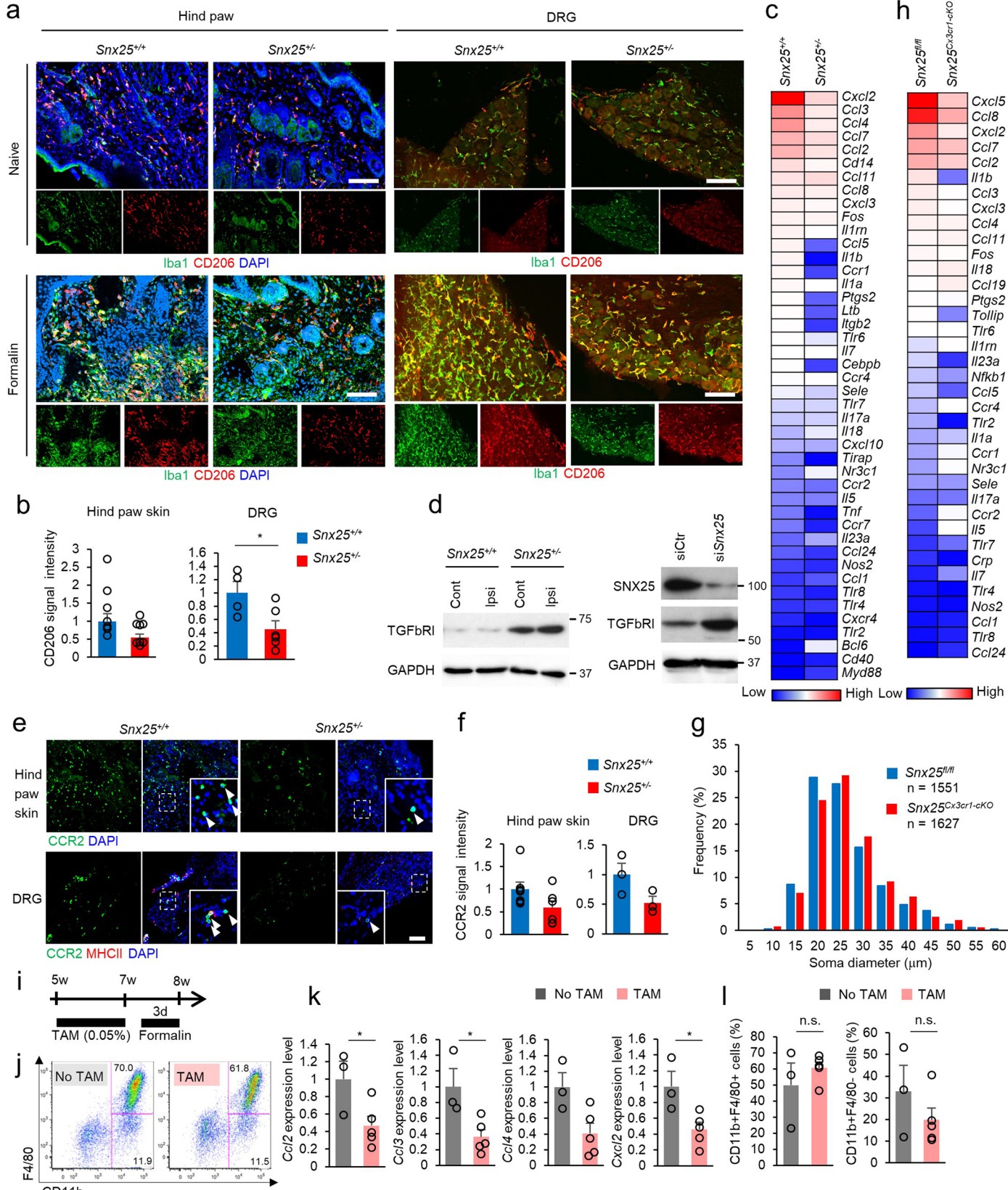

**Extended Data Fig. 7 | See next page for caption.**

**Extended Data Fig. 7 | Impairment of macrophage function in *Snx25*[+/−] mice.** (**a**) Confocal images of the hind paw skin (naïve and 7 days after formalin injection) and DRG (L4) (naïve and 7 days after formalin injection) immunolabeled for Iba1 (green) and CD206 (red) in WT and *Snx25*[+/−] mice. Scale bars, 100 μm. (**b**) Quantification of mean CD206 fluorescence intensity at 7 days after formalin injection in the hind paw skin and DRG of WT and *Snx25*[+/−] mice. Hind paw skin (WT: n = 8; *Snx25*[+/−]: n = 8, $p = 0.061$) or DRG (WT: n = 4; *Snx25*[+/−]: n = 6, $p = 0.028$) sections from at least 3 different mice. Note the reduced number of the DRG macrophages in the *Snx25*[+/−] mice that may play an additive role to that of dermal macrophages in dull response in the inflammation paradigm. (**c**) Immune-related genes were examined with a mini-microarray (QIAGEN, RT2 Profiler PCR Array, PAMM-077ZC). Relative gene expression patterns of hind paw skin at 3 days after formalin injection in the WT and *Snx25*[+/−] mice are color-coded (n = 3 mice per group). (**d**) Left, a representative immunoblot showing TGFβ receptor type I (TGFbRI) levels in the ipsilateral (Ipsi) injected side and the contralateral (contra) side of the hind paw skin of WT and *Snx25*[+/−] mice at 30 min after formalin injection. Right, a representative immunoblot showing TGFbRI levels in the macrophage cell line RAW264.7 treated with scramble siRNA (siCtr) or *Snx25* siRNA (si*Snx25*). Note that TGFbRI is upregulated in the *Snx25*-decreased tissues and cells. (**e**) Confocal microscopic images of the hind paw skin and DRG (L4) (7 days after formalin injection) immunolabeled for CCR2 (green) in WT and *Snx25*[+/−] mice. Insets show magnified views of boxed areas and arrowheads indicate CCR2[+] cells. Scale bar, 100 μm. (**f**) Quantification of mean CCR2 fluorescence intensity at 7 days after formalin injection in the hind paw skin and DRG of WT and *Snx25*[+/−] mice. Hind paw skin (WT: n = 7; *Snx25*[+/−]: n = 6, $p = 0.09$) or DRG (WT: n = 3; *Snx25*[+/−]: n = 3, $p = 0.099$) sections from at least 3 different mice. (**g**) Size distribution of DRG neuron diameters (L4) of *Snx25*[fl/fl] mice (TAM, 1551 cells from 3 mice, blue columns) and *Snx25*[Cx3cr1-cKO] mice (TAM, 1627 cells from 3 mice, red columns) are plotted. X-axis values indicated the maximum diameter in each 5-μm range. (**h**) Immune-related genes were examined with a mini-microarray as in (**c**). Relative gene expression patterns of hind paw skin at 3 days after formalin injection in *Snx25*[Cx3cr1-cKO] mice (comparison with *Snx25*[fl/fl] mice as a control) are color-coded (n = 3 mice per group). (**i**) Experimental schedule of flow cytometry using *Snx25*[Cx3cr1-cKO] mice. (**j**) Flow cytometry strategy using 7-AAD, CD45, F4/80, and CD11b marker expression. F4/80[+]/CD11b[+] cells were collected as macrophages in the hind paw skin of *Snx25*[Cx3cr1-cKO] mice after 3 days of formalin injection. (**k**) Expression patterns of representative chemokines (No TAM: n = 3; TAM: n = 5). *Ccl2*, $p = 0.045$; *Ccl3*, $p = 0.023$; *Ccl4*, $p = 0.083$; *Cxcl2*, $p = 0.034$. (**l**) Proportion of myeloid population in the hind paw skin of *Snx25*[Cx3cr1-cKO] mice (No TAM: n = 3; TAM: n = 3). CD11b[+] F4/80[+], $p = 0.0378$; CD11b[+]F4/80[−], $p = 0.029$. Results are represented as mean ± SEM. Significance was calculated using the two-tailed Student's *t*-test. *$p < 0.05$, **$p < 0.01$. Representative of three independent experiments (**a** and **e**).

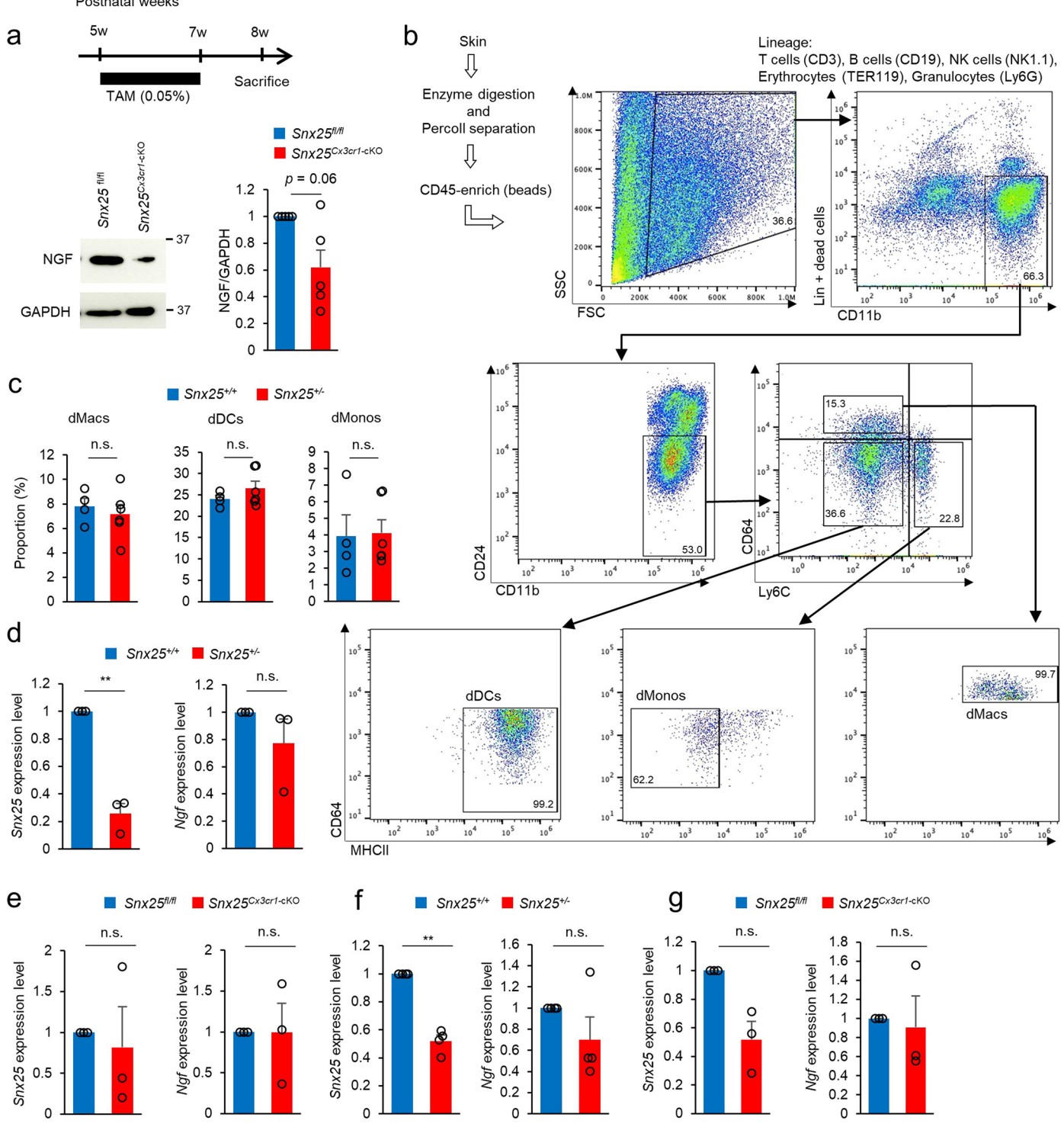

**Extended Data Fig. 8 | SNX25 in dermal macrophages regulates *Ngf* level. (a)** A representative immunoblot showing NGF levels in the hind paw skin of *Snx25*[fl/fl] mice and *Snx25*[Cx3cr1-cKO] mice after two-week TAM treatment (schedule was shown in upper panel). Semi-quantitative analyses of the NGF levels are shown (n = 5, *p* = 0.059). **(b)** Flow cytometry strategy to sort dMacs, dMonos and dDCs from the back skin of mice using CD11b, CD24, Ly6C, CD64, MHCII, and lineage (CD3, CD19, NK1.1, TER119, Ly6G) marker expression. **(c)** Proportion of myeloid population (dMacs, dMonos, and dDCs) (% each cell / CD45[+] CD11b[+] Lin[-] ×100) in the back skin of WT or *Snx25*[+/−] mice (WT: n = 4; *Snx25*[+/−]: n = 6). dMacs, *p* = 0.577; dMonos, *p* = 0.289; dDCs, *p* = 0.898. **(d)** Expression patterns of *Snx25* and *Ngf* in dMonos of

WT and *Snx25*[+/−] mice (WT: n = 3; *Snx25*[+/−]: n = 3). *Snx25*, *p* = 0.009; *Ngf*, *p* = 0.328. **(e)** Expression patterns of *Snx25* and *Ngf* in dMonos of *Snx25*[fl/fl] and *Snx25*[Cx3cr1-cKO] mice (*Snx25*[fl/fl]: n = 3; *Snx25*[Cx3cr1-cKO]: n = 3). *Snx25*, *p* = 0.748; *Ngf*, *p* = 0.988. **(f)** Expression patterns of *Snx25* and *Ngf* in dDCs of WT and *Snx25*[+/−] mice (WT: n = 4; *Snx25*[+/−]: n = 4). *Snx25*, *p* = 0.001; *Ngf*, *p* = 0.256. **(g)** Expression patterns of *Snx25* and *Ngf* in dDCs of *Snx25*[fl/fl] and *Snx25*[Cx3cr1-cKO] mice. (*Snx25*[fl/fl]: n = 3; *Snx25*[Cx3cr1-cKO]: n = 3). *Snx25*, *p* = 0.062; *Ngf*, *p* = 0.804. Results are represented as mean ± SEM. Statistical analyses were performed using the two-tailed Student's *t*-test **(c)**, Welch's *t*-test **(a, d, e, f,** and **g)**. *\*p* < 0.05, \*\**p* < 0.01. n.s., not significant.

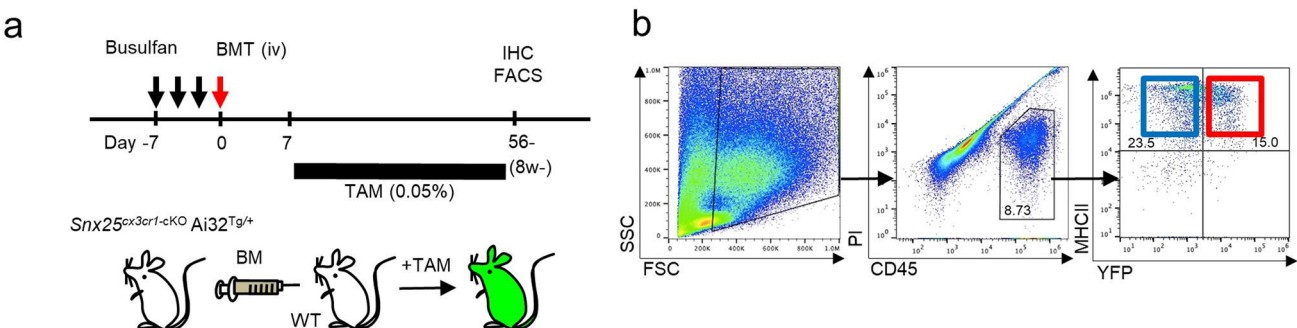

**Extended Data Fig. 9 | SNX25 in dMacs replenished from bone marrow regulates *Ngf* level.** (**a**) Schedules for generation of BM chimeric mice by transplanting *Snx25*[Cx3cr1-cKO] Ai32[Tg/+] BM into WT mice, and subsequent TAM feeding. (**b**) Flow cytometry strategy to sort YFP+ and YFP- dMacs using PI, CD45, YFP, MHCII marker expression.

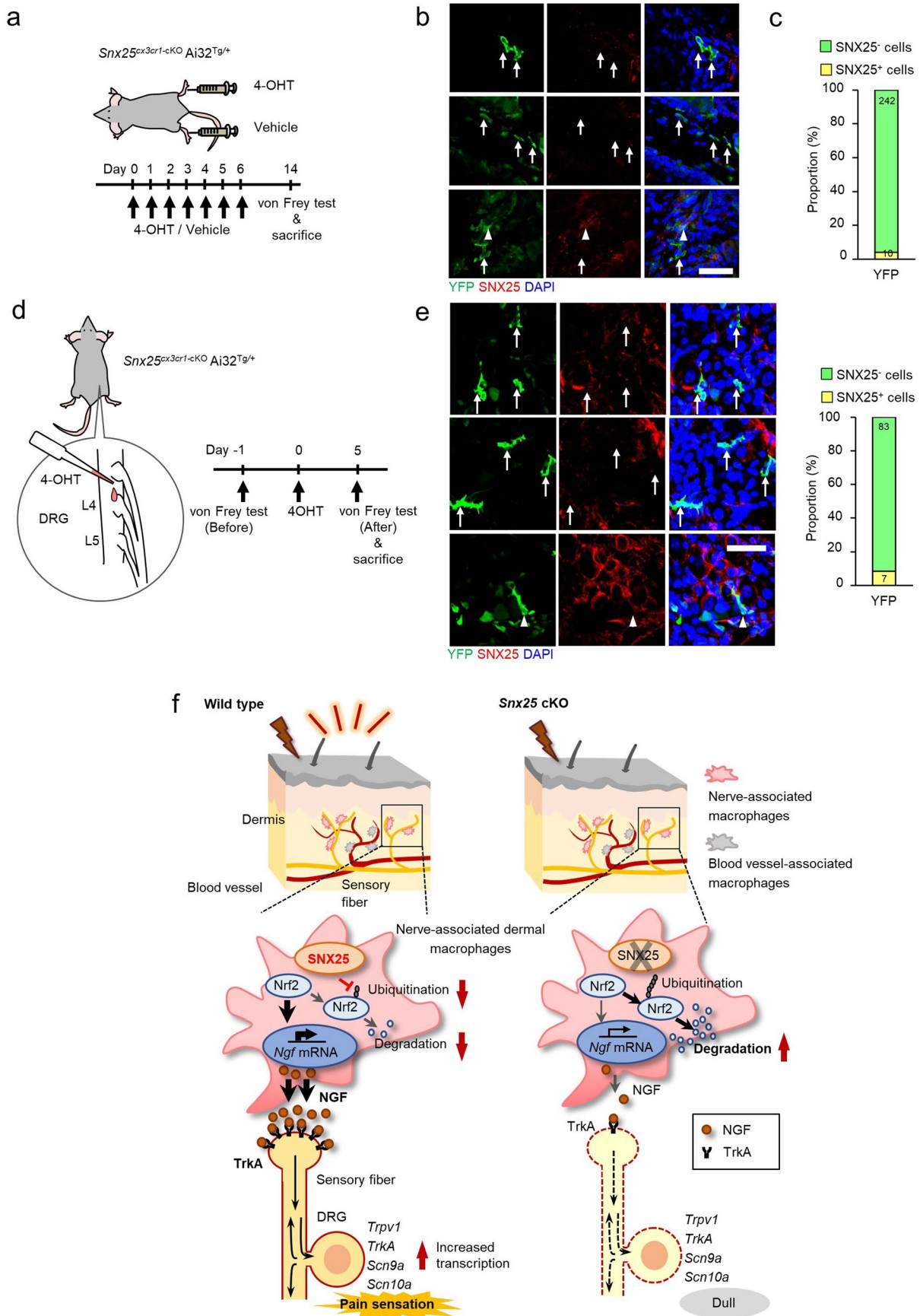

**Extended Data Fig. 10 | See next page for caption.**

**Extended Data Fig. 10 | Efficiency of SNX25 deletion in *Snx25Cx3cr1*<sup>-cKO</sup>Ai32<sup>Tg/+</sup> mice.** (**a**) Scheme depicting dermal injection of 4-OHT into the right hind paw and of vehicle into the left hind paws of a *Snx25*<sup>Cx3cr1-cKO</sup>Ai32<sup>Tg/+</sup> mouse and experimental schedule. (**b**) Confocal microscopic images of the hind paw skin immunolabeled for YFP and SNX25 in *Snx25*<sup>Cx3cr1-cKO</sup>Ai32<sup>Tg/+</sup> mice that received 4-OHT injection daily for 7 days. Arrows denote SNX25⁻ cells and arrowheads denote SNX25⁺ cells in YFP-expressing cells. Scale bar, 50 μm. (**c**) Proportion of SNX25⁻ cells (green column) in YFP⁺ cells. Note that 96.0% of YFP⁺ cells were SNX25⁻. Numbers inside columns were the actual numbers of cells counted. n = 30 hind paw skin sections from 3 different mice. (**d**) Scheme depicting injection of 4-OHT into surgically-exposed L4 DRG of *Snx25*<sup>Cx3cr1-cKO</sup>Ai32<sup>Tg/+</sup> mice and experimental schedule. (**e**) Confocal microscopic images of DRG double labeled for YFP and SNX25 in the mice indicated in **d**. Scale bar, 50 μm. The summary graph shows the proportion of SNX25⁻ cells (green column) in YFP⁺ cells. Note that 91.2% of YFP⁺ cells were SNX25⁻. Numbers inside columns are the actual numbers of cells counted. n = 16 sections from 3 different mice. (**f**) Schematic representation of how SNX25 in dMacs set pain sensitivity via Nrf2–NGF/TrkA signaling and of how *Snx25* cKO resulted in a dull phenotype. Both *Snx25*<sup>+/−</sup> and *Snx25*<sup>Cx3cr1-cKO</sup> mice display reduced pain responses under both normal and painful conditions. SNX25 inhibits the ubiquitination and subsequent proteasome degradation of Nrf2 and thereby maintains NGF production and secretion into tissues. *Snx25* cKO, in turn, accelerates Nrf2 degradation and lowers NGF levels in dermis, which results in a dull phenotype. Representative of three independent experiments (**b** and **e**).

Akio Wanaka

# Reporting Summary

## Statistics

For all statistical analyses, confirm that the following items are present in the figure legend, table legend, main text, or Methods section.

| n/a | Confirmed | |
|---|---|---|
| ☐ | ☒ | The exact sample size ($n$) for each experimental group/condition, given as a discrete number and unit of measurement |
| ☐ | ☒ | A statement on whether measurements were taken from distinct samples or whether the same sample was measured repeatedly |
| ☐ | ☒ | The statistical test(s) used AND whether they are one- or two-sided *Only common tests should be described solely by name; describe more complex techniques in the Methods section.* |
| ☐ | ☒ | A description of all covariates tested |
| ☐ | ☒ | A description of any assumptions or corrections, such as tests of normality and adjustment for multiple comparisons |
| ☐ | ☒ | A full description of the statistical parameters including central tendency (e.g. means) or other basic estimates (e.g. regression coefficient) AND variation (e.g. standard deviation) or associated estimates of uncertainty (e.g. confidence intervals) |
| ☐ | ☒ | For null hypothesis testing, the test statistic (e.g. $F$, $t$, $r$) with confidence intervals, effect sizes, degrees of freedom and $P$ value noted *Give P values as exact values whenever suitable.* |
| ☒ | ☐ | For Bayesian analysis, information on the choice of priors and Markov chain Monte Carlo settings |
| ☒ | ☐ | For hierarchical and complex designs, identification of the appropriate level for tests and full reporting of outcomes |
| ☒ | ☐ | Estimates of effect sizes (e.g. Cohen's $d$, Pearson's $r$), indicating how they were calculated |

*Our web collection on statistics for biologists contains articles on many of the points above.*

## Software and code

Policy information about availability of computer code

| Data collection | Images of micrographs were captured using a confocal laser scanning microscope (C2 ver4.10, Nikon). Immunostained cell were collected using FACSAria by BD FACSDiva 4.1, or SONY cell sorter SH 800 series by software ver2.1 . |
|---|---|
| Data analysis | Adobe Photoshop CC 2018, ImageJ, Statcel-the Useful Addin Forms on Excel-4th ed, FlowJo software (V10), IBM SPSS (V23). |

For manuscripts utilizing custom algorithms or software that are central to the research but not yet described in published literature, software must be made available to editors and reviewers. We strongly encourage code deposition in a community repository (e.g. GitHub). See the Nature Portfolio guidelines for submitting code & software for further information.

## Data

Policy information about availability of data

All manuscripts must include a data availability statement. This statement should provide the following information, where applicable:
- Accession codes, unique identifiers, or web links for publicly available datasets
- A description of any restrictions on data availability
- For clinical datasets or third party data, please ensure that the statement adheres to our policy

All data supporting the findings of this study are found within the manuscript and its Supplementary information, and are available from the corresponding author upon reasonable request. Source data are provided with this paper.

# Field-specific reporting

Please select the one below that is the best fit for your research. If you are not sure, read the appropriate sections before making your selection.

☒ Life sciences ☐ Behavioural & social sciences ☐ Ecological, evolutionary & environmental sciences

For a reference copy of the document with all sections, see nature.com/documents/nr-reporting-summary-flat.pdf

# Life sciences study design

All studies must disclose on these points even when the disclosure is negative.

| | |
|---|---|
| Sample size | No statistical methods were used to predetermine sample size, but our sample sizes are similar to those reported in previous publications. A minimum of three biologically independent samples were tested, and experiments performed in at least 2 independent instances (mostly 3) with similar results. Sample size was based on pilot experiments and previous experience with similar experiments in the laboratory. Data from individual experiments were pooled to achieve power. |
| Data exclusions | No data was excluded. |
| Replication | Behavioral experiments on mice were conducted with a sufficient number of animals (6 to 33) to take individual differences into account. Experiments were conducted at least five times. All experiments were performed in multiple independent experiments as indicated in the figure legends. All attempts for littermates were successfully performed by multiple investigators using independent litter of mice. |
| Randomization | Transgenic mice were predetermined by mouse genotype and therefore could not be randomized. Littermates were assigned into the control or knockout groups after genotyping. In all the behavioral tests, however, examiners were always blind to the genotypes of mice, the kinds of treatments, and the sides of hind paws that received injections. After the evaluation was done, the behavioral data were analyzed by a different researcher. C57BL/6J mice were randomly assigned to treatment and control groups. All mice were age-matched. |
| Blinding | Experimentors was blinded to the identify of mice being analyzed in behavioral tests. For experiments other than behavioral tests, experimentors were not blinded. Blinding was not possible as predominately one person was responsible for performing each experiment and carrying out data analysis. Blinding was not relevant because the quantification of signal intensity in Immunoblot or images, and analysis of gene expression level using thermal cycler were performed under the same conditions as in the control group, respectively. |

# Reporting for specific materials, systems and methods

We require information from authors about some types of materials, experimental systems and methods used in many studies. Here, indicate whether each material, system or method listed is relevant to your study. If you are not sure if a list item applies to your research, read the appropriate section before selecting a response.

## Materials & experimental systems

| n/a | Involved in the study |
|---|---|
| ☐ | ☒ Antibodies |
| ☐ | ☒ Eukaryotic cell lines |
| ☒ | ☐ Palaeontology and archaeology |
| ☐ | ☒ Animals and other organisms |
| ☒ | ☐ Human research participants |
| ☒ | ☐ Clinical data |
| ☒ | ☐ Dual use research of concern |

## Methods

| n/a | Involved in the study |
|---|---|
| ☒ | ☐ ChIP-seq |
| ☐ | ☒ Flow cytometry |
| ☒ | ☐ MRI-based neuroimaging |

# Antibodies

| | |
|---|---|
| Antibodies used | Mouse anti-NF200 (N0142, Sigma-Aldrich, clone N52) 1:1000, AB_477257<br>Mouse anti-PGP9.5 (ab8189, Abcam, clone 13C4 / I3C4) 1:500, AB_306343<br>Mouse anti-NeuN (MAB377, Millipore, clone A60) 1:150, AB_2298772<br>Mouse anti-GFAP (MAB360, Millipore, clone GA5) 1:500, AB_11212597<br>Rabbit anti-c-Fos (226003, Synaptic Systems) 1:10000, AB_2231974<br>Rabbit anti-TRPV1 (KM018, Trans Genic) 1:100, AB_1627247<br>Rabbit anti-TrkA (ab76291, Abcam) 1:150-1:1000, AB_1524514<br>Rabbit anti-Iba1 (019-19741, Wako) 1:500, AB_839504<br>Rabbit anti-F4/80 (28463-1-AP, Proteintech) 1:2000, AB_2881149<br>Rabbit anti-NGF (sc-548, Santa Cruz Biotechnology) 1:200-1:1000, AB_632011<br>Rabbit anti-GFP (A6455, Thermo Fisher Scientific) 1:5000, AB_221570<br>Rabbit anti-SNX25 (13294-1-AP, Proteintech) 1:500, AB_2192549<br>Rabbit anti-Nrf2 (sc-722, Santa Cruz Biotechnology) 1:500, AB_2108502<br>Rabbit anti-HO-1 (ADI-SPA-896, Enzo Life Sciences,) 1:500, AB_10614948 |

Rabbit anti-TGFbRI (sc-398, Santa Cruz Biotechnology) 1:200, AB_632493
Rabbit anti-GAPDH (ABS16, Merk Millipore) 1:2000, AB_10806772
Rat anti-MHCII (NBP1-43312, Novus Biologicals) 1:100, AB_10006677
Rat anti-F4/80 (NB600-404, Novus Biologicals) 1:500, AB_10003219
Rat anti-CD117 (MAB1356, R&D Systems) 1:100, AB_2131131
Rat anti-CCR2 (NBP1-48337, Novus Biologicals) 1:200, AB_10011101
Rat anti-GFP (04404-84, Nacalai Tesque) 1:5000, AB_10013361
Goat anti-CD206 (AF2535, R&D Systems) 1:500-1:1000, AB_2063012
Alexa Fluor 488–anti-CD11b (101219, BioLegend, clone, M1/70) 1:100, AB_493545
Alexa Fluor 647-anti-F4/80 (123121, Biolegend, clone, BM8) 1:100, AB_893480
APC anti-CD64 (139306, BioLegend, clone, X54-5/7.1) 1:25, AB_11219391
APC/Cyanine7 Ly6C (128025, BioLegend, clone HK1.4) 1:100, AB_10643867
Biotin anti-CD4 (100403, BioLegend, clone GK1.5) 1:100-1:200, AB_312688
Biotin anti-CD8a (100703, BioLegend, clone 53-6.7) 1:100-1:200, AB_312742
Biotin anti-Ly6G/Ly6C (108403, BioLegend, clone Gr1) 1:100-1:200, AB_313368
Biotin anti-NK1.1 (108703, BioLegend, clone PK136) 1:100-1:200, AB_313390
Biotin-anti-CD19 (115504, BioLegend, clone 6D5) 1:100, AB_313639
Biotin anti-CD19 (13-0193-81, eBioscience, clone 1D3) 1:100-1:200, AB_657657
Biotin anti-MHCII (107603, BioLegend, clone M5/114.15.2) 1:100, AB_313318
Biotin anti-F4/80 (123105, BioLegend, clone BM8) 1:100-1:500, AB_893499
Biotin anti-CD3 (100243, BioLegend, clone 17A2) 1:100, AB_2563946
Biotin anti-TER119 (116203, BioLegend, clone TER-119) 1:100, AB_313704
Biotin anti-Ly6G (127603, BioLegend, clone 1A8) 1:100, AB_1186105
FITC–anti-MHCII (107605, BioLegend, clone M5/114.15.2) 1:100, AB_313320
PE/Cyanine7 anti-CD45 (103113, BioLegend, clone 30-F11) 1:100, AB_312979
PE anti-CD11b (101207, BioLegend, clone M1/70) 1:100, AB_312790
PE anti-CD45 (103106, BioLegend, clone 30-F11) 1:100, AB_312971
BV421 anti-mouse CD45 (103131, BioLegend, clone 30-F11) 1:00, AB_10899570
Brilliant violet 421 anti-CD24 (101825, BioLegend, clone M1/69) 1:100, AB_10901159
PerCP-Cyanine5.5 streptavidin (405214, BioLegend) 1:200, AB_2716577
APC-Streptavidin (405207, BioLegend) 1:200
TruStain FcX™ PLUS (CD16/32 antibody, 156604, BioLegend, clone S17011E) 1:100, AB_2783138

Validation

All antibodies used are commercially available as described in the manuscript. We selected antibody clones that have been extensively used in the literature. We also titrated all antibodies prior to experiments.

Mouse anti-NF200 (N0142, Sigma-Aldrich, clone N52)
https://www.sigmaaldrich.com/deepweb/assets/sigmaaldrich/product/documents/355/551/n0142dat.pdf

Mouse anti-PGP9.5 (ab8189, Abcam, clone 13C4 / I3C4)
https://www.abcam.co.jp/pgp95-antibody-13c4--i3c4-ab8189.html

Mouse anti-NeuN (MAB377, Millipore, clone A60)
https://www.merckmillipore.com/JP/ja/product/Anti-NeuN-Antibody-clone-A60,MM_NF-MAB377

Mouse anti-GFAP (MAB360, Millipore, clone GA5)
https://www.merckmillipore.com/JP/ja/product/Anti-Glial-Fibrillary-Acidic-Protein-Antibody-clone-GA5,MM_NF-MAB360

Rabbit anti-c-Fos (226003, Synaptic Systems)
https://sysy.com/product/226003

Rabbit anti-TRPV1 (KM018, Trans Genic)
https://www.sceti.co.jp/images/upload/export/1013_KM018_p.pdf

Rabbit anti-TrkA (ab76291, Abcam)
https://www.abcam.co.jp/pan-trk-antibody-ep1058y-ab76291.html

Rabbit anti-Iba1 (019-19741, Wako)
https://labchem-wako.fujifilm.com/us/product/detail/W01W0101-1974.html

Rabbit anti-F4/80 (28463-1-AP, Proteintech)
https://www.ptglab.co.jp/products/F4-80-Antibody-28463-1-AP.htm

Rabbit anti-NGF (sc-548, Santa Cruz Biotechnology)
https://datasheets.scbt.com/sc-548.pdf

Rabbit anti-GFP (A6455, Thermo Fisher Scientific)
https://www.thermofisher.com/antibody/product/GFP-Antibody-Polyclonal/A-6455

Rabbit anti-SNX25 (13294-1-AP, Proteintech)
https://www.ptglab.co.jp/products/SNX25-Antibody-13294-1-AP.htm

Rabbit anti-Nrf2 (sc-722, Santa Cruz Biotechnology)
https://datasheets.scbt.com/sc-722.pdf

Rabbit anti-HO-1 (ADI-SPA-896, Enzo Life Sciences,)
https://www.enzolifesciences.com/ADI-SPA-896/ho-1-polyclonal-antibody/

Rabbit anti-TGFbRI (sc-398, Santa Cruz Biotechnology)
https://datasheets.scbt.com/sc-398.pdf

Rabbit anti-GAPDH (ABS16, Merk Millipore)
https://www.merckmillipore.com/JP/ja/product/Anti-GAPDH-Antibody,MM_NF-ABS16

Rat anti-MHCII (NBP1-43312, Novus Biologicals)
https://www.novusbio.com/products/mhc-class-ii-i-a-i-e-antibody-m5-114152_nbp1-43312

Rat anti-F4/80 (NB600-404, Novus Biologicals)
https://www.novusbio.com/products/f4-80-antibody-ci-a3-1_nb600-404

Rat anti-CD117 (MAB1356, R&D Systems)
https://www.rndsystems.com/products/mouse-cd117-c-kit-antibody-180627_mab1356

Rat anti-CCR2 (NBP1-48337, Novus Biologicals)
https://www.novusbio.com/products/ccr2-antibody_nbp1-48337

Rat anti-GFP (04404-84, Nacalai Tesque)
https://www.nacalai.co.jp/ss/ec2/EC-srchdetl.cfm?jump=EC-srchdetl&syohin=0440484&syubetsu=3

Goat anti-CD206 (AF2535, R&D Systems)
https://www.rndsystems.com/products/mouse-mmr-cd206-antibody_af2535

Alexa Fluor 488–anti-CD11b (101219, BioLegend, clone, M1/70)
https://www.biolegend.com/ja-jp/clone-search/alexa-fluor-488-anti-mouse-human-cd11b-antibody-2700

Alexa Fluor 647-anti-F4/80 (123121, Biolegend, clone, BM8)
https://www.biolegend.com/ja-jp/sean-tuckers-tests/alexa-fluor-647-anti-mouse-f4-80-antibody-4074

APC anti-CD64 (139306, BioLegend, clone, X54-5/7.1)
https://www.biolegend.com/ja-jp/clone-search/apc-anti-mouse-cd64-fcgammari-antibody-7874

APC/Cyanine7 Ly6C (128025, BioLegend, clone HK1.4)
https://www.biolegend.com/ja-jp/explore-new-products/apc-cyanine7-anti-mouse-ly-6c-antibody-6758?GroupID=BLG5853

Biotin anti-CD4 (100403, BioLegend, clone GK1.5)
https://www.biolegend.com/ja-jp/products/biotin-anti-mouse-cd4-antibody-247

Biotin anti-CD8a (100703, BioLegend, clone 53-6.7)
https://www.biolegend.com/ja-jp/productstab/biotin-anti-mouse-cd8a-antibody-152

Biotin anti-Ly6G/Ly6C (108403, BioLegend, clone Gr1)
https://www.biolegend.com/ja-jp/neuroscience-1/biotin-anti-mouse-ly-6g-ly-6c-gr-1-antibody-457?GroupID=BLG4876

Biotin anti-NK1.1 (108703, BioLegend, clone PK136)
https://www.biolegend.com/ja-jp/productstab/biotin-anti-mouse-nk-1-1-antibody-428

Biotin-anti-CD19 (115504, BioLegend, clone 6D5)
https://www.biolegend.com/ja-jp/products/biotin-anti-mouse-cd19-antibody-1527

Biotin anti-CD19 (13-0193-81, eBioscience, clone 1D3)
https://www.thermofisher.com/antibody/product/CD19-Antibody-clone-eBio1D3-1D3-Monoclonal/16-0193-81

Biotin anti-MHCII (107603, BioLegend, clone M5/114.15.2)
https://www.biolegend.com/fr-lu/products/biotin-anti-mouse-i-a-i-e-antibody-365?
pdf=true&displayInline=true&leftRightMargin=15&topBottomMargin=15&filename=Biotin%20anti-mouse%20I-A/I-E%
20Antibody.pdf

Biotin anti-F4/80 (123105, BioLegend, clone BM8)
https://www.biolegend.com/ja-jp/products/biotin-anti-mouse-f4-80-antibody-4066?GroupID=BLG5319

Biotin anti-CD3 (100243, BioLegend, clone 17A2)
https://production.biolegend.com/ja-jp/products/biotin-anti-mouse-cd3-antibody-10023

Biotin anti-TER119 (116203, BioLegend, clone TER-119)
https://www.biolegend.com/ja-jp/productstab/biotin-anti-mouse-ter-119-erythroid-cells-antibody-1864?GroupID=ImportedGROUP1

Biotin anti-Ly6G (127603, BioLegend, clone 1A8)
https://www.biolegend.com/ja-jp/products/biotin-anti-mouse-ly-6g-antibody-4772

FITC–anti-MHCII (107605, BioLegend, clone M5/114.15.2)
https://www.biolegend.com/ja-jp/clone-search/fitc-anti-mouse-i-a-i-e-antibody-366?GroupID=BLG11931

PE/Cyanine7 anti-CD45 (103113, BioLegend, clone 30-F11)
https://www.biolegend.com/ja-jp/products/pe-cyanine7-anti-mouse-cd45-antibody-1903

PE anti-CD11b (101207, BioLegend, clone M1/70)
https://www.biolegend.com/ja-jp/products/pe-anti-mouse-human-cd11b-antibody-349

PE anti-CD45 (103106, BioLegend, clone 30-F11)
https://www.biolegend.com/ja-jp/explore-new-products/pe-anti-mouse-cd45-antibody-100

BV421 anti-mouse CD45 (103131, BioLegend, clone 30-F11)
https://www.biolegend.com/ja-jp/explore-new-products/brilliant-violet-421-anti-mouse-cd45-antibody-7253

Brilliant violet 421 anti-CD24 (101825, BioLegend, clone M1/69)
https://www.biolegend.com/ja-jp/explore-new-products/brilliant-violet-421-anti-mouse-cd24-antibody-7323

PerCP-Cyanine5.5 streptavidin (405214, BioLegend)
https://www.biolegend.com/ja-jp/products/percp-cyanine5-5-streptavidin-4212
AAPC-Streptavidin (405207, BioLegend)
https://www.biolegend.com/ja-jp/explore-new-products/apc-streptavidin-1470

TruStain FcX™ PLUS (CD16/32 antibody, 156604, BioLegend, clone S17011E)
https://www.biolegend.com/en-us/punchout/punchout-products/trustain-fcx-plus-anti-mouse-cd16-32-antibody-17085?
GroupID=GROUP20

# Eukaryotic cell lines

Policy information about cell lines

| | |
|---|---|
| Cell line source(s) | 293T (RRID:CVCL_0063), RAW264.7 (RRID:CVCL_0493) |
| Authentication | 293T cells (ECACC 12022001), RAW264.7 (ECACC 91062702) |
| Mycoplasma contamination | Cell-lines were not tested for mycoplasma contamination. |
| Commonly misidentified lines (See ICLAC register) | N/A |

# Animals and other organisms

Policy information about studies involving animals; ARRIVE guidelines recommended for reporting animal research

| | |
|---|---|
| Laboratory animals | C57BL/6J, RRID:IMSR_JAX:000664

Mlc1Tgmice (B6; CBB6(129)-Tg(Mlc1-tTA)2Rhn), RBRC05450

Snx25 constitutive KO (Snx25+/-) mice (C57BL/6N-Atm1Brd Snx25tm1a(KOMP)Wtsi/NjuMmucd, strain number, T001400) , RRID:MMRRC_068035-UCD

CAG-Flpo mice (B6.Cg-Tg(CAG-FLPo)/1Osb), RBRC09982

Advillin-Cre mice (B6.Cg-Tg(Avil-Cre/ERT2)AJwo/J) (Jackson Laboratory, Stock No: 032027), RRID:IMSR_JAX:032027

Cx3cr1CreERT2 mice (B6.129P2(C)-Cx3cr1tm2.1(Cre/ERT2)Jung/J) (Jackson Laboratory, Stock No: 020940) , RRID:IMSR_JAX:020940

Ai39 mice, RCL-eNpHR3.0-EYFP, Jackson Laboratory, Stock No: 014539, RRID:IMSR_JAX:014539

Ai32 mice, RCL-ChR2(H134R)/EYFP, Jackson Laboratory, Stock No: 012569, RRID:IMSR_JAX:012569

GFP mice (C57BL/6-Tg (CAG-EGFP)), RBRC00267

All the protocols for the animal experiments were approved by the Animal Care Committee of Nara Medical University in accordance with the policies established in the NIH Guide for the Care and Use of Laboratory Animals. This study was also carried out in compliance with the ARRIVE guidelines (https://arriveguidelines.org/). |
| Wild animals | No wild animals were used in this study. |
| Field-collected samples | No field collected samples were used in this study. |
| Ethics oversight | All the protocols for the animal experiments were approved by the Animal Care Committee of Nara Medical University in accordance with the policies established in the NIH Guide for the Care and Use of Laboratory Animals. This study was also carried out in compliance with the ARRIVE guidelines (https://arriveguidelines.org/). |

# Flow Cytometry

## Plots

Confirm that:

☒ The axis labels state the marker and fluorochrome used (e.g. CD4-FITC).

☒ The axis scales are clearly visible. Include numbers along axes only for bottom left plot of group (a 'group' is an analysis of identical markers).

☒ All plots are contour plots with outliers or pseudocolor plots.

☒ A numerical value for number of cells or percentage (with statistics) is provided.

## Methodology

| | |
|---|---|
| Sample preparation | Skin from mice were collected and dissociated using Multi Tissue Dissociation Kit 1 (Miltenyi Biotec, Germany). |
| Instrument | FACSAria (BD), Cell Sorter SH800 (Sony) |
| Software | Data was collected using BD FACSDiva Software and analyzed using FlowJo software (Tree Star). |
| Cell population abundance | Purity of sorted macrophages was more than 94%. |

Gating strategy

For flow cytometry gating strategy for skin macrophages under inflammatory condition, cells isolated from hind paw skin were stained with CD11b, CD45 and F4/80. Living leukocytes were identified as 7-AAD negative CD45+ cells within medium forward scatter (FSC) and low side scatter (SSC) population. Macrophage compartment was identified as CD11b+F4/80+ cells within living leukocytes.

Dermal macrophages are defined by Fc-g receptor 1 (CD64) expression in lineage (CD3, CD19, Ly6G, NK1.1, TER119, and Langerhans cell marker CD24) negative CD45+ CD11b+ Ly6C- population and subdivided by the expression of MHCII. CD45 positive cells were isolated using magnetic beads from enzymatically digesting saline-perfused mouse skins, and then myeloid populations, including dermal macrophages (CD64+ Ly6C- MHCII+), dermal monocytes (CD64- Ly6C+ MHCII lo), and dermal dendritic cells (CD64- Ly6C- MHCII+) in the lineage- CD11b+ live cells were sorted.

For flow cytometry gating strategy for dermal myeloid cells including macrophages in mixed BM chimeric mice, dermal myeloid cells including macrophages were selectively collected from the skin of the mixed BM chimeric mice by FACS using FVS700, CD11b, CD45.1, CD45.2 and lineage (CD3, CD19, NK1.1, TER119, Ly6G) marker expression.

☒ Tick this box to confirm that a figure exemplifying the gating strategy is provided in the Supplementary Information.

