## [Peer Review File · Nature Immunology]

Peer Review Information

Journal: Nature Immunology

Manuscript Title: Dermal macrophages set pain sensitivity by modulating the amount of tissue NGF through a SNX25–Nrf2 pathway

Corresponding author name(s): Tatsuhide Tanaka, Akio Wanaka

Reviewer Comments & Decisions:

Decision Letter, initial version:
--

Subject: Decision on Nature Immunology submission NI-A32798

Message: 17th Nov 2021

Dear Dr. Tanaka,

Your Article, "Dermal macrophages set pain sensitivity by modulating tissue NGF levels through SNX25–Nrf2 signaling" has now been seen by 3 referees. While we find your work of considerable potential interest, the reviewers have raised substantial concerns that must be addressed. As such, we cannot accept the current version of the manuscript for publication, but would be happy to consider a revised version that addresses these concerns, as long as novelty is not compromised in the interim.

Please revise the manuscript to address all issues raised by the referees. At resubmission, please include a point-by-point "Response to referees" detailing how you have addressed each referee comment (please specify page and figure number where the new data can be found in the revised manuscript). This response will be sent back to the referees along with the revised manuscript.

In addition, please include a revised version of any required reporting checklist. It will be available to referees (and, potentially, statisticians) to aid in their evaluation if the manuscript goes back for peer review. A revised checklist is essential for re-review of the paper.

The Reporting Summary can be found here:
<https://www.nature.com/documents/nr-reporting-summary.pdf>

When submitting the revised version of your manuscript, please pay close attention to our [href="https://www.nature.com/nature-research/editorial-policies/image-integrity">Digital Image Integrity Guidelines. and to the following points below:](https://www.nature.com/nature-research/editorial-policies/image-integrity)

-- that unprocessed scans are clearly labelled and match the gels and western blots

presented in figures.

-- that control panels for gels and western blots are appropriately described as loading on sample processing controls

-- all images in the paper are checked for duplication of panels and for splicing of gel lanes.

[REDACTED]

We hope to receive the revised manuscript within 6 months. If you cannot send it within this time, please let us know. We will be happy to consider your revision so long as nothing similar has been accepted for publication at Nature Immunology or published elsewhere.

Nature Immunology is committed to improving transparency in authorship. As part of our efforts in this direction, we are now requesting that all authors identified as 'corresponding author' on published papers create and link their Open Researcher and Contributor Identifier (ORCID) with their account on the Manuscript Tracking System (MTS), prior to acceptance. ORCID helps the scientific community achieve unambiguous attribution of all scholarly contributions. You can create and link your ORCID from the home page of the MTS by clicking on 'Modify my Springer Nature account'. For more information please visit www.springernature.com/orcid.

Thank you for the opportunity to review your work.

Sincerely,

Ioana Visan, Ph.D.
Senior Editor
Nature Immunology

Tel: 212-726-9207
Fax: 212-696-9752
www.nature.com/ni

Reviewers' Comments:

Reviewer #1:

Remarks to the Author:

A major finding of this paper is the identification of a potential new modulator of nociception, Snx25, and the discovery that Snx25 can regulate NGF production through reduced Nrf2 ubiquitination. A major weakness of the paper is the lack of data connecting these two findings. The authors propose that Snx25 deficiency leads to behavioral changes via reduced NGF signaling to DRG neurons, and reduced sensitization of Trpv1. While the authors show that NGF is reduced in Snx25 knockdown mice, they do not show that NGF can rescue the behavioral abnormalities or if NGF loss, separately from the other downstream effects of Snx25 knockdown including cytokines and chemokines, is sufficient to produce the phenotype.

Major comments:

1. It is unclear why the authors chose to focus on Snx25 and not the other two genes that were deleted in the TG mice (Slc25a4 and Cfap97). Slc25a4 has been implicated in skeletal muscle movement and could explain reduced withdrawal and licking responses.
2. Snx25 het mice have less Trpv1, Trka, as well as Scn9a and Snc10a. Broad changes in gene expression within neurons could explain the phenotype. They also need to rule out any developmental problems in these mice that might affect their behavior.
3. The need to reconcile the fact that the Snx25(-/-) is embryonic lethal with the fact that the original strain described here (TG mice) had null Snx25 expression (Fig S1)?
4. Snx25 het mice have defective response to capsaicin this, clearly shows a neuronal intrinsic role for Snx25. Despite this, the authors conclude that the pain phenotype is due to Snx25 function in macrophages only.
5. Trpv1 sensitization is not required for acute nociception and therefore that mechanism would not explain the behavioral results in naïve mice.
6. The authors also show a reduction in Nav channels which could very well be the actual mechanism by which these mice have reduced sensitivity.
7. Macrophage characterization is insufficient. Dermal macrophages should be facs sorted and investigated for expression of Snx25 and NGF as compared to other myeloid populations. Flowcytometry should be included to show no alterations in myeloid cell proportions in the dermis (monocytes, dermal Macs, DCs etc).
8. The replacement of dermal Macs in the BMT experiments is difficult to confirm just by IHC. Flow cytometry should be performed to show which populations in the skin immune cells is replaced and to what extent. Chimerism in Blood does not mean chimerism in the skin. The BMT transplant experiments missing WT -> WT and Het -> Het control. Het -> Het control is particularly important to see if the pain phenotype gets more pronounced.
9. For the Cx3cr1 cre ER experiments, the authors cite a study where Cx3cr1 hi Macs were shown to interact with dermal Macs. However, the BMT strategy used here has been shown to not replace the Cx3cr1 hi population, since it was embryonic. Therefore, Tamoxifen induced deletion data should be included to show specific and sufficient deletion.
10. NGF expression was only checked in Snx25het mice. What about the expression of NGF expression in Mac specific conditional KO? This is critical piece of evidence for the proposed mechanism.
11. None of the experiments tie together the Snx25 - Nrf2 - Ngf axis. Can authors

overexpress Nrf2 in Snx25 deficient cells and look for NGF expression? Ideally, the ubiquitination site should be mutated.

Other comments:

Figure 2:

1. The authors focus on mechanical allodynia and then go on to look at Trpv1 levels and capsaicin response without showing any heat sensitivity assays, in S1 they show the mice show normal heat sensitivity at 2 months but insensitivity at 6-8months. The authors need to reconcile the in vitro data from figure 1 with the delayed in vivo response.
2. The authors attribute the decreased calcium response to "Trpv1 channel inactivation" but they do not conduct any electrophysiological studies to show that. This could merely be due to reduced Trpv1 expression.
3. 10uM capsaicin is cytotoxic to neurons.

Figure 3:

Authors need to show the withdrawal threshold in WT mice with BM transferred from WT mice, as a control.

Figure 4:

Il1b is reduced in Snx25(+/-) and is known to sensitize nociceptors, why did the authors choose to focus on NGF and not Il1b?

Reviewer #2:

Remarks to the Author:

In this study, the authors identified Snx25 as a pain-modulating gene. Interestingly, by employing Snx25 conditional-mutant mice, the authors demonstrate that monocyte/macrophage-lineage cells are responsible for reduced pain responses. SNX25 enhanced the expression of macrophage-derived NGF via the inhibition of ubiquitin-mediated degradation of Nrf2.

This is an interesting and timely study indicating that in addition to their canonical defence function, dermal macrophages set pain thresholds and sensitivity via NGF production. The experiments have been performed to a high standard and vast majority of the claims are supported by the data. Nevertheless, some additional points should be addressed to improve the manuscript.

Specific points:

- 1) In figure 2k the authors show representative images of Trka expression and conclude that the expression is reduced in Snx25+/- mice when comparing with Snx+/. It would be beneficial to include quantification across several experiments. The same applies to Figure 7c.
- 2) It is not obvious how the qPCR data was calculated and therefore it is difficult to interpret the results. Figure panels or legends should specify what method was used to quantify the expression levels in 2m, 4d, 5f, 6a.
- 3) Bone-marrow mixed chimeras of Snx25+/- and Snx25 +/+ against a WT competitor should be employed to formally demonstrate the cell-autonomous impact of Snx25 on macrophage derived NGF.

4) The authors claim that “most of the present data were obtained from male mice”. Given that sex dimorphism plays such a role in pain research, it would be highly beneficial to specify the sexes of animals used in specific experiments. And to provide data for both males and females, when possible

5) All data that is mentioned in the manuscript as “data not shown” should be included in the supplementary figures.

Reviewer #3:

Remarks to the Author:

The authors found that Snx25-deficient mice display reduced pain behavior. Conditionally deleting Snx25 in Cx3cr1+ cells also led to a pain-insensitive phenotype whereas mice in which Snx25 was specifically deleted in DRG (using AvilCreER mice) had normal pain responses. Using BM-chimeric mice, they further conclude that BM-derived dermal macrophages contribute to pain sensation. In Snx25+/- mice levels of Ngf or Nrf2 were reduced, indicating that Snx25 in macrophages acts via Ngf signaling.

It is an interesting study describing a role for dermal macrophages in pain sensitivity. However, their claim is overinterpreted. Dermal macrophages should be characterized in more detail. It is not entirely clear that they express Ngf and Nrf2 in vivo or that it is reduced in dermal macrophages in Snx25+/- mice. Whether DRG macrophages further contribute to the observed phenotype is also not entirely clear.

Specific comments

They state that a population of macrophages (positive for MHCII, CD206, or F4/80) were closely associated with PGP9.5-positive sensory fibers and were SNX25-immunoreactive (Fig. 3a, Extended Data Fig. 4a). However, Fig. 4a only shows MHCII+ cells, which could also be dendritic cells. A co-staining of MHCII with another macrophage marker and PGP9.5 should be shown.

In Ext. Fig. 4a it appears that all Snx25 expressing cells are macrophages. Is this the case? Or do other cells in the dermis also express Snx25?
What about macrophages in the DRG? Do they express Snx25?

Ext. Fig. 4b, the authors conclude that the numbers of dermal macrophages are not reduced in Snx25+/- mice. Also here, they should co-stain the MHCII+ cells (nerve-associated macrophages) with a macrophage marker and quantify them.
Are numbers and distribution of macrophages in the DRG in Snx25+/- mice different compared to WT mice?

BM chimeric studies: It is not clear why they perform BM transplantation studies. The ontogeny of dermal macrophages and their partial replacement by BM monocytes over time has been reported (for example Liu et al. 2019). In contrast to fate-mapping studies, in a BM-transplant setting, most macrophage populations are replaced by donor cells with the exception of a few populations (for example Langerhans cells or microglia).
Are DRG macrophages replaced by donor-derived cells? Do they express Cx3cr1, Snx25, Nrf2, Ngf?

In addition, the images shown are not sufficient to define the replacement of dermal macrophages by GFP+ donor-derived cells.

- For example in Fig. 3d, MHCII is not enough to define macrophages as MHCII+ cells could be DCs.

- In Ext. Data Fig. 5. Co-stainings of GFP and CD206 or F4/80 are not evident. It appears that the GFP+ cells are in fact not positive for any of those markers. Insets with higher magnification and single stainings for the different markers should be shown.

- Ext. Data Fig. 5g-l, it is unclear why other immune cells positive for Gr1, CD8, CD4, or NK1.1 were negative for GFP.

They state that BM transplantation between the same genotypes did not change mechanical sensitivities (data not shown). This should be explained. Do they mean that Snx25+/- -> Snx25+/- BM chimeras are similar to Snx25+/- mice or similar to WT -> WT BM chimeras? Their hypothesis is that BM-derived cells contribute to the observed phenotype in Snx25+/- -> WT BM chimeras. One would expect the same result for Snx25+/- -> Snx25+/- BM chimeras.

They state that the number of macrophages after formalin injection was reduced in Snx25+/- mice (Ext. Data Fig. 6). However, this is not quantified and again they just show one single image of one marker (Iba1). Why not show a costaining of Iba1 with CD206 or MHCII also in the hind paw? It is also confusing why two different time points after formalin injections were chosen, d3 for hind paw and d7 for DRG.

In addition, they refer to an accumulation of macrophages after formalin injection. Untreated Snx25+/+ and Snx25+/- mice should be shown in comparison to the formalin injected ones to demonstrate an accumulation of macrophages in the treated group. Where do they reside? Also adjacent to nerves?

After formalin injection, there will probably be a lot of infiltrating immune cells and monocyte-derived cells/macrophages distinct to the resident macrophages. This should also be analyzed.

In Ext. Data Fig. 6a, many of the Iba1+ cells also highly express CD206. Are these CD206+ cells in the DRG replaced in their BM-chimeric mice and are they also targeted in the Cx3cr1CreER mice?

Throughout the manuscript, it would have been better to treat all groups with tamoxifen and take as the control group for example Cx3cr1CreER/WT mice instead of the same genotype +/- tamoxifen.

They sorted F4/80+CD11b+ cells (defined as macrophages) to check for expression of chemokines. They should also analyze whether these cells express Snx25 (control mice) and verify the targeting efficiency, in particular in the Cx3cr1hiMHCIIhi nerve-associated macrophages.

Other myeloid cells including monocytes or Langerhans cells should be excluded and the two subsets of dermal macrophages distinguished by MHCII lo and hi.

The authors claim that SNX25 and Ngf in dermal macrophages are required for pain sensation under normal conditions. While they show that macrophages can express SNX25, and that Ngf is reduced in the Clodronate injected area, they do not demonstrate that Ngf is indeed produced by dermal macrophages.

Ngf was reduced in BMDMs of Snx25^{+/-} mice or in total skin area of Snx25^{+/-} mice (Fig. 5A). This should be shown in dermal macrophages and that Ngf is decreased in dermal macrophages in Snx25^{+/-} mice.
 Fig. 5C, Ngf seem to be expressed by many other cells. Which cells express it also?
 In addition, is Ngf reduced in dermal macrophages of Cxc3cr1CreERSnx25fl/fl mice? What about microglia or macrophages in the DRG?

- Figure 5E, this image is not convincing. Trka staining is very weak.
- Ext. Fig. 1a, they describe a reduction of c-Fos⁺ cells, this is not evident in the figure. Can they explain what c-Fos⁺ cells are?
- Fig. 2I, the figure legend is not clear. What is the light red group and the light blue group?
- For all the immunohistochemistry images, single stains should be shown along the merged file. And images quantified.
- Figure legends lack detailed information. For example, which method was used in Fig. 4e? or is Ext. Data Fig. 6g. expression levels of sorted cells or total skin and was it assessed by RT PCR?

Author Rebuttal to Initial comments

Responses to the Reviewers' Comments

Reviewer #1

Comment:

(Remarks to the Author)

A major finding of this paper is the identification of a potential new modulator of nociception, Snx25, and the discovery that Snx25 can regulate NGF production through reduced Nrf2 ubiquitination. A major weakness of the paper is the lack of data connecting these two findings. The authors propose that Snx25 deficiency leads to behavioral changes via reduced NGF signaling to DRG neurons, and reduced sensitization of Trpv1. While the authors show that NGF is reduced in Snx25 knockdown mice, they do not show that NGF can rescue the behavioral abnormalities or if NGF loss, separately from the other downstream effects of Snx25 knockdown including cytokines and chemokines, is sufficient to produce the phenotype.

Response:

We thank reviewer 1 for the critical review and constructive suggestions. We admit that the link between *Snx25* and NGF production might not have been substantiated enough to convince readers of our story in the original manuscript. According to the reviewer's suggestion, we attempted to reinforce the *Snx25*–*Nrf2*–*Ngf* axis with a set of additional experiments including FACS data (please see the responses below to each comment). In addition to those comments, the reviewer suggested a pivotal experiment in the “Remarks to the Author,” in response to which we investigated whether the pain-insensitive phenotype was rescued by NGF injection in *Snx25* +/- mice. The NGF-injected heterozygote mice became sensitive (very close to the thresholds of NGF-injected WT mice) to mechanical stimuli at 24 h after NGF injection, while vehicle-injected mice (either WT or heterozygote) did not show threshold changes (**Revised Fig. 6j**). This result strongly supports our hypothesis that NGF plays a major role in setting the mechanical sensitivity in the *Snx25*-deficient conditions, although we do not negate additional minor effects of cytokines and chemokines.

We have added a description of the rescue experiment as follows:

Next, we investigated whether the dull phenotype was rescued by NGF injection in *Snx25* +/- mice; the mice became sensitive to pain after NGF injection (**Fig. 6j**). These results suggest that NGF directly causes pain and that NGF level is modulated by *SNX25*–*Nrf2* signaling. (Page 18, lines 4–7.)

The following figure was added to the revised manuscript:

Fig. 6j

Major comments:

Comment:

1. It is unclear why the authors chose to focus on *Snx25* and not the other two genes that were deleted in the TG mice (*Slc25a4* and *Cfap97*). *Slc25a4* has been implicated in skeletal muscle movement and could explain reduced withdrawal and licking responses.

Response:

As we described in the Results section, we found three candidates (*Snx25*, *Slc25a4*, and *Cfap97*) as pain-modulating genes. One or a combination of these three genes could be responsible for the painless phenotype and we ideally needed to have mutants of each to identify the causal gene(s). However, when we started this study (in 2004), gene editing technologies were scarcely available, and time-consuming knock-out was the only (trial-and-error) route for us to take. We therefore had to choose one of the three candidates and to produce or purchase a mutant for that gene. *Cfap97* was excluded because its expression level was low (**Extended Data Fig. 1g**). On the other hand, *Slc25a4* functions as an ATP/ADP co-transporter in the inner mitochondrial membrane, and its role and expression seemed too ubiquitous to account for the specific phenotype. We chose the remaining *Snx25*, whose functions were relatively obscured. *Snx25* heterozygous KO mice reproduced the same pain-insensitive phenotype as that of the *Mlc1* TG mice, and so we further delved into the functions of *Snx25* in the present study. As the reviewer notes, *Slc25a4* has been implicated in skeletal muscle movement (Wang et al., *iScience* 2021, 25:103715; Echaniz-Laguna et al., *J Med Genet.* 2012, 49:146–150), and muscle weakness could result in abnormal behaviors in mechano-sensation tests. In addition, *Slc25a4* may be involved in inflammatory pain, as we have previously found that its knockdown in the macrophage cell line RAW264.7 attenuates IL-6 expression (Nakahara, Tanaka et al., *FEBS Lett.* 2018, 592:3750–3758). Overall, albeit with the above caveat about its ubiquity, we agree that *Slc25a4* is an attractive additional candidate for the knock-out experiment, and we shall consider it for our future work.

Comment:

2. *Snx25* het mice have less *Trpv1*, *Trka*, as well as *Scn9a* and *Scn10a*. Broad changes in gene expression within neurons could explain the phenotype. They also need to rule out any developmental problems in these mice that might affect their behavior.

Response:

We agree with the reviewer that broad changes in gene expression (*Trpv1*, *TrkA*, *Scn9a*, *Scn10a*) in DRG neurons could explain the phenotype. The reviewer may think that *SNX25* in DRG neurons is responsible for the changes. We think this is not the case because DRG neuron-specific cKO of *Snx25* with *Advillin (Avil)^{CreERT2}* driver and tamoxifen treatment (**Extended Data Fig. 5d** and **e**) never caused threshold changes in mechano-sensitivity (**Extended Data Fig. 5f** and **g**). On the other hand, *Cx3cr1^{CreERT2}*-driven cKO caused a pain-insensitive phenotype comparable to that observed in *Snx25* heterozygous KO mice (**Revised Fig. 4b**).

Expression of *Scn9a* and *Scn10a* genes decreased in the DRG of *Cx3cr1^{CreERT2}*-driven cKO mice (**Revised Fig. 4d**). Based on these data, we concluded that changed gene expression of pain-related factors in the DRG and peripheral nerves in *Snx25* +/- mice is the cause of the pain-insensitive phenotype, and that NGF is a major player in making the changes.

We thank the reviewer for pointing out the possibility of abnormal development in the mutant mice. To rule out developmental abnormalities that might affect pain behavior, we performed a von Frey test in young (2 and 3 weeks after birth) *Snx25* +/- mice and WT mice (**Revised Extended Data Fig. 3a**). The pain response to mechanical stimuli was similar between two-week-old WT and *Snx25* +/- mice, indicating that heterozygote mice developed without overt abnormality in pain behavior up to two weeks after birth. heterozygous KO mice began to show reduced pain response at three weeks of age. Even at this stage, cellular size distribution and the expression of small and large neuron markers in the DRG of *Snx25* +/- mice were comparable to those of WT mice (**Revised Extended Data Fig. 3b–e**). We also checked the distribution pattern of dermal macrophages in the three-week-old mice by immunohistochemistry and found that the numbers of CD206- or MHCII-positive macrophages were equivalent between WT and heterozygous KO mice (**Revised Extended Data Fig. 4a and b**). However, the expression level of NGF was somewhat lower in *Snx25* +/- mice (**Revised Extended Data Fig. 4c and d**). Based on the findings, we consider that *Snx25* +/- mice developed normally in the early postnatal period, but later began to acquire a pain-insensitive phenotype due to low NGF expression. We again thank the reviewer for drawing our attention to this important issue.

We have added a description about the normal development of *Snx25* +/- mice as follows:

The pain-insensitive phenotype was not overt at 2 weeks but apparent at 3 weeks of age in *Snx25* +/- mice (**Extended Data Fig. 3a**). We examined whether developmental problems in these mice might affect dull behavior. The cellular size distribution and expression of small and large neuron markers in the DRG of the *Snx25* +/- mice at 3 weeks of age did not change compared to WT mice (**Extended Data Fig. 3b–e**), indicating that *Snx25* +/- mice develop normally. (Page 7, lines 5–10.)

The following figures were added to the revised manuscript:

Extended Data Fig. 3a–e

Extended Data Fig. 4a–d

Comment:

3. The need to reconcile the fact that the *Snx25*(-/-) is embryonic lethal with the fact that the original strain described here (TG mice) had null *Snx25* expression (Fig S1)?

Response:

We agree with the reviewer that there is a discrepancy between *Snx25* -/- mice and *Mlc1* TG mice. This point is of particular interest, and we have a hypothesis that may reconcile the discrepancy. In *Mlc1* TG mice, SNX25 expression was barely detectable at the protein level (**Extended Data Fig. 1i**) using our antibody (Proteintech, 13294-1-AP, 1:500), but the transcript level was slightly higher depending on the primer combination (particularly on the 3' end side) (**Figure A for reviewer 1; please see below**). Although we do not know whether these transcripts can be translated into protein, a weakly expressed variant SNX25 may rescue the *Mlc1* TG mice from embryonic lethality. Alternatively, other compensatory functions could be working in *Mlc1* TG mice. This comment is very important, and we would like to address the problem in our future work.

Figure A for reviewer 1

Start (including putative second translation initiation site) and stop codons are highlighted in red letters.

Snx25 (mouse/C57BL/6) 840aa

```

1 agccctgtat atggaatcc acaggagaca gctcagggca gaagagtggc catttctcac
61 aacatggatc gtbttctgag agatgtgttc gactacagtt acagagacta tattctctcc
121 tggtacggga acctcagcag agatgacgga cagctctacc atctgctctt agatgacttc
181 tgggaaattg tcaagcagat tgcagagag ctgagtcacg tggatgtggt taaagtgtc
241 tgtaatgata tgtaaagcc tttgctcact cacttctgtg acctgaagcc tgctactgcc
301 agacatgaag aacagccaag gccttttgtg ttgcatgcat gcttgaagga ttcacatgat
361 gaagtaagat tccttcaaac atgttctcag gttctggtgt tatgccttct cccttctaag
421 gatatccagt ctctcagctt acgtacaatg ctgcagaaaa ttcttacgac aaaagtcttg
481 aagccagtag tggaaattgct gaytaacctt gactacatta accaaatgct gctcaggcag
541 ttggagtaca gagagcagat gagtgagcat cacaagcgag cctacacgta cgcgccttcc
601 tacgaggact ttatcaagct catcaacagc aactctgatg tggacttctt gaagcagctc
661 aggtatcaaa ttgtagtgga gataatccaa gcaactacaa tcagcagctt tccccagctg
721 aagagacaca agggtaagga atcagctgcc atgaaaaactg atctcttgag ggccagyaac
781 atgaaaaggt acattaacca gctgactgtg gcaaaagaagc agtgtgagaa gagaatccga
841 atctctggag gtcctgccta tgaccagcaa gaggatggag cctcggatga aggggaaggg
901 cctcagagcc aaaagattct tcagtttgaa gatattatga ccaatccttt ctaccgagag
961 cgctttggaa catacatgga acggtatgac aagcgggctc tagttggctt ttggagttc
1021 gccgaacacc tgaagaatgc taacaagagt gaaattccac aattagttag tgaatgttat
1081 cagaatttct tcytggaaag caaggaatc tccgtggaaa agtcaactta caaagaatc
1141 cagcagtgtc ttgtgggaaa cagagcctc gaggtgttca gcaaaatcca agccgatgtg
1201 tccgaggtgc tgagggagcg gtattacccc tccttctctg tcagcgacct gtagagaaag
1261 ctcatgctgt aggaagaaga ggaggagcct gacgctcagc tggcctctga gaaggtgag
1321 ctgggttcag gaggtgagcc tgggtgaggag gctgtagaag gcaccagtgg ggctcagtgt
1381 ccagccagct ttgctgtaat caaactccga gagctaaatg agaaacttga atacaaaag
1441 caagctctaa gttctattca gaatgcacca aaacctgata agaagattat ttccaagttg
1501 aaggtgaaa tacttctgat agagaaagaa tgcacggctc ttcagctgca catggcagga
1561 acagattggt ggtgtgagaa cctgggctgt tggagagcat ccatcaccag cgcagaggtg
1621 acagaagaga atggcgagca aatgccttgt tactttgtca gggtaaatct acaagaagtt
1681 ggagggattg aaactaagaa ctggaccgtc ccaagaagcc tcagtgagtt tcagaattta
1741 catcggaaac ttagttagtg tgtccctctt ttaaaaaaag tccagttgcc ttctctcaat
1801 aagctgcctt tcaaatctat agatcacaag ttcctgggaa agtcagaaga tcagttaaat
1861 gcatttttac agaactgctt ttcagatgaa agactgttcc agagcgaagc actttatgcc
1921 tttttgagcc ctctctctga ctacctcaag gtcattgatg tacaggggaa aaagacctcc
1981 ttctcattgt ctctatttct ggaaaaactt cctcgtgact tcttctccca tcagagagag
2041 gagatagaag aggacagcga cctgtccgat tacggggatg atgtggacgg gaagaaggtat
2101 tccttgctgt aaccgtggtt catgctgatc ggggagattt tcgaactctg aggaatgttt
2161 aaatgggtgc gaagaacatt aatcgtctct gttcaggtca cgtttggaag aacctcaac
2221 aagcaaatcc gcgacacggt gagctggatc tccagcgagc agatgctggt ctactacatc
2281 agtgctttcc gggatgcctt ttggccaac gggaaactgg cacctccaac aagaatccga
2341 agcgtggcac agagttagga gacgaagcag agagcacagc agaagctgct cgagaacatt
2401 ccagatcac ttcagagcct tgttggacag caaaatgccc gccatggtat aataaaaaata
2461 ttcaagggcg tgcaagagac aaaggccaac aaacacctgc tctatgtgct gatggagctg
2521 ctgctgacag agctctgccc tgagctgagg gctcacctgg atcagttcaa agctggccaa
2581 gtctgagagc gcacagagca gccgccagag aaatgtctgt gtaaaaaatg acattaataa

```

Primer set A

Primer set B

Primer set C

Primer set D

Forward primer
Reverse primer

Comment:

4. *Snx25* het mice have defective response to capsaicin this, clearly shows a neuronal intrinsic role for *Snx25*. Despite this, the authors conclude that the pain phenotype is due to *Snx25* function in macrophages only.

Response:

The defective response to capsaicin in *Snx25* +/- mice is likely caused by the reduced expression of TRPV1 (receptor for capsaicin). To test whether *SNX25* in DRG neurons directly regulates TRPV1 expression, we used DRG neuron-specific *Snx25* cKO mice (**Extended data Fig. 5d**).

Advillin (Avil) is well known for its specific expression in DRG neurons (Lau et al., 2011, *Mol. Pain* 7: 1–13) and *Avil*^{CreERT2} mice can target the neurons. Double-TG mice (*Avil*^{CreERT2/WT}; *Snx25*^{loxP/loxP}) treated with tamoxifen could specifically delete the *Snx25* gene in the DRG (Extended data Fig. 5e) but did not show the pain-insensitive phenotype (Extended data Fig. 5f-g). Notably, the DRG neuron-specific *Snx25* cKO mice had *Trpv1*, *Scn9a*, and *Scn10a* expression levels comparable to those of control (*Snx25*^{loxP/loxP}) mice (Extended data Fig. 5i). These data clearly indicate that the defective response to capsaicin in *Snx25* +/- mice is not due to neuron-intrinsic functions of SNX25.

Given these findings, we looked for responsible cellular species other than DRG neurons and found that SNX25 in macrophages regulates NGF expression, and that NGF is the starting point for diminished expression of pain-related genes in sensory neurons, which results in pain insensitivity. Since macrophage-derived NGF regulates the expression of pain factors, we focused on macrophage function in the present study.

Comment:

5. *Trpv1* sensitization is not required for acute nociception and therefore that mechanism would not explain the behavioral results in naïve mice.

Response:

We thank the reviewer for pointing out that *Trpv1* sensitization is not required for acute nociception. Indeed, TRPV1-DTA mice retain normal touch, mechanical pain and proprioceptive responses (Mishra et. al., *EMBO J.* 2011, 30:582–593). We found reduced levels of pain-related factors (*TrkA*, *Scn9a*, *Scn10a*) in the peripheral sensory neurons of *Snx25* +/- mice in addition to the *Trpv1*. Given the above findings, the insensitive phenotype in naïve *Snx25* +/- mice is likely due to decreased expression of *Scn9a* and *Scn10a*. Together with the next comment, the reviewer kindly underlined the implication of the Nav channels. We fully agree with this suggestion.

Comment:

6. The authors also show a reduction in Nav channels which could very well be the actual mechanism by which these mice have reduced sensitivity.

Response:

We agree with the reviewer that reduction in Nav channels within neurons could explain the phenotype. In the revised manuscript, we have added a description of the importance of Nav channels in the pain-insensitive phenotype of *Snx25* cKO mice as follows:

From these data, we concluded that the pain-insensitive phenotype of the *Snx25* cKO mice was due to reduced levels of pain-related factors (especially Na channels) via NGF in the peripheral sensory neurons. (Page 23, lines 22–page 24, line 1).

Comment:

7. (Point 1) Macrophage characterization is insufficient. Dermal macrophages should be facs sorted and investigated for expression of Snx25 and NGF as compared to other myeloid populations.

(Point 2) Flowcytometry should be included to show no alterations in myeloid cell proportions in the dermis (monocytes, dermal Macs, DCs etc).

Response (We have divided the comment into Points 1 and 2; responses are described separately. Point 2 is dealt with first and point 1 follows.)

Point 2.

In accordance with the reviewer's recommendation, we used FACS sorting to examine whether the *Snx25* deletion in macrophages caused any alteration of myeloid proportions. We sorted myeloid proportions using anti-CD45, -CD11b, and -F4/80 antibodies from skin cells of *Cx3cr1^{CreERT2/WT}; Snx25^{loxP/loxP}* mice with and without tamoxifen treatment (**Revised Extended Data Fig. 10b**). We compared the CD45⁺ CD11b⁺ F4/80⁺ population (macrophage-enriched population) and the CD45⁺ CD11b⁺ F4/80⁻ population (non-macrophage myeloid population). There were no significant alterations in these myeloid cell proportions in the dermis (**Revised Extended Data Fig. 10d**). The sorting was insufficient for the detailed differentiation of the myeloid population requested by the reviewer in Point 1. We therefore refined the sorting method by incorporating lineage depletion and subpopulation-specific antibodies.

The following figures were added to the revised manuscript:

Extended Data Fig. 10b and d

Point 1.

In response to the reviewer's request, we tried to sort dermal macrophages by FACS. **Revised Extended Data Fig. 11b** illustrates the steps of FACS sorting of the myeloid cells of WT mice. Briefly (details are given in the figure legend and Methods section), cells were prepared by enzymatic digestion of saline-perfused mouse skin, followed by removal of dead cells and lineage depletion (T-cells, B-cells, NK cells, erythrocytes, granulocytes, Langerhans cells). The CD11b⁺ CD24^{low} population was further sorted by antibody staining for macrophage markers. The combination of Fc-γ receptor 1 (CD64) and MHC class II expression successfully differentiates dermal macrophages (Lin^{neg} CD45⁺ CD11b⁺ Ly6C⁻ CD64⁺ MHCII⁺), monocytes (Lin^{neg} CD45⁺ CD11b⁺ Ly6C⁺ CD64⁻ MHCII⁻), and dendritic cells (Lin^{neg} CD45⁺ CD11b⁺ Ly6C⁻ CD64⁻ MHCII⁺) (Tamoutounour et al., 2013, *Immunity* 39:925–938, Kolter et al., 2019, *Immunity* 50:1482–1497). We investigated the expression level of *Snx25* and *Ngf* in dermal macrophages as compared to other myeloid populations including monocytes and dendritic cells of WT mice. *Snx25* mRNA level was lower in dendritic cells but not significantly different between dermal macrophages and monocytes (**Revised Extended Data Fig. 11c**). *Ngf* mRNA level was a little higher in monocytes but not significantly different among the three groups (**Revised Extended Data Fig. 11c**).

The following figures were added to the revised manuscript:

Extended Data Fig. 11b and c

Regarding Point 2 again.

We applied the refined sorting method to *Snx 25* +/- mice (**Revised Extended Data Fig. 11d**) and found no alterations in myeloid cell proportions (dermal macrophages, dermal monocytes, and dermal dendritic cells) in the dermis in *Snx25* +/- mice compared to WT mice.

We have added a description of these FACS data (refined method) in the Results section. (Page 15 line 1–Page 16 line 24.)

The following figure was added to the revised manuscript:

Extended Data Fig. 11d

Comment:

8. **(Point 1)** The replacement of dermal Macs in the BMT experiments is difficult to confirm just by IHC. Flow cytometry should be performed to show which populations in the skin immune cells is replaced and to what extent. Chimerism in Blood does not mean chimerism in the skin.

(Point 2) The BMT transplant experiments missing WT -> WT and Het -> Het control. Het -> Het control is particularly important to see if the pain phenotype gets more pronounced.

Response (Again we have divided the comment into Points 1 and 2 and responded to them separately.)

Point 1.

In response, we examined the replacement of dermal macrophages by FACS after transferring GFP mouse bone marrow into WT mice (**Fig. 3b**). We also used F4/80 antibody to label macrophages in addition to MHCII. The majority of the replaced cells in recipient hind paw skin were MHCII- and F4/80-positive (**Revised Fig. 3f**, GFP⁺ MHCII⁺ cells, 39.3%; GFP⁺ F4/80⁺ cells, 38.2%, respectively). We found that the proportions of MHCII⁺ (68%) and F4/80⁺ (62%) populations in total GFP⁺ cells were high, and in turn each marker-positive population contained more than 80% of GFP⁺ cells (**Revised Fig. 3g**). Consistent with the immunohistochemical data (**Revised Extended Data Fig. 8a**), Gr1-positive cells were only 1.0% (GFP⁺ Gr1⁺) of the total GFP⁺ cells (**Revised Extended Data Fig. 8g**). We also found that very small proportions of CD19⁻, CD8a⁻, CD4⁻, and NK1.1-positive cells were replaced by GFP⁺ cells (**Revised Extended Data Fig. 8g**, 0.3%, 0.5%, 1.0%, and 0.2%, respectively). Again, these findings are consistent with the immunohistochemical data (**Revised Extended Data Fig. 8b–e**). Similar results were obtained in recipient back skin (**Revised Extended Data Fig. 8h**, GFP⁺ MHCII⁺ cells, 53.3%; GFP⁺ F4/80⁺ cells, 41.8%; GFP⁺ Gr1⁺ cells, 3.4%; GFP⁺ CD19⁺ cells, 0.4%; GFP⁺ CD8a⁺ cells, 1.8%; GFP⁺ CD4⁺ cells, 2.8%; GFP⁺ NK1.1⁺ cells, 0.5%).

We revised the manuscript as follows:

We also examined the replacement of dermal macrophages by fluorescence-activated cell sorting (FACS). The majority of dermal macrophages defined by the expression of F4/80 or MHCII were replaced into donor GFP-positive cells in recipient hind paw skin (**Fig. 3f** and **g**). Other populations in GFP-positive cells were also rare by flowcytometry (**Extended Data Fig. 8g**). Similar results were obtained for the dermal macrophages of back skin. (**Extended Data Fig. 8h**) These results suggest that dermal macrophages are constantly replaced by bone marrow-derived cells. (Page 10, lines 12–18.)

The following figures were added to the revised manuscript:

Fig. 3f and g

Extended Data Fig. 8g and h

Point 2.

We thank the reviewer for this important suggestion. We performed the von Frey test in WT mice with BM transferred from WT mice, as a control (**Revised Extended Data Fig. 8j**), which confirmed that the withdrawal threshold did not change before and after BMT. Likewise, the high withdrawal threshold in *Snx25* +/- mice did not change in *Snx25* +/- mice with BM transferred from *Snx25* +/- mice (**Revised Extended Data Fig. 8j**).

We added a description about these BMT experiments between the same genotype as follows:

BM transplantation between the same genotypes (WT to WT or *Snx25* +/- to *Snx25* +/-) did not affect mechanical sensitivities before or after transplantation (**Extended Data Fig. 8j**). (Page 11, line 3–5.)

The following figure was added to the revised manuscript:

Extended Data Fig. 8j

Comment:

9. For the *Cx3cr1* cre ER experiments, the authors cite a study where *Cx3cr1* hi Macs were shown to interact with dermal Macs. However, the BMT strategy used here has been shown to not replace the *Cx3cr1* hi population, since it was embryonic. Therefore, Tamoxifen induced deletion data should be included to show specific and sufficient deletion.

Response:

This is one of the most critical issues in the present study and we again thank the reviewer for this constructive suggestion. We consider that the dermal macrophages we are dealing with in the present work are not the same as the “*Cx3cr1* hi population” described by Kolter et al. (2019, *Immunity* 50:1482–1497), but comprise a broader population (including their “*Cx3cr1* int population”), judging by the following marking experiment. To visualize replacement of the *Cx3cr1*⁺ population, we crossed *Cx3cr1*^{CreERT2/WT}; *Snx25*^{loxP/loxP} mice with Ai32; *Snx25*^{loxP/loxP} mice to generate triple TG mice (*Cx3cr1*^{CreERT2/WT}; *Snx25*^{loxP/loxP}; Ai32/+). The Ai32 mice harbor a transgene (CAG-Flex-ChR2(H134R)-YFP) and, upon Cre recombinase activation, they express channelrhodopsin2 tagged with YFP protein (<https://www.jax.org/strain/012569>). Treatment with the soluble-type tamoxifen derivative 4-OH tamoxifen (4-OHT) (1 μM, 7–8 days) (**Figure B for reviewer 1, please see below**) switched on YFP expression in bone marrow-derived macrophages (BMDMs) of triple TG mice (**Revised Fig. 5h and i**; average YFP-positive live cells, 44.2% (n = 6), **please also see below**). YFP-positive macrophages were sorted from 4-OHT-treated BMDMs of these triple TG mice and we examined the expression of *Snx25* and *Ngf* transcripts. *Ngf* as well as *Snx25* mRNA were significantly reduced in the YFP-positive macrophages as compared to YFP-negative ones (**Revised Fig. 5j, please also see below**). Furthermore, bone marrow transplantation was performed to find out how much of the *Cx3cr1* population would be replaced. We transplanted bone marrow from triple TG mice into WT mice (**Revised Extended Data Fig. 12a, please also see below**). YFP expression was detected in dermal macrophages in tamoxifen-treated triple TG mice (**Revised Fig. 5k, Revised Extended Data Fig. 12b, please also see below**). These data suggest that the *Cx3cr1*⁺ population not only exists from the embryonic stage but also is replenished from the bone marrow. We also selectively collected YFP-expressed donor-derived *Cx3cr1*⁺ macrophages from the skin of the WT recipient mice and examined their gene expression patterns. *Snx25* expression was significantly decreased in YFP-positive macrophages compared to YFP-negative macrophages ($p = 0.008$, **Revised Extended Data Fig. 12c, please also see below**). These results indicate that the expression of SNX25 in *Cx3cr1*⁺ macrophages can be sufficiently attenuated by tamoxifen treatment after BMT even in adult mice.

Based on these additional findings, we added a description about the replacement of dermal macrophages in the BMT experiments in the Discussion section as follows:

Interestingly, a $Cx3cr1^{int\ to\ hi}$ population does exist not only during the embryonic stage but also in the adult dermis, and was replaced in our BMT experiments (**Fig. 5k, Extended Data Fig. 12a and b**). (Page 21, lines 20–22.)

Figure B for reviewer 1

Revised Fig. 5h-i

Revised Fig. 5j

Extended Data Fig. 12a-c

Revised Fig. 5k

The following figures were added to the revised manuscript.

Fig. 5h-k

Extended Data Fig. 12a-c

Comment:

10. NGF expression was only checked in *Snx25*het mice. What about the expression of NGF expression in Mac specific conditional KO? This is critical piece of evidence for the proposed mechanism.

Response:

To answer the reviewer's important question, we performed Western blot analysis to confirm the expression of NGF at the protein level in TAM-administered *Cx3cr1*^{CreERT2/WT}; *Snx25*^{loxP/loxP} mice. NGF expression in hind paw skin in cKO mice was lower than that in *Snx25*^{loxP/loxP} mice (**Revised Extended Data Fig. 11a**). We further examined whether *Ngf* transcript level was reduced in dermal macrophages. Dermal macrophages were selectively collected from the skin of the *Cx3cr1*^{CreERT2/WT}; *Snx25*^{loxP/loxP} mice by FACS and gene expression patterns were examined. SNX25 depletion in dermal macrophages decreased the expression of *Ngf*, although the decrease did not quite reach statistical significance (**Revised Extended Data Fig. 11b and f**,

$p = 0.057$). These results indicate that SNX25 in dermal macrophages contributes to *Ngf* expression.

We added a description about the SNX25–NGF relationship in the FACS-sorted macrophages in the Results section (Page 15 line 10 to Page 16 line 10).

The following figures were added to the revised manuscript:

Extended Data Fig. 11a, b, and f

Comment:

11. None of the experiments tie together the *Snx25* - *Nrf2* - *Ngf* axis. Can authors overexpress *Nrf2* in *Snx25* deficient cells and look for NGF expression? Ideally, the ubiquitination site should be mutated.

Response:

This suggestion is very reasonable, and we tried to overexpress *Nrf2* in the macrophage cells using the pcDNA3.1/Myc-His vector. We tried a battery of transfection systems such as including Lipofectamine 2000, Lipofectamine 3000, and Lipofectamine LTX (all from Thermo Fisher Scientific) as well as jetPEI-Macrophage (Polyplus transfection), following the manufacturers' instructions, and many times, but the expression was never as strong as expected. Macrophages and macrophage cell lines are notorious for their recalcitrance to gene transfection. We had to abandon the overexpression experiments. Instead, we took an alternative approach to activate *Nrf2*. *Nrf2* level in the cell is known to be regulated by continuous ubiquitination and proteasome degradation, which is blocked by *Keap1* protein. *Keap1* knock-down thus leads to *Nrf2* activation (Kensler et al., 2007, *Annu Rev Pharmacol Toxicol.* 47, 89–116). We found that suppressing *Keap1* expression with *Keap1* siRNA in BMDMs of *Snx25* +/- mice rescued NGF expression (**Revised Fig. 6i**).

We added a description about the *Keap1* experiment as follows:

We also found that suppressing Keap1 (known to accelerate Nrf2 degradation) expression with *Keap1* siRNA in BMDMs of *Snx25* +/- mice rescued *Ngf* expression (**Fig. 6i**). (Page 18, lines 2–4.)

The following figure was added to the revised manuscript with a legend:

Fig. 6i

Other comments:

Comment:

Figure 2:

1. The authors focus on mechanical allodynia and then go on to look at Trpv1 levels and capsaicin response without showing any heat sensitivity assays, in S1 they show the mice show normal heat sensitivity at 2 months but insensitivity at 6-8months. The authors need to reconcile the in vitro data from figure 1 with the delayed in vivo response.

Response:

We accept that there are some discrepancies between *in vitro* and *in vivo* data from the point of view of heat sensitivity. For some unknown reasons, thermal nociception was normal in the 2-month-old *Snx25* +/- mice but was later affected at 6–8 months of age (**Extended Data Fig. 2b**). We do not have an appropriate explanation for this phenomenon; interestingly, however, we note that heterozygous knock-in mice carrying the human R100W-mutated NGF, which is the causal gene in Hereditary Sensory and Autonomic Neuropathy type V (HSAN V), have a similar phenotype: thermal nociception was normal at 2 months of age and decreased at 6 months, with adult NGF^{R100W/wt} mice displaying a higher latency to respond to a high-temperature stimulus (Testa et al., *J. Neurosci.* 2019, 39: 9702-9715). Because *Snx25* deletion also caused a reduction in NGF, there may be some unknown mechanism(s) that compensate for NGF-related heat sensitivity in younger mice. We would like to investigate possible mechanisms underlying the discrepancy in our future work.

Comment:

2. The authors attribute the decreased calcium response to “Trpv1 channel inactivation” but they do not conduct any electrophysiological studies to show that. This could merely be due to reduced Trpv1 expression.

Response:

We accept that the expression “TRPV1 channel inactivation” in the original manuscript (original manuscript, page 7, line 14) was an overstatement as the reviewer indicated. What we did show was that the dull phenotype of *Snx25* +/- mice is due to reduced expression of pain-related factors, including TRPV1. We thank the reviewer for pointing out the discrepancy.

We revised the manuscript as follows:

Capsaicin elevated the intracellular Ca level in a population of primary cultured DRG neurons, but the amplitude of this Ca elevation was smaller in *Snx25* +/- neurons than in WT neurons, indicating that SNX25 deficiency resulted in a reduction of TRPV1 channel expression in the DRG neurons (**Fig. 2I**). (Page 7, lines 18–21.)

Comment:

3. 10uM capsaicin is cytotoxic to neurons.

Response:

We appreciate this comment because we made a careless mistake in the original manuscript. We carefully reviewed the raw data and found that the concentration thought to be 10µM capsaicin was actually 1µM. We are really embarrassed by this mistake and again thank the reviewer for leading us to find it. Accordingly, the description of the concentration was corrected and revised the figure (**Revised Fig. 2I**).

The following figure was revised:

Fig. 21

Comment:

Figure 3:

Authors need to show the withdrawal threshold in WT mice with BM transferred from WT mice, as a control.

Response:

We complied with the reviewer's request and performed the control von Frey test in WT mice with BM transferred from WT mice. This confirmed that the withdrawal threshold did not change before and after BM transplantation (**Revised Extended Data Fig. 8j**).

The following figure was added to the revised manuscript:

Extended Data Fig. 8j

Comment:

Figure 4:

Il1b is reduced in *Snx25*(+/-) and is known to sensitize nociceptors, why did the authors choose to focus on NGF and not Il1b?

Response:

Thank you for this pertinent comment. *Snx25* cKO in macrophage-lineage cells reduced pain responses in both normal and neuropathic conditions. We demonstrated that *Snx25* cKO in macrophage-lineage cells reduced the NGF expression; therefore, SNX25 activates NGF production under both normal and painful conditions via NGF/TrkA signaling. We also showed that the expression of IL-1 β was lower in 5% formalin-injected hind paw skin of *Cx3cr1*^{CreERT2/WT}; *Snx25*^{loxP/loxP} mice than in *Snx25*^{loxP/loxP} mice (**Fig. 4e**). IL-1 β could indeed be a legitimate candidate considering its well-known functions in inflammation and injury. However,

the level of IL-1 β in hind paw skin of *Snx25* +/- mice under normal conditions was comparable to that in the *Snx25* ++ mice (**Figure C for reviewer 1, please see below**), while NGF showed the reduced level even in the naïve condition (Fig. 5a). We further investigated the expression level of *Il1b* in dermal macrophages (Lin^{neg} CD45⁺ CD11b⁺ Ly6C⁻ CD64⁺ MHCII⁺) of *Snx25* +/- mice and compared it with WT mice under normal conditions. The mRNA expression level of *Il1b* was not different between WT and *Snx25* +/- mice (**Figure D for reviewer 1, please see below**). We would like to emphasize the mechanical sensitivity in normal conditions as much as in pathological conditions. In this regard, NGF fits much better in the proposed mechanisms. We therefore focused on NGF rather than IL-1 β .

Figure C for reviewer 1

Figure D for reviewer 1

Reviewer #2

(Remarks to the Author)

In this study, the authors identified *Snx25* as a pain-modulating gene. Interestingly, by employing *Snx25* conditional-mutant mice, the authors demonstrate that monocyte/macrophage-lineage cells are responsible for reduced pain responses. *SNX25* enhanced the expression of macrophage-derived NGF via the inhibition of ubiquitin-mediated degradation of Nrf2.

This is an interesting and timely study indicating that in addition to their canonical defence function, dermal macrophages set pain thresholds and sensitivity via NGF production. The experiments have been performed to a high standard and vast majority of the claims are supported by the data. Nevertheless, some additional points should be addressed to improve the manuscript.

Specific points:

Comment:

1) In figure 2k the authors show representative images of Trka expression and conclude that the expression is reduced in *Snx25* $+/-$ mice when comparing with *Snx* $+/+$. It would be beneficial to include quantification across several experiments. The same applies to Figure 7c.

Response:

We thank the reviewer for the comment. In response to this suggestion, we performed semi-quantitative analysis of the images in question using ImageJ software.

The following figures were revised:

Fig. 2k

Fig. 7c

Comment:

2) It is not obvious how the qPCR data was calculated and therefore it is difficult to interpret the results. Figure panels or legends should specify what method was used to quantify the expression levels in 2m, 4d, 5f, 6a.

Response:

We are sorry for not including a detailed explanation. Quantification of gene expression was calculated by the $\Delta\Delta C_t$ method (Livak and Schmittgen. *Methods*, 2001, 25: 402-408). Furthermore, the signal value was denoted as a fold change corrected by the signal value of control (*Snx25* +/+, scramble siRNA, etc.). We added detailed information to the Methods section. (Page 48, line 5–page 48, line 10.)

Comment:

3) Bone-marrow mixed chimeras of *Snx25* +/- and *Snx25* +/+ against a WT competitor should be employed to formally demonstrate the cell-autonomous impact of *Snx25* on macrophage derived NGF.

Response:

This is a very important point, and we thank the reviewer for the invaluable suggestion. In response, we performed a mixed BM competition assay. Donor BM cells from GFP mice (*Snx25* +/+, CD45.2) and *Snx25* +/- (CD45.2) mice were mixed at a 1:1 ratio and transferred to Ly5.1 (CD45.1) recipient mice. Donor BM-derived *Snx25*-knockdown macrophages in recipient mice expressed lower levels of *Snx25* and *Ngf* compared to donor BM-derived SNX25-normal-

expressing macrophages, although these decreases marginally failed to reach statistical significance (n = 4, **Revised Extended Data Fig. 13a-c**).

To further establish the importance of SNX25 expressed in BM-derived dermal macrophages, we also transplanted BM from triple TG mice (*Cx3cr1*^{CreERT2/WT}; *Snx25*^{loxP/loxP}; Ai32/+) into WT mice. Tamoxifen-induced YFP expression was detected in dermal macrophages in tamoxifen-treated triple TG mice (**Revised Extended Data Fig. 12a, Revised Fig. 5k**). Based on these results, macrophages were also selectively collected from the skin of the WT recipient mice and their gene expression patterns were examined (**Revised Extended Data Fig. 12b**). SNX25 depletion in macrophages derived from BM had a significantly decreased *Ngf* level (n = 5, **Revised Extended Data Fig. 12c**). These results indicate that SNX25 expressed in BM-derived dermal macrophages contributes to *Ngf* expression.

We added a description about the mixed BM competition assay in the Results section (Page 16, lines 18–24).

The following figures were added to the revised manuscript:

Fig. 5k

Extended Data Fig. 12a–c

Extended Data Fig. 13a–c

Comment:

4) The authors claim that “most of the present data were obtained from male mice”. Given that sex dimorphism plays such a role in pain research, it would be highly beneficial to specify the sexes of animals used in specific experiments. And to provide data for both males and females, when possible.

Response:

We appreciate the reviewer’s deep thought. As indicated, sexual dimorphisms of pain perception have recently been demonstrated, and males and females reportedly have different pain mechanisms. We noticed similar discrepancies in mechano-sensation between male and female *Snx25* +/- mice. For example, a von Frey test in female WT and *Snx25* +/- mice revealed that the

withdrawal threshold tended to increase in *Snx25* +/- mice as in males, but the difference was not significant (**Revised Extended Data Fig. 2a**). In the present study, we limited the analysis to male mice, and we clearly indicated this limitation in Methods (Page 43, lines 6–9).

We also added a description about this gender issue in the Discussion section, as follows:

In our experiments, the 50% withdrawal threshold to mechanical stimuli in the paws tended to increase in the female *Snx25* +/- mice, but the difference between the two groups was not statistically significant (**Extended Data Fig. 2a**). Therefore, we focused on male mice in the present study. (Page 23, lines 8–11.)

The following figure was added to the revised manuscript:

Extended Data Fig. 2a

Comment:

5) All data that is mentioned in the manuscript as “data not shown” should be included in the supplementary figures.

Response:

Following the reviewer’s suggestion, all such data are now presented in the supplementary figures.

The following figures were added to the revised manuscript:

Fig. 4g

Extended Data Fig. 8i

Extended Data Fig. 8j

Extended Data Fig. 9e

Reviewer #3:

Remarks to the Author:

The authors found that *Snx25*-deficient mice display reduced pain behavior. Conditionally deleting *Snx25* in *Cx3cr1*⁺ cells also led to a pain-insensitive phenotype whereas mice in which *Snx25* was specifically deleted in DRG (using *AvilCreER* mice) had normal pain responses. Using BM-chimeric mice, they further conclude that BM-derived dermal macrophages contribute to pain sensation. In *Snx25*^{+/-} mice levels of *Ngf* or *Nrf2* were reduced, indicating that *Snx25* in macrophages acts via *Ngf* signaling.

It is an interesting study describing a role for dermal macrophages in pain sensitivity. However, their claim is overinterpreted. Dermal macrophages should be characterized in more detail. It is not entirely clear that they express *Ngf* and *Nrf2* *in vivo* or that it is reduced in dermal macrophages in *Snx25*^{+/-} mice. Whether DRG macrophages further contribute to the observed phenotype is also not entirely clear.

Specific comments

Comment:

They state that a population of macrophages (positive for MHCII, CD206, or F4/80) were closely associated with PGP9.5-positive sensory fibers and were SNX25-immunoreactive (Fig. 3a, Extended Data Fig. 4a). However, Fig. 4a only shows MHCII⁺ cells, which could also be dendritic cells. A co-staining of MHCII with another macrophage marker and PGP9.5 should be shown.

Response:

This is a very important point, and we thank the reviewer for the invaluable suggestion. To determine the relationship between SNX25-positive dermal macrophages and peripheral nerves, we crossed *Cx3cr1*^{CreERT2/WT}; *Snx25*^{loxP/loxP} mice with reporter mice harboring CAG-Flex-ChR2(H134R)-EYFP (Ai32, <https://www.jax.org/strain/012569>). Tamoxifen administration resulted in YFP expression in *Cx3cr1*/MHCII-positive macrophages. We also confirmed that F4/80-positive and CD206-positive dermal macrophages were co-stained with YFP in tamoxifen-administered *Cx3cr1*^{CreERT2/WT}; *Snx25*^{loxP/loxP}; Ai32 mice (**Revised Fig. 4g**).

Furthermore, following the reviewer's suggestion, we performed co-staining of MHCII with another macrophage marker (CD206) and PGP9.5 (**Revised Fig. 4i**).

We added a description about the macrophage markers staining as follows:

Immunohistochemistry with two other macrophage markers, F4/80 and CD206, confirmed that dermal macrophages colocalize with peripheral nerves (**Fig. 4i**). (Page 13, lines 4–5.)

The following figures were revised to the revised manuscript:

Fig. 4g

The following figures were added to the revised manuscript:

Fig. 4i

Comment:

(Point 1) In Ext. Fig. 4a it appears that all Snx25 expressing cells are macrophages. Is this the case? Or do other cells in the dermis also express Snx25?

(Point 2) What about macrophages in the DRG? Do they express Snx25?

Response (The comment was divided into Point 1 and 2 and a separate response is provided for each):

Point 1.

To answer the reviewer's question, SNX25 expression in other cells was examined by double-labeling immunohistochemistry (**Revised Extended Data Fig. 6b**). CD117 and Gr1 immunoreactivities were found in the SNX25⁺ cells, suggesting that SNX25 is also expressed in mast cells, neutrophils, and monocytes, although not as much as in macrophages. On the other hand, B, T, and NK cells were hardly stained (**Revised Extended Data Fig. 6b**). Interestingly, Western blot analyses also revealed that the SNX25 expression level was significantly lower in the clodronate liposome-injected skin area than in the control liposome-injected area (**Fig. 7f and h**, indicating that SNX25 is most abundantly expressed in phagocytes such as macrophages).

The following figure was added to the revised manuscript:

Extended Data Fig. 6b

Point 2.

This is an interesting and important point that we have been paying attention to, since recent work has shown that macrophages in the DRG mediate neuropathic pain (Yu, X. et al. 2020, *Nat. Commun.* 11, 1–12). In response to the reviewer's query, we confirmed that the expression of SNX25 in DRG macrophages is not as high as in skin (**Revised Extended Data Fig. 14b**). To test the contribution of DRG macrophages to pain sensitivity, we administered 4-OHT onto exposed DRGs (L4 and L5) of *Cx3cr1^{CreERT2/WT}; Snx25^{loxP/loxP}; Ai32/+* mice. At 5 days after administration, ipsilateral hind paws did not show a pain-insensitive phenotype in *Cx3cr1^{CreERT2/WT}; Snx25^{loxP/loxP}; Ai32/+* mice (**Revised Extended Data Fig. 14j**), suggesting that SNX25 in DRG macrophages is not a pivotal factor for pain sensation.

The following figures were added to the revised manuscript:

Extended Data Fig. 14b

Extended Data Fig. 14j

Comment:

(Point 1) Ext. Fig. 4b, the authors conclude that the numbers of dermal macrophages are not reduced in Snx25^{+/-} mice. Also here, they should co-stain the MHCII⁺ cells (nerve-associated macrophages) with a macrophage marker and quantify them.

(Point 2) Are numbers and distribution of macrophages in the DRG in Snx25^{+/-} mice different compared to WT mice?

Response:

Point 1.

According to the reviewer's suggestion, we co-stained the MHCII⁺ cells with the macrophage marker F4/80 in hind paw skin of WT and *Snx25* +/- mice and quantified them (**Revised Extended Data Fig. 6c and d**). The mean fluorescence intensities of CD206, MHCII, and F4/80 immunoreactivities showed no difference between wild and SNX25 heterozygote mice.

We added a description about the staining for multiple macrophage markers as follows:

Immunohistochemistry revealed that the numbers of CD206-, MHCII-, or F4/80-positive macrophages in hind paw skin of *Snx25* +/- mice were comparable to those in the WT mice (**Extended Data Fig. 6c and d**). (Page 9, lines 15–17.)

The following figures were revised:

Extended Data Fig. 6c and d

Point 2.

We checked the numbers and distribution of macrophages in the DRG of WT and *Snx25* +/- mice. The number and distribution of macrophages in DRG in *Snx25* +/- mice was normal; however, the accumulation of macrophages after formalin injection was reduced in *Snx25* +/- mice (**Revised Extended Data Fig. 9a and b**).

The following figures were revised:

Extended Data Fig. 9a and b

Comment:

BM chimeric studies: It is not clear why they perform BM transplantation studies. The ontogeny of dermal macrophages and their partial replacement by BM monocytes over time has been reported (for example Liu et al. 2019). In contrast to fate-mapping studies, in a BM-transplant setting, most macrophage populations are replaced by donor cells with the exception of a few populations (for example Langerhans cells or microglia).

Are DRG macrophages replaced by donor-derived cells? Do they express Cx3cr1, *Snx25*, Nrf2, Ngf?

Response:

We appreciate the reviewer's comment. Indeed, multiple studies have reported that BM monocytes replace dermal macrophages at a certain rate. The main reason why we performed the BM transplantation was that we wished to limit the SNX25 conditional deletion to dermal macrophages. *Cx3cr1^{CreERT2/WT}; Snx25^{loxP/loxP}* mice could delete SNX25 in the monocyte/macrophage lineage cells, but this conditional KO also affects microglia in the central nervous system. As microglia in the spinal cord play important roles in pain sensation, we had to differentiate these two populations, and BM transplantation allowed us to circumvent the problem.

The BM transplantation experiment using GFP mice (C57BL/6-Tg (CAG-EGFP)) revealed that DRG macrophages (MHCII-positive cells) were partly derived from bone marrow and that they turned over (**Revised Extended Data Fig. 14c-d**). These donor-derived DRG macrophages expressed Nrf2 weakly, whereas we could not detect NGF expression in these cells. We also compared the expression level of NGF in hind paw skin, DRG, and spinal cord of the same individual. The expression of NGF was much lower in DRG and spinal cord than in hind paw skin (**Figure C for reviewer 3, please see below**). For detection of *Cx3cr1* and SNX25, we crossed *Cx3cr1^{CreERT2/WT}; Snx25^{loxP/loxP}* mice with *Ai32*; *Snx25^{loxP/loxP}* mice to generate triple TG mice (*Cx3cr1^{CreERT2/WT}; Snx25^{loxP/loxP}; Ai32/+*), in which we could visualize cells with recombination by YFP fluorescence, and transplanted BM from triple TG mice into WT mice. YFP (*Cx3cr1*-expressing cells with knocked-out *Snx25*) expression was detected in MHCII-positive DRG macrophages in TAM-treated triple TG mice (**Figure D for reviewer 3, please see below**). Considering the difference in their expression level, SNX25 expression and the mechanism of NGF regulation may differ between dermal macrophages and DRG macrophages.

The following figures were added to the revised manuscript:

Extended Data Fig. 14c and d

Figure C for reviewer 3

Figure D for reviewer 3

*Comment:*

(Point 1) In addition, the images shown are not sufficient to define the replacement of dermal macrophages by GFP+ donor-derived cells.

- For example in Fig. 3d, MHCII is not enough to define macrophages as MHCII+ cells could be DCs.

(Point 2) - In Ext. Data Fig. 5. Co-stainings of GFP and CD206 or F4/80 are not evident. It appears that the GFP+ cells are in fact not positive for any of those markers. Insets with higher magnification and single stainings for the different markers should be shown.

(Point 3) - Ext. Data Fig. 5g-l, it is unclear why other immune cells positive for Gr1, CD8, CD4, or NK1.1 were negative for GFP.

Response (We have divided the comments into Points 1, 2 and 3 and responded separately to each):

Point 1.

This is a very important point, and we thank the reviewer for the valuable comment. We confirmed that donor-derived dermal macrophages expressed CD206 and F4/80 in addition to MHCII (**Revised Extended Data Fig. 7b–d**). The replacement of donor-derived cells was also examined by FACS. We used F4/80 antibody to define cells as macrophages in addition to MHCII. The majority of the replaced cells in recipient hind paw skin were F4/80- and MHCII-positive cells (**Revised Fig. 3f**, GFP+ MHCII+ cells, 39.3%; GFP+ F4/80+ cells, 38.2%), consistent with the results of immunohistochemistry. These results indicate that macrophages are continuously replaced by bone marrow-derived cells.

The following figures were revised:

Extended Data Fig. 7b–d

Fig. 3f

Point 2.

Following the reviewer's suggestion, immunostaining images were added so that individual staining reactions and magnified images are now displayed. Double-positive cells are marked on the magnified merged images (**Revised Extended Data Fig. 7b–d**).

The following figures were added:

Extended Data Fig. 7b–d

Point 3.

The replacement of other immune cells positive for Gr1, CD8, CD4, or NK1.1 was also examined by FACS. Gr1-, CD19-, CD8a-, CD4-, or NK1.1- and GFP-double-positive cells were hardly replaced (**Revised Extended Data Fig. 8g**, 1.0%, 0.3%, 0.5%, 1.0%, and 0.2%,

respectively). Similar results were obtained in recipient back skin (**Revised Extended Data Fig. 8h**, GFP+ Gr1+ cells, 3.4%; GFP+ CD19+ cells, 0.4%; GFP+ CD8a+ cells, 1.8%; GFP+ CD4+ cells, 2.8%; and GFP+ NK1.1+ cells, 0.5%). These results were consistent with those of immunohistochemistry, further indicating that macrophages are constantly replaced by bone marrow-derived cells, whereas neutrophils, B cells, T cells, and NK cells are stable in the naive condition.

The following figures were added to the revised manuscript:

Extended Data Fig. 8g

Extended Data Fig. 8h

We added a description about the replacement of dermal macrophages and other immune cells in the BMT experiments as follows:

We also examined the replacement of dermal macrophages by fluorescence-activated cell sorting (FACS). The majority of dermal macrophages defined by the expression of F4/80 or MHCII were replaced into donor GFP-positive cells in recipient hind paw skin (**Fig. 3f and g**). Other populations in GFP-positive cells were also rare by flowcytometry (**Extended Data Fig. 8g**). Similar results were obtained for the dermal macrophages of back skin. (**Extended Data Fig. 8h**) These results suggest that dermal macrophages are constantly replaced by bone marrow-derived cells. (Page 10, lines 12–18)

Comment:

They state that BM transplantation between the same genotypes did not change mechanical sensitivities (data not shown). This should be explained. Do they mean that Snx25^{+/-} -> Snx25^{+/-} BM chimeras are similar to Snx25^{+/-} mice or similar to WT -> WT BM chimeras? Their hypothesis is that BM-derived cells contribute to the observed phenotype in Snx25^{+/-} -> WT BM chimeras. One would expect the same result for Snx25^{+/-} -> Snx25^{+/-} BM chimeras.

Response:

We are sorry for the lack of the detailed explanation in this case. We meant that Snx25^{+/-} -> Snx25^{+/-} BM chimeras are similar to Snx25^{+/-} mice. Likewise, WT -> WT BM chimeras are

similar to WT mice. To make these phenomena clearly understandable, we added a new figure (**Revised Extended Data Fig. 8j**). We performed the von Frey test in WT mice with BM transferred from WT mice, as a control. We confirmed that the withdrawal threshold did not change before and after BMT. Furthermore, the high withdrawal threshold in *Snx25* +/- mice did not change in *Snx25* +/- mice with BM transferred from *Snx25* +/- mice (**Revised Extended Data Fig. 8j**).

Text in the Results section was revised as follows to explain the result more precisely:

BM transplantation between the same genotypes (WT to WT or *Snx25* +/- to *Snx25* +/-) did not affect mechanical sensitivities before or after transplantation (**Extended Data Fig. 8j**). (Page 11, lines 3–5)

The following figure was added to the revised manuscript with legend.

Extended Data Fig. 8j

Comment:

They state that the number of macrophages after formalin injection was reduced in *Snx25*+/- mice (Ext. Data Fig. 6). However, this is not quantified and again they just show one single image of one marker (Iba1). Why not show a costaining of Iba1 with CD206 or MHCII also in the hind paw? It is also confusing why two different time points after formalin injections were chosen, d3 for hind paw and d7 for DRG.

Response:

In response to the reviewer's comment, we performed immunohistochemical analysis in the hind paw skin 7 days after formalin injection, i.e., at the same time point that we used for the DRG. In addition, double staining of Iba1 and CD206 was performed with hind paw skin after formalin injection (**Revised Extended Data Fig. 9a**). We also performed quantitative analysis of immunohistochemistry images after formalin injection (**Revised Extended Data Fig. 9b**).

We added a description about the Extended Data Fig. 9b as follows:

Immunohistochemistry revealed that fewer macrophages accumulated after formalin injection in *Snx25* +/- mice (**Extended Data Fig. 9a and b**). (Page 11, lines 12–14.)

The following figures were added to the revised manuscript:

Extended Data Fig. 9a and b

Comment:

(Point 1) In addition, they refer to an accumulation of macrophages after formalin injection. Untreated *Snx25*+/+ and *Snx25*+/- mice should be shown in comparison to the formalin injected ones to demonstrate an accumulation of macrophages in the treated group. Where do they reside? Also adjacent to nerves?

(Point 2) After formalin injection, there will probably be a lot of infiltrating immune cells and monocyte-derived cells/macrophages distinct to the resident macrophages. This should also be analyzed.

Response:

Point 1.

Following the reviewer's suggestion, we have added images of untreated (naive) *Snx25*+/+ and *Snx25*+/- mice (**Revised Extended Data Fig. 9a**). Immunohistochemistry revealed that the accumulation of macrophages in the DRG after formalin injection was reduced in *Snx25* +/- mice; however, the number of macrophages in hind paw skin in *Snx25* +/- mice was normal. Furthermore, we performed immunohistochemical analysis to demonstrate where the macrophages reside (**Revised Extended Data Fig. 14a**). Some macrophages associate with nerves, while others are located around the cell body. There was no appreciable difference in overall macrophage location between WT and *Snx25* +/- mice.

The following figures were added to the revised manuscript:

Extended Data Fig. 9a

Extended Data Fig. 14a

Point 2.

Following the reviewer's suggestion, we performed immunohistochemical analysis of infiltrating immune cells after formalin injection (**Revised Extended Data Fig. 9g**). CCR2-positive monocytes are reportedly recruited from peripheral blood to wound tissue following injury (Boniakowski et al., *Eur. J. Immunol.* 2018, 48: 1445–1455). Immunohistochemistry revealed that the accumulation of CCR2-positive infiltrating immune cells in the DRG after formalin injection was reduced in *Snx25* +/- mice (**Revised Extended Data Fig. 9g**). We also found that at 3 days after formalin injection, the expression of a cluster of chemokines was lower in *Snx25* +/- mice and *Cx3cr1^{CreERT2/WT}; Snx25^{loxP/loxP}* mice than in control mice (**Revised Extended Data Fig. 9c and f**). Reduced macrophage accumulation in *Snx25* +/- mice and *Cx3cr1^{CreERT2/WT}; Snx25^{loxP/loxP}* mice may be caused by this decrease of chemokine expression. These immune phenotypes may be attributable to upregulation of TGF-beta receptor-1 (**Revised Extended Data Fig. 9d**), which is known to suppress immune responses and to be degraded by SNX25.

We described these infiltrating cells after formalin injection as follows:

We also found that at 3 d after formalin injection, the expression of a cluster of chemokines was lower in *Snx25* +/- mice than in WT mice (**Extended Data Fig. 9c**). Low macrophage accumulation in *Snx25* +/- mice may be due to this reduction of chemokine expression. These immune phenotypes may be attributable to upregulation of TGF-beta receptor-1 (**Extended Data Fig. 9d**), which is known to suppress immune responses²⁵ and to be degraded by SNX25¹⁸. (Page 11, lines 14–19)

The following figures were revised or added to the revised manuscript:

Extended Data Fig. 9c, d, f and g

Comment:

In Ext. Data Fig. 6a, many of the Iba1+ cells also highly express CD206. Are these CD206+ cells in the DRG replaced in their BM-chimeric mice and are they also targeted in the *Cx3cr1CreER* mice?

Response:

This is also an important point and we performed experiments to answer these questions. A BM transplantation experiment using GFP mice (C57BL/6-Tg (CAG-EGFP)) revealed that CD206-positive DRG macrophages were partly derived from bone marrow (**Figure E for reviewer, please see below**). We crossed $Cx3cr1^{CreERT2/WT}; Snx25^{loxP/loxP}$ mice with Ai32; $Snx25^{loxP/loxP}$ mice to generate triple TG mice ($Cx3cr1^{CreERT2/WT}; Snx25^{loxP/loxP}; Ai32/+$) to visualize cells with recombination by YFP fluorescence, and transplanted BM from triple TG mice into WT mice. YFP (Cx3cr1-expressing cells) expression was merged with CD206-positive cells in tamoxifen-treated triple TG mice, indicating that CD206-positive macrophages are targeted in the $Cx3cr1^{CreERT2/WT}$ mice (**Figure F for reviewer, please see below**).

Figure E for reviewer 3

Figure F for reviewer 3

Comment:

Throughout the manuscript, it would have been better to treat all groups with tamoxifen and take as the control group for example Cx3cr1CreER/WT mice instead of the same genotype +/- tamoxifen.

Response:

We thank the reviewer for this constructive comment. In response, we revised the behavioral data of von Frey tests (**Revised Fig. 4b**) and formalin tests (**Revised Fig. 4c** and **Revised Extended Data Fig. 5h**). We, however, kept the comparison between same genotype with and without tamoxifen treatment in some of the experiments (**Revised Fig. 4m** and **Revised Extended Data Fig. 10c** and **d**) because these experiments were initially planned to compare the conditions with and without tamoxifen treatment (**Fig. 4j** and **Extended Data Fig. 10a–b**) and were difficult to change.

The following figures were revised:

Fig. 4b

Fig. 4c

Extended Data Fig. 5h

Comment:

They sorted F4/80+CD11b+ cells (defined as macrophages) to check for expression of chemokines. They should also analyze whether these cells express *Snx25* (control mice) and verify the targeting efficiency, in particular in the Cx3cr1hiMHCIIhi nerve-associated macrophages.

Other myeloid cells including monocytes or Langerhans cells should be excluded and the two subsets of dermal macrophages distinguished by MHCII lo and hi.

Response:

Following the reviewer's suggestion, we sorted out dermal macrophages by FACS and investigated the expression level of *Snx25* compared with monocytes and dendritic cells in WT (control mice). **Revised Extended Data Fig. 11b** illustrates the steps of FACS sorting of the myeloid cells of WT mice. Briefly (details are described in the figure legend), cells were prepared by enzymatic digestion of saline-perfused mouse skin, followed by removal of dead cells and lineage depletion (T-cells, B-cells, NK cells, erythrocytes, granulocytes). The CD11b+/CD24 low population (CD24 antibody is used as a Langerhans cells marker) was further sorted by antibody staining for macrophage markers. The combination of Fc- γ receptor 1 (CD64) and MHC class II expression successfully differentiates dermal macrophages (Lin^{neg} CD45⁺ CD11b⁺ Ly6C⁻ CD64⁺ MHCII⁺), monocytes (Lin^{neg} CD45⁺ CD11b⁺ Ly6c⁺ CD64⁻ MHCII⁻), and dendritic cells (Lin^{neg} CD45⁺ CD11b⁺ Ly6C⁻ CD64⁻ MHCII⁺) (Tamoutounour et al., *Immunity* 2013, 39: 925–938, Kolter et al., *Immunity* 2019, 50: 1482–1497). We investigated the expression levels of *Snx25* and *Ngf* in dermal macrophages, as compared to other myeloid populations including monocytes and dendritic cells, of WT mice. The mRNA expression level of *Snx25* was lower in dendritic cells but not significantly different between dermal macrophages and monocytes (**Revised Extended Data Fig. 11c**).

We further verified the targeting efficiency in dermal macrophages (Lin^{neg} CD45⁺ CD11b⁺ Ly6c⁻ CD64⁺ MHCII⁺) of the *Cx3cr1*^{CreERT2/WT}; *Snx25*^{loxP/loxP} mice by FACS. Continuous feeding with TAM-containing chow markedly reduced the expression of *Snx25* in the dermal macrophages (**Revised Extended Data Fig. 11f**).

To verify the targeting efficiency in Cx3cr1 hi MHCII hi macrophages in other ways, we crossed *Cx3cr1*^{CreERT2/WT}; *Snx25*^{loxP/loxP} mice with Ai32; *Snx25*^{loxP/loxP} mice to generate triple TG

mice (*Cx3cr1^{CreERT2/WT}*; *Snx25^{loxP/loxP}*; Ai32/+) to visualize cells with recombination by YFP fluorescence. Unlike tissue-resident macrophages such as Langerhans cells, dermal macrophages in the skin have been shown to be derived from bone marrow and to turn over (Hoeffel et al., 2012. *J. Exp. Med.* 209: 1167–1181, Tamoutounour et al., 2013. *Immunity* 39: 925–938). Therefore, we next transplanted bone marrow from triple TG mice into WT mice (**Revised Extended Data Fig. 12a**). YFP expression was detected in dermal macrophages in TAM-treated triple TG mice (**Revised Fig. 5k, Revised Extended Data Fig. 12b**). YFP-expressing donor-derived *Cx3cr1^{hi}* and *MHCII^{hi}* macrophages were selectively collected from the skin of the WT recipient mice and the *Snx25* gene expression pattern was examined. Langerhans cells can be excluded by sorting YFP cells because they are embryonic in origin. As expected, *Snx25* was significantly decreased in YFP-positive macrophages compared to YFP-negative macrophages (**Revised Extended Data Fig. 12c**, 0.56-fold of control).

We described the FACS sorting and characterization of dermal macrophages and other myeloid cells in the Results section (Page 15, line 1 to Page 16, line 17).

The following figures were added to the revised manuscript:

Extended Data Fig. 11b–f

Extended Data Fig. 12a–c

Comment:

(Point 1) The authors claim that SNX25 and Ngf in dermal macrophages are required for pain sensation under normal conditions. While they show that macrophages can express SNX25, and that Ngf is reduced in the Clodronate injected area, they do not demonstrate that Ngf is indeed produced by dermal macrophages.

(Point 2) Ngf was reduced in BMDMs of *Snx25*^{+/-} mice or in total skin area of *Snx25*^{+/-} mice (Fig. 5A). This should be shown in dermal macrophages and that Ngf is decreased in dermal macrophages in *Snx25*^{+/-} mice.

(Point 3) Fig. 5C. Ngf seem to be expressed by many other cells. Which cells express it also?

(Point 4) In addition, is Ngf reduced in dermal macrophages of *Cxc3cr1CreERSnx25^{fl/fl}* mice?

(Point 5) What about microglia or macrophages in the DRG?

Response:

Point 1.

This is a very important point, and we thank the reviewer for raising it. Immunohistochemistry revealed that NGF was expressed in F4/80-, Iba1-positive cells in addition to MHCII-positive cells, confirming that NGF is produced by dermal macrophages (**Revised Fig. 5c**).

The following figure was revised:

Fig. 5c

Point 2.

In accordance with the reviewer's suggestion, we performed co-staining of the NGF+ cells with a macrophage marker, Iba1, in dermal macrophages of hind paw skin of WT and *Snx25* +/- mice and confirmed that NGF expression level was lower in *Snx25* +/- mice compared to WT mice (**Figure G for reviewer 3, please see below**). To further investigate the relationship between SNX25 and NGF in dermal macrophages, we also sorted dermal macrophages (Lin^{neg} CD45⁺ CD11b⁺ Ly6c⁻ CD64⁺ MHCII⁺) by FACS from WT and *Snx25* +/- mice and analyzed the expression of *Ngf* by qPCR. We found significant differences in *Ngf* expression level of dermal macrophages between WT and *Snx25* +/- mice. This strongly suggests that *Snx25* regulates *Ngf* expression in dermal macrophages (**Revised Extended Data Fig. 11e**).

The FACS sorting of dermal macrophages and NGF expression study are described in the Results section (Page 15, line 13 to Page 16, line 10).

The following figure was added to the revised manuscript:

Extended Data Fig. 11e

Figure G for reviewer 3

Point 3.

Immunohistochemistry revealed that NGF was also expressed in CD117-positive mast cells in addition to dermal macrophages (**Revised Fig. 5c**).

The following figure was revised:

Fig. 5c

Point 4.

We performed Western blot analysis to confirm the expression of NGF at the protein level in TAM-administered *Cx3cr1^{CreERT2/WT}*; *Snx25^{loxP/loxP}* mice (**Revised Extended Data Fig. 11a**). NGF expression of hind paw skin in cKO mice tended to be lower than in *Snx25^{loxP/loxP}* mice ($p = 0.059$). Furthermore, we sorted dermal macrophages by FACS, and found that SNX25 depletion in dermal macrophages decreased the expression of *Ngf* ($p = 0.057$) (**Revised Extended Data Fig. 11f**). These results indicate that SNX25 in dermal macrophages contributes to *Ngf* expression.

The FACS sorting and NGF expression in the dermal macrophages are described in the Results section (Page 15, line 13 to Page 16, line 10).

The following figures were added to the revised manuscript:

Extended Data Fig. 11a and f

Point 5.

We agree that it is of interest to investigate whether SNX25 regulates NGF expression in DRG macrophages or microglia in the central nervous system. We first performed double-labeling immunohistochemistry (NGF and Iba1) in the DRG as well as in the hind paw skin. We hardly detected NGF immunoreactivity in Iba1-positive macrophages in the DRG (**Revised Extended Data Fig. 14k**). We next compared NGF expression in the hind paw skin, the DRG, and the spinal cord of the same individual mice with Western blotting (**reiterated Figure C for reviewer 3, please see below**). Western blot analysis revealed that NGF was strongly expressed in the hind paw skin and was much more weakly expressed in the DRG and spinal cord, suggesting that DRG macrophages and spinal cord microglia have a different mechanism of NGF expression from peripheral tissues (especially from dermal macrophages).

We added a description of these results as follows:

Indeed, NGF expression was low in Iba1+ DRG macrophages under normal conditions (**Extended Data Fig. 14k**), as was SNX25 expression (**Extended Data Fig. 14b**). This suggests that DRG macrophages have a different mechanism of NGF expression and pain conduction from that of dermal macrophages. (Page 20, lines 6–10.)

The following figure was added to the revised manuscript:

Extended Data Fig. 14b and k

Figure C for reviewer 3

Comment:

- Figure 5E, this image is not convincing. Trka staining is very weak.

Response:

A magnified image is now presented in which the fluorescence signal can be clearly seen (**Revised Fig. 5e**).

The following figure was revised:

Fig. 5e

Comment:

- Ext. Fig. 1a, they describe a reduction of c-Fos⁺ cells, this is not evident in the figure. Can they explain what c-Fos⁺ cells are?

Response:

We are sorry for the lack of a detailed explanation. c-Fos is the product of an immediate early gene and is frequently used as a marker for activated neurons (Lima et al., 1993, *Neuroreport*. 4: 747–750). We also confirmed that c-Fos merges with NeuN (neuron marker) and does not merge with either GFAP (astrocyte marker) or Iba1 (microglia marker) (**Revised Extended Data Fig. 1b**). Therefore, Extended Data Fig. 1a in the original manuscript indicates that the nerve activation (i.e., conduction of pain sensation) after formalin injection is attenuated in the TG mice compared to the WT mice.

The following figure was added to the revised manuscript:

Extended Data Fig. 1b

Comment:

- Fig. 21, the figure legend is not clear. What is the light red group and the light blue group?

Response:

We apologize for the incomprehensible figure. The light blue and the light red represented individual data points. We revised the figure to show only the average values for *Snx25*^{+/+} and *Snx25*^{+/-} DRG neurons. Furthermore, we reviewed the data and noticed a serious mistake: the concentration that we thought to be 10 μ M capsaicin was actually 1 μ M. The description of the concentration was corrected in the revised figure (**Revised Fig. 21**).

The following figure was revised:

Fig. 21

Comment:

- For all the immunohistochemistry images, single stains should be shown along the merged file. And images quantified.

Response:

All of the immunostaining images now comprise singly stained and merged panels. Furthermore, images that needed to be compared by mouse genotype or drug administration were quantified by calculating fluorescence intensity.

The following figures were revised or added to the revised manuscript:

Fig. 3a, d, e

Fig. 5c, h, k

Fig. 7m

Extended Data Fig. 1b

Extended Data Fig. 2d, e

Extended Data Fig. 3d

Extended Data Fig. 6b, d

Extended Data Fig. 7b-d

Extended Data Fig. 8a-f

Extended Data Fig. 9a, g

Extended Data Fig. 14a-f, i, k

Comment:

- Figure legends lack detailed information. For example, which method was used in Fig. 4e? or is Ext. Data Fig. 6g. expression levels of sorted cells or total skin and was it assessed by RT PCR?

Response:

We are sorry for the lack of explanation about methods. We assessed the mRNA expression levels of total hind paw skin or sorted cells by qRT-PCR in **Fig. 4e** and original Extended Data Fig. 6g (**Revised Extended Data Fig. 10c**). qRT-PCR was calculated by the $\Delta\Delta C_t$ method (Livak and Schmittgen. *Methods*, 2001, 25: 402-408). We added detailed information to the figure legend and Methods section (Page 48, line 5–page 48, line 10).

Decision Letter, first revision:

Subject: Decision on Nature Immunology submission NI-A32798A

Message: 14th Jul 2022

Dear Dr. Tanaka,

Thank you for your response to the reviewers' comments on your article "Dermal macrophages set pain sensitivity by modulating tissue NGF levels through SNX25–Nrf2 signaling". Although we are interested in the possibility of publishing your study in *Nature Immunology*, the issues raised by the referees need to be addressed.

Please revise along the lines specified in your letter. At resubmission, please include a "Response to referees" detailing, point-by-point, how you addressed each referee comment. If no action was taken to address a point, you must provide a compelling argument. This response will be sent back to the referees along with the revised manuscript.

Please include a revised version of any required reporting checklist. It will be available to referees to aid in their evaluation.

Reporting summary:

When submitting the revised version of your manuscript, please pay close attention to our [href="https://www.nature.com/nature-portfolio/editorial-policies/image-integrity">Digital Image Integrity Guidelines. and to the following points below:](https://www.nature.com/nature-portfolio/editorial-policies/image-integrity)

[REDACTED]

Note: This URL links to your confidential home page and associated information about manuscripts you may have submitted, or that you are reviewing for us.

If you wish to forward this email to co-authors, please delete the link to your homepage.

We hope to receive your revised manuscript within three months. If you cannot send it within this time, please let us know. We will be happy to consider your revision so long as nothing similar has been accepted for publication at Nature Immunology or published elsewhere.

Nature Immunology is committed to improving transparency in authorship. As part of our efforts in this direction, we are now requesting that all authors identified as 'corresponding author' on published papers create and link their Open Researcher and Contributor Identifier (ORCID) with their account on the Manuscript Tracking System (MTS), prior to acceptance. ORCID helps the scientific community achieve unambiguous attribution of all scholarly contributions. You can create and link your ORCID from the home page of the MTS by clicking on 'Modify my Springer Nature account'. For more information please visit www.springernature.com/orcid.

Sincerely,

Ioana Visan, Ph.D.
Senior Editor
Nature Immunology

Tel: 212-726-9207
Fax: 212-696-9752
www.nature.com/ni

Reviewers' Comments:

Reviewer #1:

Remarks to the Author:

The authors have adequately addressed almost all the issues raised.

Reviewer #2:

Remarks to the Author:

This resubmission is a much-improved version over the original study.

The authors have addressed most of the queries appropriately. However, there are two points that still require attention.

1. The authors must perform additional TrkA measurements or, alternatively, tune down

their incorrect statements referring to statistically non-significant TrkA levels (Fig.2k).

2. BM mixed chimaeras were not performed following gold-standards. In order to establish cell-autonomous effects, the authors must generate BM mixed chimeras against a third-part WT reference (competitor). Thus, Snx25^{+/-} BM must be mixed with a WT reference competitor and transplanted into host mice (group 1). In parallel, Snx25^{+/+} (littermate of Snx25^{+/-}) BM is to be mixed with the same WT reference competitor and transplanted into host mice (group 2).

Only this experimental setting can formally demonstrate whether Snx25 controls macrophage Ngf expression in a cell-autonomous manner.

Reviewer #3:

Remarks to the Author:

The data in the manuscript are very convoluted. They show a lot of Immunohistochemistry images but proper quantifications are often lacking throughout the manuscript or were performed by western blot of a total tissue but not of the cells of interest.

In addition, in the title and abstract, the authors conclude that dermal macrophages set pain sensitivity. However, it could also be that DRG macrophages contribute to the observed phenotype in Cx3cr1CreER Snx25^{fl/fl} and SNX25^{+/-} mice. This would of course also be interesting, but would have to be mentioned.

Also, they have not in detail analyzed the resident macrophages (phenotypically, transcriptionally, locally) or the 'inflammatory' monocyte-derived macrophages in the inflammatory model.

They have addressed some of the points raised but I still have several comments:

Specific comments

In the new Ext. Data Fig. 6b, they show co-staining of SNX25 with other immune cell markers and they suggest that it is also expressed by other myeloid cells (neutrophils, monocytes, mast cells) but not as high as in macrophages. If they would like to make a statement about the levels of SNX25 expression or the percentage of cells expressing it, they should quantify this, or show it for example by gene expression.

In Ext. Data Fig. 14b, they assessed whether DRG macrophages also express SNX25. In the first row, F4/80 and SNX25 seem to have a lot of background staining and it appears that SNX25 is not expressed in F4/80+ cells. In the second row, they show MHCII+ cells, which is not specific for macrophages, in the 3rd row, they show a cell clearly positive for CD206 and SNX25, suggesting that macrophages in the DRG also express SNX25. Again, this would need to be quantified. In particular, as they claim here that DRG macrophages express SNX25 to a lesser extent than dermal macrophages, which is not clear in the one image they show. This would need to be quantified side by side with the skin.

They also show that 4-OHT injection into the DRG of Cx3cr1CreER Snx25^{fl/fl} mice did not lead to a pain-insensitive phenotype, suggesting that DRG macrophages don't play a role. To confirm this, deletion of SNX25 in DRG macrophages would need to be confirmed. Why

not use the anti-SNX25 antibody they used for their immunohistochemistry stainings to show its absence in the Cre+ animals? As a control for this experiment, 4-OHT would need to be injected into the DRG of control mice.

Altogether, it is still not clear whether they also target DRG macrophages in the Cx3cr1CreER SNX25fl/fl mice and whether they contribute to the pain-insensitive phenotype.

The authors state that 'a population of dermal macrophages (positive for MHCII, CD206, and F4/80) was closely associated with PGP9.5-positive sensory fibers and was SNX25-immunoreactive, compared with other myeloid cells. In Ext. Data Fig. 6b, also Gr1+, CD4+ and CD117+ cells seem to express SNX25. Again, this would also need to be quantified (how many of the respective cells express SNX25?).

The authors assume that YFP expression in Cx3cr1CreER SNX25fl/fl Ai39 mice correlates with deletion of SNX25. This is not necessarily the case, as it is a different locus to be recombined and might not have the same recombination efficiency as the reporter.

Ext. Fig. 7e-f, I do not understand these experiments. In f) they show for example that only 10-30% of macrophages (CD206+ or F4/80+ of Lyve1+) are GFP+, which would mean that the dermal macrophages in their setting are actually not replaced. Even more confusing is their statement that 'macrophages are constantly replaced by BM-derived cells, whereas neutrophils, B cells, T cells, NK cells are stable in naïve condition'. Are they stating that neutrophils, lymphocytes and NK cells are not replaced by BM-derived cells?

In Ext. Data Fig. 9a,b, the authors analyzed SNX25+/- hind paw skin and DRG 7d after formalin injection. Accumulation of macrophages was reduced in SNX25+/- mice in the DRG but not in the hind paw. Additionally, a difference was noticed in infiltrating Ccr2+ cells in the DRG between WT and SNX25+/- mice after formalin injection. This was not quantified and just one image shown. Why did they not analyze also the hindpaw for infiltrating monocytes?

Altogether, this all points towards a phenotype in DRG macrophages/infiltrating cells rather than only macrophages in the hindpaw.

Figure 3 for reviewer 3, they show Ngf expression by western blot of DRG, hindpaw and spinal cord. Total Ngf in different tissues is difficult to compare.

Ext. Fig. 11c, monocytes appear to have the highest expression of SNX25 and Ngf, it would be nice to compare this expression in the SNX25+/- and Cx3cr1CreER Snx25fl/fl mice, in particular as they claim that the observed phenotype is due to specific SNX25 deletion in dermal macrophages.

Author Rebuttal, first revision:

Responses to the Reviewers' Comments

Reviewer #1

Comment:

Remarks to the Author:

The authors have adequately addressed almost all the issues raised.

Response:

We thank the reviewer for the constructive comments that improved our manuscript.

Reviewer #2

Comment:

Remarks to the Author:

This resubmission is a much-improved version over the original study.

The authors have addressed most of the queries appropriately. However, there are two points that still require attention.

Response:

We appreciate the reviewer's positive remarks about the revised manuscript. The two points raised are both important, and we performed additional experiments as described below.

Comment:

1. The authors must perform additional TrkA measurements or, alternatively, tune down their incorrect statements referring to statistically non-significant TrkA levels (Fig.2k).

Response:

In response to the reviewer's request, immunostaining experiments for TrkA in DRGs of *Snx25* $+/+$ and *Snx25* $+/-$ mice were performed and quantified (*Snx25* $+/+$, $n=13$; *Snx25* $+/-$, $n=12$). Quantitative analysis revealed that TrkA expression in the DRG was significantly lower in *Snx25* $+/-$ mice. We have left the text unchanged but modified **Fig. 2k** in the re-revised version.

The following figure was revised:

Fig. 2k
Comment:

2. BM mixed chimaeras were not performed following gold-standards. In order to establish cell-autonomous effects, the authors must generate BM mixed chimeras against a third-part WT reference (competitor). Thus, *Snx25*^{+/-} BM must be mixed with a WT reference competitor and transplanted into host mice (group 1). In parallel, *Snx25*^{+/+} (littermate of *Snx25*^{+/-}) BM is to be mixed with the same WT reference competitor and transplanted into host mice (group 2).

Only this experimental setting can formally demonstrate whether *Snx25* controls macrophage *Ngf* expression in a cell-autonomous manner.

Response:

This is a very important point, and we thank the reviewer for the invaluable suggestion. Following the Reviewer's instructions, we transplanted mixed bone marrow from *Snx25*^{+/-} mice on the CD45.2 background (CD45.2⁺) and from congenic WT mice on the CD45.1 background (CD45.1⁺) at a 1:1 ratio into CD45.1/CD45.2 mice (Group 1). We also made the Group 2 (mixed bone marrow from *Snx25*^{+/+} littermates (CD45.2⁺) and from WT mice (CD45.1⁺) transplanted into CD45.1/CD45.2 mice). After BM transplantation, dermal myeloid cells including macrophages were selectively collected from the skin of the mixed BM chimeric mice by FACS using FVS700, CD11b, CD45.1, CD45.2 and lineage (CD3, CD19, NK1.1, TER119, Ly6G) marker expression. We then examined whether *Snx25* and *Ngf* expressions are reduced in the CD45.2⁺ dermal myeloid cell population (CD45.1(-) CD45.2(+)) from each Group (**Revised Extended Data Fig. 16b** and **c**). Donor BM-derived *Snx25* heterozygous myeloid cells in Group 1 mice expressed lower levels of *Snx25* and *Ngf* compared to the donor BM-derived *Snx25*^{+/+} myeloid cells in Group 2 mice although the differences did not quite reach the threshold of significance (n = 3, **Revised Extended Data Fig. 16c**). We also examined whether *Snx25* and *Ngf* expressions are reduced in the FACS-isolated CD45.1(-) CD45.2(+) cell population compared to CD45.1(+) CD45.2(-) cell population in Group 1 mice (**Revised Extended Data Fig. 16b** and **d**). Donor BM-derived *Snx25* heterozygous myeloid cells tended to express lower levels of *Ngf* compared to the donor BM-derived WT myeloid cells (n = 3, **Revised Extended Data Fig. 16d**). From these data, we were not able to formally conclude the cell-autonomous effect of *Snx25* on the *Ngf* synthesis although *Snx25* heterozygous myeloid cells tended to express lower levels of *Ngf*. Therefore, we changed “the cell-autonomous impact of SNX25” in the original manuscript (**Page 15, line 2–3**) to “the impact of SNX25” in the re-revised manuscript. We have added new experimental data in **Extended Data Fig. 16** in the re-revised version.

We added a new section describing mixed BM competition assay (**Page 16, line 20–page 17, line 7**).

The following figure was revised:

Extended Data Fig. 16a–d

Reviewer #3*Comment:*

Remarks to the Author:

The data in the manuscript are very convoluted. They show a lot of Immunohistochemistry images but proper quantifications are often lacking throughout the manuscript or were performed by western blot of a total tissue but not of the cells of interest.

In addition, in the title and abstract, the authors conclude that dermal macrophages set pain sensitivity. However, it could also be that DRG macrophages contribute to the observed phenotype in Cx3cr1CreER Snx25fl/fl and SNX25^{+/-}-mice. This would of course also be interesting, but would have to be mentioned.

Also, they have not in detail analyzed the resident macrophages (phenotypically, transcriptionally, locally) or the ‘inflammatory’ monocyte-derived macrophages in the inflammatory model.

They have addressed some of the points raised but I still have several comments:

Response:

We accept that our revised version lacked quantitative analyses in the experiments pointed out by the reviewer and we have now complemented these in the re-revised version. The reviewer also suggests an involvement of DRG macrophages in the pain-less phenotype, which we do not deny in the present study. We consider that DRG macrophages and dermal macrophages are not mutually exclusive in pain sensation, but rather may cooperate – possibly by different mechanisms. We describe DRG macrophages in the Abstract and the Discussion, based on the additional data obtained in our responses to the specific comments below. We also present a detailed analysis of resident macrophages and monocyte-derived macrophages in inflammatory models in the re-revised version.

We have revised the last sentence of the Abstract as follows:

“We conclude that dermal macrophages set pain sensitivity by producing and secreting NGF into the dermis in addition to their host defense functions, and may cooperate with dorsal root ganglion macrophages in pain perception.” (Page 2, line 11–14)

We have also revised the second paragraph of the Discussion as follows:

“Notably, *Snx25* +/- mice or *Cx3cr1*^{CreERT2/WT}; *Snx25*^{loxP/loxP} mice were insensitive to mechanical stimuli in normal conditions without any neuropathic or inflammatory paradigm (Fig. 2c, Fig. 4b). Interestingly, our preliminary data revealed that these mice exhibited reduced responses to innocuous low-pressure stimuli^{43, 44} (Extended Data Fig. 21a and b), suggesting that SNX25 in dermal macrophages is involved in innocuous tactile sensing in addition to mechanical pain. This aspect is now under investigation. We concluded that SNX25 in dermal macrophages is a pivotal factor for pain sensation under normal and painful conditions. However, recent work has shown that macrophages in the DRG mediate neuropathic pain³⁷. Considering that the expression levels of both SNX25 and NGF were lower in DRG (Extended Data Fig. 18b, Extended Data Fig. 20) than in skin, dermal macrophages and DRG macrophages may contribute to pain via different mechanisms.” (Page 22, line 22–page 23, line 9.)

We would like to keep the original title because it succinctly describes the points of the present study; adding DRG macrophages into the title would make it long and unfocused. Instead, we clearly mention the DRG macrophages in both the Abstract and the Discussion, as quoted above.

Specific comments

Comment:

In the new Ext. Data Fig. 6b, they show co-staining of SNX25 with other immune cell markers and they suggest that it is also expressed by other myeloid cells (neutrophils, monocytes, mast cells) but not as high as in macrophages. If they would like to make a statement about the levels of SNX25 expression or the percentage of cells expressing it, they should quantify this, or show it for example by gene expression.

Response:

We thank the reviewer for this comment, in response to which we performed and quantified further immunostaining experiments for other immune cell markers in hind paw skin. SNX25 was found to be expressed in most macrophages (90.1% of CD206+ cells were SNX25+; 84.1% of F4/80+ cells were SNX25+). It was also expressed in the majority of Gr1+ neutrophils (67.2%). On the other hand, SNX25 expression was found to be less common (17.3–29.5%) in the remaining immune cells (CD117+, CD4+, CD8+, CD19+, and NK1.1+ cells). These data consolidated our view of the importance of SNX25 in macrophages and we thank the reviewer for the constructive comment.

The following figure was revised:

Extended Data Fig. 6a, b

Comment:

In Ext. Data Fig. 14b, they assessed whether DRG macrophages also express SNX25.

In the first row, F4/80 and SNX25 seem to have a lot of background staining and it appears that SNX25 is not expressed in F4/80+ cells. In the second row, they show MHCII+ cells, which is

not specific for macrophages, in the 3rd row, they show a cell clearly positive for CD206 and SNX25, suggesting that macrophages in the DRG also express SNX25. Again, this would need to be quantified. In particular, as they claim here that DRG macrophages express SNX25 to a lesser extent than dermal macrophages, which is not clear in the one image they show. This would need to be quantified side by side with the skin.

Response:

We apologize for the low-quality image with high background. We also acknowledge that MHCII is not specific for macrophages. Taking account of these comments, we undertook additional immunostaining experiments for macrophage markers (CD206 and F4/80) in DRGs. Unlike in paw skin, SNX25 expression level was so low that it was only detected by enhancing the signal with a TSA (tyramide signal amplification) Plus kit. The immunostaining revealed differences between paw skin and DRG: considering that SNX25 was detected without signal enhancement in hind paw skin, the expression level of SNX25 itself is evidently low in DRG. We detected SNX25 in DRG macrophages, but only in a small fraction of macrophages (13% of CD206+ cells and 15% of F4/80+ cells) (**Revised Extended Data Fig. 18b**). This is in clear contrast to the immunostaining in dermal macrophages (90.1% of CD206+ cells and 84.1% of F4/80+ cells; **Revised Extended Data Fig. 6**, see above). We also detected SNX25-positive DRG neurons with the enhanced immunostaining. We have previously reported that SNX25 is strongly expressed in CNS neurons (Morita et al., *Brain Struct. Funct.* 225: 2615–2642, 2020). In the present study, we confirmed that SNX25 in the DRG neurons has nothing to do with pain sensation by means of a DRG neuron-specific conditional knockout experiment (**Extended Data Fig. 5f–h**). We would like to elucidate what functions SNX25 may have in DRG neurons in future studies.

The following text was added to result section:

“Some DRG macrophages associate with nerves (**Extended Data Fig. 18a**) and these macrophages express SNX25 at the low-moderate level (**Extended Data Fig. 18b**).” (**Page19, line24–page20, line2**)

The following figure was revised:

Extended Data Fig. 18b

Comment:

They also show that 4-OHT injection into the DRG of *Cx3cr1CreER Snx25fl/fl* mice did not lead to a pain-insensitive phenotype, suggesting that DRG macrophages don't play a role. To confirm this, deletion of SNX25 in DRG macrophages would need to be confirmed. Why not use the anti-SNX25 antibody they used for their immunohistochemistry stainings to show its absence in the Cre⁺ animals? As a control for this experiment, 4-OHT would need to be injected into the DRG of control mice.

Altogether, it is still not clear whether they also target DRG macrophages in the *Cx3cr1CreER SNX25fl/fl* mice and whether they contribute to the pain-insensitive phenotype.

Response:

Following the reviewer's suggestion, we performed immunostaining to investigate whether SNX25 is attenuated in DRG macrophages after injecting 4-OHT into the DRG of triple TG mice (*Cx3cr1^{CreERT2/WT}; Snx25^{loxP/loxP}; Ai32/+*). We also performed a control experiment in which 4-OHT was injected into the DRG of control mice (*Snx25^{loxP/loxP}; Ai32/+*, littermates of triple TG mice). As mentioned in the response to the above comment, SNX25 expression is very low in DRG and the signal was amplified with TSA for detection. In the control experiment, we

identified on SNX25- and CD206-double-positive macrophages and measured the fluorescence intensity of SNX25 in those cells with ImageJ software. In the triple TG mice, we focused on YFP- and CD206-double-positive macrophages and measured the fluorescence intensity of SNX25 in those cells in the same way. Quantitative analysis showed that SNX25 expressed in DRG macrophages was significantly downregulated in triple TG mice (**Revised Extended Data Fig. 19b**, see below). This indicates that SNX25 in DRG macrophages was deleted by 4-OHT treatment.

We added a new section describing SNX25 expression in DRG macrophages after injecting 4-OHT into the DRG of triple TG mice (*Cx3cr1^{CreERT2/WT}; Snx25^{loxP/loxP}; Ai32/+*). (**Page 20, line 12–20**)

The following figure was revised:

Extended Data Fig. 19b

Comment:

The authors state that ‘a population of dermal macrophages (positive for MHCII, CD206, and F4/80) was closely associated with PGP9.5-positive sensory fibers and was SNX25-immunoreactive, compared with other myeloid cells. In Ext. Data Fig. 6b, also Gr1+, CD4+ and CD117+ cells seem to express SNX25. Again, this would also need to be quantified (how many of the respective cells express SNX25?).

Response:

First, we carefully reexamined the extent to which myeloid cells, including macrophages, associate with sensory fibers by measuring the distance between the nearest PGP9.5+ fiber and each cell (100 μm was used as the cutoff value). We found that macrophages were in close proximity to PGP9.5+ fibers (**Revised Extended Data Fig. 7a**). Other myeloid cells were further from fibers, although a few neutrophils (Gr1+) and mast cells (CD117+) localized close to a fiber (**Revised Extended Data Fig. 7a**). Based on these results, we next investigated the expression of SNX25 in macrophages, neutrophils, and mast cells that localize near PGP9.5+ fibers (< 20 μm). SNX25 expression was significantly higher in macrophages that contacted or were close to PGP9.5+ fibers than in neutrophils and mast cells that were comparably close to fibers (**Revised Extended Data Fig. 7b**).

We have added these data as supplementary data to show that dermal macrophages that associate with nerves are important for pain sensation.

The following text was added to result section:

“We confirmed that a population of dermal macrophages (positive for MHCII, CD206, and F4/80) was closely associated with PGP9.5-positive sensory fibers and was SNX25-immunoreactive (**Fig. 3a, Extended Data Fig. 6a**) compared with other myeloid cells (**Extended Data Fig. 6b, Extended Data Fig. 7**).” (**Page 9, line 10–13**)

The following figure was revised:

Extended Data Fig.7a, b

Comment:

The authors assume that YFP expression in Cx3cr1CreER SNX25^{fl/fl} Ai39 mice correlates with deletion of SNX25. This is not necessarily the case, as it is a different locus to be recombined and might not have the same recombination efficiency as the reporter.

Response:

We thank the reviewer for this important comment, with which we fully agree. After treating triple TG mice with 4-OHT, we examined in detail the deletion rate of SNX25 in YFP-expressing cells: in paw skin, SNX25 was deleted in 97% of YFP+ cells (arrowheads in **Revised Extended Data Fig. 17a** and green bar in **Revised Extended Data Fig. 17b**) and only 3% of YFP+ cells expressed SNX25 (arrows in **Revised Extended Data Fig. 17a** and yellow bar in **Revised Extended Data Fig. 17b**). A similar analysis in DRGs showed that SNX25 was absent in 93% of YFP+ cells (**Revised Extended Data Fig. 19a**). These data strongly suggest that Cre recombinase in the cells is active enough to simultaneously induce YFP expression and to delete the *Snx25* gene, even though the YFP and floxed *Snx25* genes are not necessarily on the same locus.

The following text was added to result section:

“After treatment of triple TG mice with 4-OHT, the deletion rate of SNX25 in YFP-expressing cells was examined: in paw skin, SNX25 was deleted in 97% of YFP+ cells (**Extended Data Fig. 17**).” (Page 19, line 17–19)

The following text describing deletion of SNX25 in DRG was also added to result section:

“After treatment of triple TG mice with 4-OHT, 91% of YFP+ cells were SNX25-negative (**Extended Data Fig. 19a**).” (Page 20, line 12–13)

The following figure was revised:

Extended Data Fig. 17a, b

Extended Data Fig. 19a

**Comment:**

Ext. Fig. 7e-f, I do not understand these experiments. In f) they show for example that only 10-30% of macrophages (CD206+ or F4/80+ of Lyve1+) are GFP+, which would mean that the dermal macrophages in their setting are actually not replaced. Even more confusing is their statement that ‘macrophages are constantly replaced by BM-derived cells, whereas neutrophils, B cells, T cells, NK cells are stable in naïve condition’. Are they stating that neutrophils, lymphocytes and NK cells are not replaced by BM-derived cells?

Response:

We apologize for the incomprehensible statement. The reviewer may be concerned that the replacement ratio is very low for the dermal macrophages. The replacement of 10–30% of dermal macrophages by BM-derived cells is consistent with a previous study (Tamoutournor et al., *Immunity* 39:925–938, 2013) and our FACS data show that 38% of GFP+ cells are F4/80+ and 39% of GFP+ cells are MHCII+ in hind paw skin (**Fig. 3f**). Ten to 40% replacement can be effective because dermal macrophages are closely associated with peripheral nerves. In contrast, other immune cells (Gr1, CD19, CD8a, CD4, and NK1.1+ cells) are replaced in the range of 0.3 to 1% (**Extended Data Fig. 10g**). Similar results were obtained for the dermal macrophages of

back skin (**Extended Data Fig. 10h**). However, we also think that the statement “macrophages are constantly replaced by BM-derived cells, whereas neutrophils, B cells, T cells, NK cells are stable in naïve condition” is an overstatement. Since our experimental data do not allow us to convince readers that neutrophils, B cells, T cells, and NK cells are stable in the naïve condition, we simply state that "only a few GFP-positive Gr1, CD19, CD8a, CD4, and NK1.1+ cells were detected in the naïve condition" (**Page 10, line 10–12**) in the re-revised manuscript.

Fig. 3f

Extended Data Fig. 10g

Extended Data Fig. 10h

Comment:

In Ext. Data Fig. 9a,b, the authors analyzed SNX25^{+/-} hind paw skin and DRG 7d after formalin injection. Accumulation of macrophages was reduced in SNX25^{+/-} mice in the DRG but not in the hind paw. Additionally, a difference was noticed in infiltrating Ccr2⁺ cells in the DRG between WT and SNX25^{+/-} mice after formalin injection. This was not quantified and just one image shown. Why did they not analyze also the hindpaw for infiltrating monocytes?

Altogether, this all points towards a phenotype in DRG macrophages/infiltrating cells rather than only macrophages in the hindpaw.

Response:

In the revised version, for the immunostaining of CD206 at 7 days after formalin injection, n = 4–5 was used for the DRG, but only n = 3 for hind paw skin. To respond to the reviewer's first comment, we increased the latter sample number to n = 8. CD206 quantification in hind paw skin showed a decrease in *Snx25* ^{+/-} mice, although the difference was marginally less than significant (**Revised Extended Data Fig. 11b**).

In response to the second comment, we examined the infiltration of CCR2⁺ cells after 5% formalin injections into the paw skin and DRGs with immunohistochemistry and quantified the observations (**Revised Extended Data Fig. 11g, h**). The results showed fewer CCR2⁺

infiltrating monocytes in *Snx25* +/- mice in both paw skin and DRG, although again the differences did not quite reach the threshold of significance. Overall, SNX25 deletion attenuates tissue infiltration of macrophage/monocyte lineage cells in response to inflammation. Reduced number of the DRG macrophages in the *Snx25* +/- mice may play an additive role to that of dermal macrophages in dull response in the inflammation paradigm. We would like to address this issue in our future studies.

The following text was added to result section:

“Immunohistochemistry also revealed that the accumulation of CCR2-positive infiltrating immune cells in the hind paw skin and DRG after formalin injection was lower, albeit not significantly so, in *Snx25* +/- mice (**Extended Data Fig. 11g and h**)”. (**Page 12, line 10–12**)

The following text was also added to legend of **Extended Data Fig. 11b**:

“SNX25 deletion attenuates tissue infiltration of macrophages in response to inflammation. Reduced number of the DRG macrophages in the *Snx25* +/- mice may play an additive role to that of dermal macrophages in dull response in the inflammation paradigm.”

The following figure was revised:

Extended Data Fig. 11b, g, h

Comment:

Figure 3 for reviewer 3, they show Ngf expression by western blot of DRG, hindpaw and spinal cord. Total Ngf in different tissues is difficult to compare.

Response:

We agree with the reviewer on this point, and we have omitted the comparison of total NGF detected by Western blot from the re-revised manuscript. We performed immunostaining of NGF in DRG macrophages and spinal cord, in addition to hind paw skin, and quantified the results.

NGF expression was high in hind paw skin and was expressed in most macrophages (86% of F4/80+ cells), whereas expression was low in DRG and spinal cord, suggesting that DRG macrophages and spinal cord microglia have a mechanism of NGF expression that differs from peripheral tissues (especially from dermal macrophages).

The following text was added to result section:

“These data indicate that SNX25 in dermal macrophages, but not in DRG macrophages, is a pivotal factor for pain sensation under normal conditions. This may be due to tissue-specific differences in SNX25 and NGF expression. Indeed, NGF expression was low in F4/80+ DRG macrophages under normal conditions (**Extended Data Fig. 20**), as was SNX25 expression (**Extended Data Fig. 18b**).” (Page 20, line 20–24)

The following figure was revised:

Extended Data Fig. 20a, b

Comment:

Ext. Fig. 11c, monocytes appear to have the highest expression of SNX25 and *Ngf*, it would be nice to compare this expression in the SNX25^{+/-} and *Cx3cr1*CreER *Snx25*^{fl/fl} mice, in particular as they claim that the observed phenotype is due to specific SNX25 deletion in dermal macrophages.

Response:

We thank the reviewer for this important suggestion. We consider it crucial to examine NGF expression in monocytes and dendritic cells in establishing that SNX25 in dermal macrophages is important for NGF production and the pain phenotype. We investigated the expression level of *Snx25* and *Ngf* in dermal monocytes ($\text{Lin}^{\text{neg}} \text{CD45}^+ \text{CD11b}^+ \text{Ly6C}^+ \text{CD64}^- \text{MHCII}^-$) and dendritic cells ($\text{Lin}^{\text{neg}} \text{CD45}^+ \text{CD11b}^+ \text{Ly6C}^- \text{CD64}^- \text{MHCII}^+$) of *Snx25*^{+/-} and *Cx3cr1*^{CreERT2/WT}; *Snx25*^{loxP/loxP} mice. *Ngf* mRNA level was not significantly different in monocytes and dendritic cells of both *Snx25*^{+/-} and *Cx3cr1*^{CreERT2/WT}; *Snx25*^{loxP/loxP} mice as compared to control littermate mice (Revised Extended Data Fig. 14). These results indicate that *Snx25* is important for *Ngf* production in dermal macrophages, but not in monocytes or dendritic cells.

The following text was added to result section:

“SNX25 depletion decreased the expression of *Ngf* in dermal macrophages (**Extended Data Fig. 13e and f**), but not in dermal monocytes or dendritic cells (**Extended Data Fig. 14**).” (**Page 16, line 9-11**)

The following figure was revised:

Extended Data Fig. 14a–d

Decision Letter, second revision:

Subject: Your manuscript, NI-A32798B

Message: Our ref: NI-A32798B

5th Dec 2022

Dear Dr. Tanaka,

Thank you for your patience as we've prepared the guidelines for final submission of your Nature Immunology manuscript, "Dermal macrophages set pain sensitivity by modulating tissue NGF levels through SNX25–Nrf2 signaling" (NI-A32798B). Please carefully follow the step-by-step instructions provided in the attached file, and add a response in each row of the table to indicate the changes that you have made. Please also check and comment on any additional marked-up edits we have proposed within the text. Ensuring that each point is addressed will help to ensure that your revised manuscript can be swiftly handed

over to our production team.

We would like to start working on your revised paper, with all of the requested files and forms, as soon as possible (preferably by Wednesday, December 14th). Please get in contact with us if you anticipate delays.

When you upload your final materials, please include a point-by-point response to any remaining reviewer comments and please make sure to upload your checklist.

If you have not done so already, please alert us to any related manuscripts from your group that are under consideration or in press at other journals, or are being written up for submission to other journals (see: <https://www.nature.com/nature-portfolio/editorial-policies/plagiarism#policy-on-duplicate-publication> for details).

In recognition of the time and expertise our reviewers provide to Nature Immunology's editorial process, we would like to formally acknowledge their contribution to the external peer review of your manuscript entitled "Dermal macrophages set pain sensitivity by modulating tissue NGF levels through SNX25–Nrf2 signaling". For those reviewers who give their assent, we will be publishing their names alongside the published article.

Nature Immunology offers a Transparent Peer Review option for new original research manuscripts submitted after December 1st, 2019. As part of this initiative, we encourage our authors to support increased transparency into the peer review process by agreeing to have the reviewer comments, author rebuttal letters, and editorial decision letters published as a Supplementary item. When you submit your final files please clearly state in your cover letter whether or not you would like to participate in this initiative. Please note that failure to state your preference will result in delays in accepting your manuscript for publication.

Cover suggestions

As you prepare your final files we encourage you to consider whether you have any images or illustrations that may be appropriate for use on the cover of Nature Immunology.

Nature Immunology has now transitioned to a unified Rights Collection system which will allow our Author Services team to quickly and easily collect the rights and permissions required to publish your work. Approximately 10 days after your paper is formally accepted, you will receive an email in providing you with a link to complete the grant of rights. If your paper is eligible for Open Access, our Author Services team will also be in touch regarding any additional information that may be required to arrange payment for your article.

Please note that *Nature Immunology* is a Transformative Journal (TJ). Authors may publish their research with us through the traditional subscription access route or make their paper immediately open access through payment of an article-processing charge (APC). Authors will not be required to make a final decision about access to their article until it has been accepted. [Find out more about Transformative Journals](https://www.springernature.com/gp/open-research/transformative-journals).

If you have any questions about costs, Open Access requirements, or our legal forms, please contact ASJournals@springernature.com.

Authors may need to take specific actions to achieve [compliance with funder and institutional open access mandates](https://www.springernature.com/gp/open-research/funding/policy-compliance-faqs). If your research is supported by a funder that requires immediate open access (e.g. according to [Plan S principles](https://www.springernature.com/gp/open-research/plan-s-compliance)) then you should select the gold OA route, and we will direct you to the compliant route where possible. For authors selecting the subscription publication route, the journal's standard licensing terms will need to be accepted, including [self-archiving policies](https://www.springernature.com/gp/open-research/policies/journal-policies). Those licensing terms will supersede any other terms that the author or any third party may assert apply to any version of the manuscript.

Please use the following link for uploading these materials: [REDACTED]

Best regards,

Elle Morris
Senior Editorial Assistant
Nature Immunology
Phone: 212 726 9207
Fax: 212 696 9752
E-mail: immunology@us.nature.com

On behalf of

Ioana Visan, Ph.D.
Senior Editor
Nature Immunology

Tel: 212-726-9207
Fax: 212-696-9752
www.nature.com/ni

Reviewer #2:

Remarks to the Author:

The authors have appropriately addressed my comments experimental. While not statistically significant, likely due to a low $n=3$, the mixed BM chimera results are aligned with the previous findings. The supplemental data arising from the new experiments must be included as supplemental figures.

Reviewer #3:

Remarks to the Author:

The authors have addressed my points.

Final Decision Letter:

Subject: Decision on Nature Immunology submission NI-A32798C

Message: In reply please quote: NI-A32798C

Dear Dr. Tanaka,

I am delighted to accept your manuscript entitled "Dermal macrophages set pain sensitivity by modulating the amount of tissue NGF through a SNX25–Nrf2 pathway" for publication in an upcoming issue of Nature Immunology.

Over the next few weeks, your paper will be copyedited to ensure that it conforms to Nature Immunology style. Once your paper is typeset, you will receive an email with a link to choose the appropriate publishing options for your paper and our Author Services team will be in touch regarding any additional information that may be required.

Due to the importance of these deadlines, we ask that you please let us know now whether you will be difficult to contact over the next month. If this is the case, we ask you provide us with the contact information (email, phone and fax) of someone who will be able to check the proofs on your behalf, and who will be available to address any last-

minute problems.

Please note that *Nature Immunology* is a Transformative Journal (TJ). Authors may publish their research with us through the traditional subscription access route or make their paper immediately open access through payment of an article-processing charge (APC). Authors will not be required to make a final decision about access to their article until it has been accepted. [Find out more about Transformative Journals](https://www.springernature.com/gp/open-research/transformative-journals).

Your paper will be published online soon after we receive your corrections and will appear in print in the next available issue. Content is published online weekly on Mondays and Thursdays, and the embargo is set at 16:00 London time (GMT)/11:00 am US Eastern time (EST) on the day of publication. Now is the time to inform your Public Relations or Press Office about your paper, as they might be interested in promoting its publication. This will allow them time to prepare an accurate and satisfactory press release. Include your manuscript tracking number (NI-A32798C) and the name of the journal, which they will need when they contact our office.

About one week before your paper is published online, we shall be distributing a press release to news organizations worldwide, which may very well include details of your work. We are happy for your institution or funding agency to prepare its own press release, but it must mention the embargo date and *Nature Immunology*. Our Press Office will contact you closer to the time of publication, but if you or your Press Office have any enquiries in the meantime, please contact press@nature.com.

Also, if you have any spectacular or outstanding figures or graphics associated with your manuscript - though not necessarily included with your submission - we'd be delighted to consider them as candidates for our cover. Simply send an electronic version

(accompanied by a hard copy) to us with a possible cover caption enclosed.

Please note that we encourage the authors to self-archive their manuscript (the accepted version before copy editing) in their institutional repository, and in their funders' archives, six months after publication. Nature Portfolio recognizes the efforts of funding bodies to increase access of the research they fund, and strongly encourages authors to participate in such efforts. For information about our editorial policy, including license agreement and author copyright, please visit www.nature.com/ni/about/ed_policies/index.html

Sincerely,

Ioana Visan, Ph.D.
Senior Editor
Nature Immunology

Tel: 212-726-9207
Fax: 212-696-9752
www.nature.com/ni